# Exploring a blue-light-sensing transcription factor to double the peak productivity of oil in *Nannochloropsis oceanica*

Peng Zhang[1,2,3,6], Yi Xin[1,2,3,6], Yuehui He[1,2,3], Xianfeng Tang[3,4], Chen Shen[1,2,3], Qintao Wang[1,2,3], Nana Lv[1,2,3], Yun Li[1,2,3], Qiang Hu[2,5] & Jian Xu [1,2,3✉]

Oleaginous microalgae can produce triacylglycerol (TAG) under stress, yet the underlying mechanism remains largely unknown. Here, we show that, in *Nannochloropsis oceanica*, a bZIP-family regulator NobZIP77 represses the transcription of a type-2 diacylglycerol acyl-transferase encoding gene *NoDGAT2B* under nitrogen-repletion (N+), while nitrogen-depletion (N−) relieves such inhibition and activates *NoDGAT2B* expression and synthesis of TAG preferably from C16:1. Intriguingly, NobZIP77 is a sensor of blue light (BL), which reduces binding of NobZIP77 to the *NoDGAT2B*-promoter, unleashes NoDGAT2B and elevates TAG under N−. Under N+ and white light, *NobZIP77* knockout fully preserves cell growth rate and nearly triples TAG productivity. Moreover, exposing the *NobZIP77*-knockout line to BL under N− can double the peak productivity of TAG. These results underscore the potential of coupling light quality to oil synthesis in feedstock or bioprocess development.

---

[1] Single-Cell Center, CAS Key Laboratory of Biofuels, Shandong Key Laboratory of Energy Genetics and Shandong Energy Institute, Qingdao Institute of Bioenergy and Bioprocess Technology, Chinese Academy of Sciences, Qingdao, Shandong 266101, China. [2] Qingdao National Laboratory for Marine Science and Technology, Qingdao, Shandong 266237, China. [3] University of Chinese Academy of Sciences, Beijing 100049, China. [4] Qingdao Engineering Research Center of Biomass Resources and Environment, Qingdao Institute of Bioenergy and Bioprocess Technology, Chinese Academy of Sciences, Qingdao, Shandong 266101, China. [5] Institute for Advanced Study, Shenzhen University, Nanshan District, Shenzhen, Guangdong 518060, China. [6]These authors contributed equally: Peng Zhang, Yi Xin. ✉email: xujian@qibebt.ac.cn

Due to their high energy density, triacylglycerols (TAGs, or oils) are a universal energy storage form in the cell. They are the main constituent of plant vegetable oil and human body fat[1], and also precursors for biodiesels. Microalgae are promising feedstock for TAG production, due to their rapid autotrophic growth and high (up to 60%) oil content in the total biomass[2–4]. In oleaginous microalgae, oil accumulation is typically induced by environmental stresses, such as nutrient deprivation, high light, or heat[3,5,6]. The stress responses are foundation of a sustainable microalgal production scheme, yet exploitation of them for superior oil productivity has been hindered by lack of mechanistic insights. For example, nitrogen depletion (N-) has been intensively studied and practically used for triggering microalgal TAG accumulation[7,8]; however, despite the many studies in various microalgae profiling transcriptomics, proteomics, and metabolomics of N− induced TAG synthesis[9], as well as the plethora of transcriptional factors (TFs) predicted to be involved (e.g., *Tisochrysis lutea*[10] and *N. oceanica*[11]), the molecular mechanisms of signaling that links TAG synthesis to stress exposure have largely remained elusive.

Although effective in inducing microalgal TAG accumulation, depletion of N in culture as a way of process control (usually via natural consumption or transfer to N− medium) is inherently limited by difficulties in flexible, fast, and spatiotemporally precise control[3,12]. Moreover, nitrogen starvation usually leads to compromised photosynthesis and growth arrest[13], as N itself is a key nutrient for cells. Consistently, growth inhibition has frequently (although not always) been a side effect of metabolic engineering that boosts cellular TAG content[14] (e.g., resulting in ~25% biomass decrease[15,16]). Despite the progresses made in uncoupling TAG synthesis and cellular growth[17–22], the oil productivity of industrial microalgae remains too low to support a cost-competitive industry. Therefore, environmental stimuli that allow efficient and precise control of cellular TAG assembly yet without compromising biomass productivity are highly desirable.

Here, in the industrial oleaginous microalga *Nannochloropsis oceanica*, we discover that a TF NobZIP77 couples BL sensing to TAG production during N− induced oleaginousness. Exploiting this N− and BL co-regulated oleaginous mechanism, we show that a BL-induction strategy that exposes *NobZIP77*-knockout *N. oceanica* to white light and then to BL doubles the peak productivity of TAG. These findings demonstrate a strategy where cellular sensing of light quality is explored for microalgal feedstock development and photobioreactor design.

## Results

**Identifying NobZIP77 as a transcription factor regulating N− induced TAG synthesis**. In the industrial oleaginous microalga *N. oceanica* IMET1, 125 TFs were identified via characteristic domains of plant TFs, and a global regulation network predicted links of 35 TFs to 2801 target genes[11]. To pinpoint key TFs on N− induced TAG synthesis, we analyzed the transcript levels of these TFs over six timepoints (3, 4, 6, 12, 24, and 48 h) under N+ and N− conditions[23]. Altogether, 32 TFs respond to N−, with eleven upregulated and fourteen downregulated (seven of them showing distinct trends of regulation among timepoints). Of them, sixteen TFs were predicted to regulate lipid-related genes (LRGs; Fig. 1A, Supplementary Data 1)[11], and among them, scaffold00007.g77 (or g77, Genbank ID MT273120), which is downregulated by N−, targets seven such genes that include a diacylglycerol acyltransferases (DGAT) and a few additional enzymes that are related to lipid metabolism (Supplementary Fig. 1A, Supplementary Data 1). Notably, transcripts in the LRG of g77 exhibit distinct or even opposite fold-changes, e.g., upregulation of the lipase gene g10411 yet downregulation of the

DGAT gene of g6725 (Supplementary Data 1). As the activities of lipases and the DGAT are exactly opposite (oil degradation and assembly, respectively), these results indicate g77 as a negative regulator of TAG synthesis.

The *g77* gene encodes a 466-aa protein harboring three conserved domains: nuclear localization signal (NLS; 209–219 aa), Per-ARNT-Sim (PAS; 323–437 aa), and basic leucine zipper (bZIP; 246–284 aa; Fig. 1B). This bZIP domain, consisting of a DNA-binding region (246–254 aa) and a leucine zipper (255–284 aa), is characterized by an asparagine and an arginine residue that are conserved among higher plants, fungi, microalgae, and animals (thus g77 was termed NobZIP77; Supplementary Fig. 2A). Consistently, homologs of the NobZIP77-bZIP domain are broadly distributed in Stramenopiles, Viridiplantae, Fungi, and Metazoa (Supplementary Fig. 2B).

To probe its in vivo role, *NobZIP77* was overexpressed or knocked down in *N. oceanica* (Supplementary Fig. 3; "Methods"). Compared to the EV (empty-vector transformed) control, *NobZIP77* transcripts in overexpression lines (NobZIP77o-1 or NobZIP77o-2) increased by 1.7–2.2-fold, while in knockdown lines (NobZIP77i-1- or NobZIP77i-2) reduced by 21.0–31.7% at 0 h, 6 h, 24 h, and 48 h under N− (Supplementary Fig. 4A). Under N+, *NobZIP77* overexpression results in 9.03% and 1.23% lower growth rates (the average over the eight-day culture) than EV, while *NobZIP77*-knockdown lines remained unchanged (Supplementary Fig. 4B). Therefore, under such experimental conditions, *NobZIP77* expression level does not seem to exert significant effects on microalgal growth.

As for TAG content, significant difference was observed under N+ (i.e., at 0 h under N−). Specifically, in NobZIP77o-1 and NobZIP77o-2, TAG content is 22.3% and 35.2% lower than EV. In contrast, in NobZIP77i-1 and NobZIP77i-2, it is 53.3% and 196.7% higher than EV, respectively (Fig. 1C and Supplementary Fig. 4C). As for TAG yield, compared to EV, NobZIP77i-1 exhibited a 31.1% higher level under N+, while NobZIP77i-2 increased 31.8–197.1% during 0–72 h under N− (Fig. 1D; Supplementary Fig. 4D). Furthermore, in *NobZIP77*-knockdown lines, under N+, TAG-associated C16:1 is 59.0–106.4% higher, while C18:0 and C20:5 are 31.2–35.5% and 16.8–63.0% lower than EV, respectively. In contrast, in the *NobZIP77*-overexpression lines, compared to EV, TAG-associated C16:1 is 19.6–34.7% lower, while C18:0 and C20:5 are 48.6–66.5% and 41.2–89.2% higher (Supplementary Fig. 4E). As for total fatty acids (TFA) content, no significant difference from EV was observed in *NobZIP77* overexpression or knockdown lines (Supplementary Fig. 4F). Therefore, by increasing C16:1 while reducing C18:0 and C20:5 in TAGs, NobZIP77 negatively regulates TAG content and yield in *N. oceanica*, yet preserves its growth rate.

Notably, among the two overexpression lines (also among the knockdown lines), the expression pattern of *NobZIP77* is consistent, but not TAG content or yield. This can be due to the unpredictable effect of integration of overexpression cassette or silencing of the RNAi constructs. To settle this concern, *NobZIP77* was knocked out in *N. oceanica* (Supplementary Fig. 5A; "Methods"). Under N+, knockout lines showed essentially no effects on microalgal growth (Supplementary Fig. 6A). However, significant difference in TAG content was observed in NobZIP77-knockout lines. Specifically, versus the wild type (WT) and under N−, TAG content was 178.1% and 51.0% higher at 0 h and 24 h for NobZIP77ko-1, while 160.0% and 49.5% higher at 0 h and 24 h for NobZIP77ko-2 (Fig. 1E and Supplementary Fig. 6B). For TAG yield, *NobZIP77* knockout resulted in 61.6% and 63.6% increase at 0 h, respectively (52.7% and 44.4% at 24 h; under N−; Fig. 1F and Supplementary Fig. 6C). Notably, TAG-associated C18:0 and C18:2 are 26.0–32.4% and 59.0–59.5% lower in the *NobZIP77*-knockout

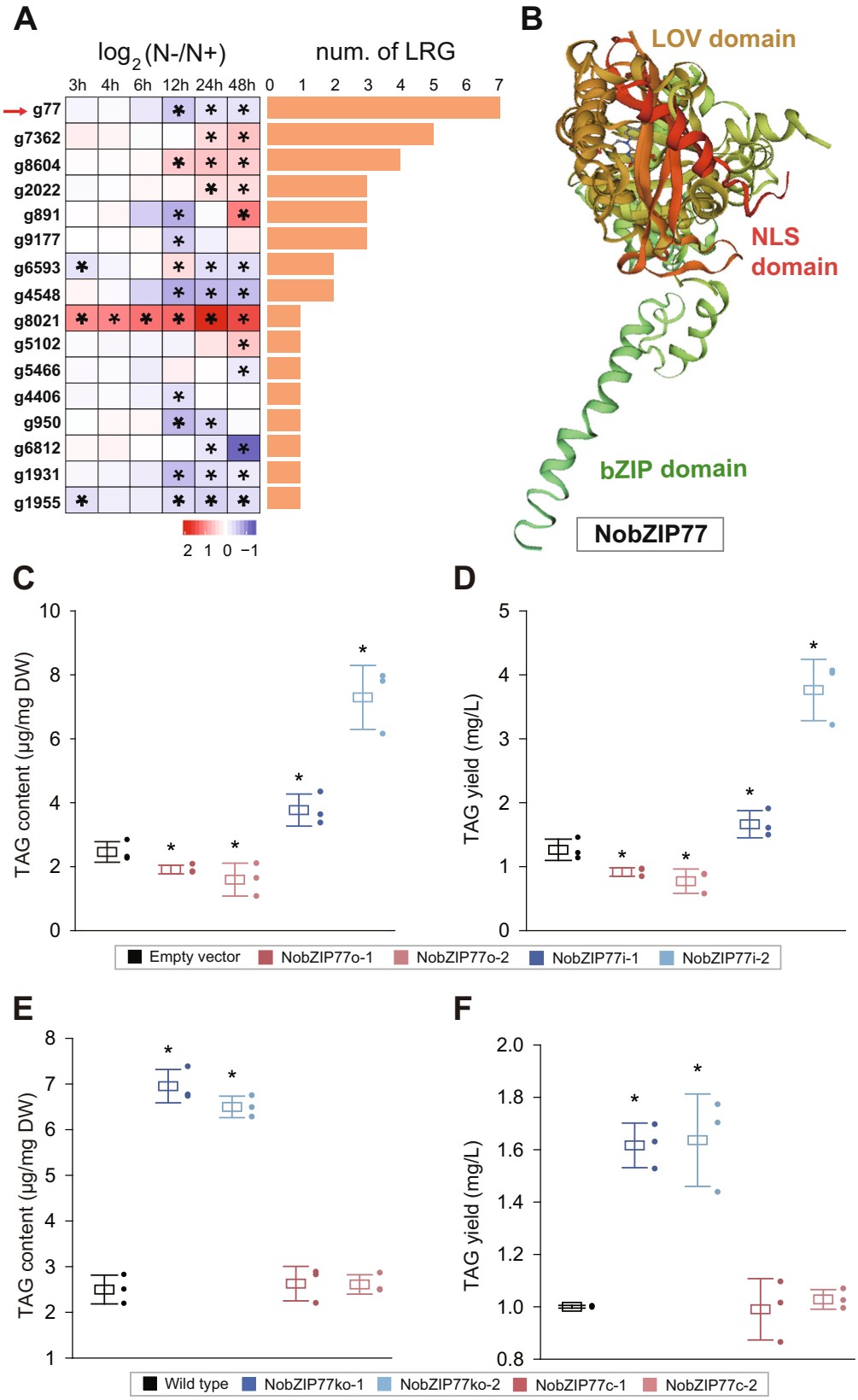

lines under N+ (versus WT; Supplementary Fig. 6D). As for TFA content, no difference was found versus WT via knockout, under either N− or N+ (Supplementary Fig. 6E). These results indicated that the knockout lines behave similarly as silencing lines. To further pinpoint its in vivo role, *NobZIP77* was genetically complemented in its knockout lines (Fig. 1D and Supplementary Fig. 5; "Methods"). Importantly, these complementation lines showed essentially identical phenotype to WT in *N. oceanica*. These results firmly pinpoint NobZIP77 as a TF that inhibits N− induced TAG synthesis.

**Fig. 1 Identification of NobZIP77 (g77) as a key TF on N− induced TAG synthesis. A** Transcriptomic response of *N. oceanica* during N− induced TAG synthesis at 3, 4, 6, 12, 24, and 48 h under N−. Fold change was calculated as $\log_2(\mathrm{FPKM(Tx, N-)}/\mathrm{FPKM\ (Tx, N+)})$ (FPKM = the normalized abundance of transcript, Tx = time point). Red arrow indicates NobZIP77. LRG, lipid-related gene. **B** Tertiary structural model of NobZIP77 (249–391), which was modeled using a Circadian locomotor output cycles protein kaput as initial template (15.44% sequence identity). **C** TAG content of the *NobZIP77* overexpression and knockdown lines under N+. **D** TAG yield of the *NobZIP77* overexpression and knockdown lines under N+. **E** TAG content of the *NobZIP77* knockout and complementation lines under N+. **F** TAG yield of the *NobZIP77* knockout and complementation lines under N+. Data are represented as mean ± SD ($n = 3$ biologically independent samples). *: significant change ($p \leq 0.05$) by one-sided Student's *t*-test versus N+ (**A**), EV (**C**, **D**), or WT (**E**, **F**). Source data are provided as a Source data file.

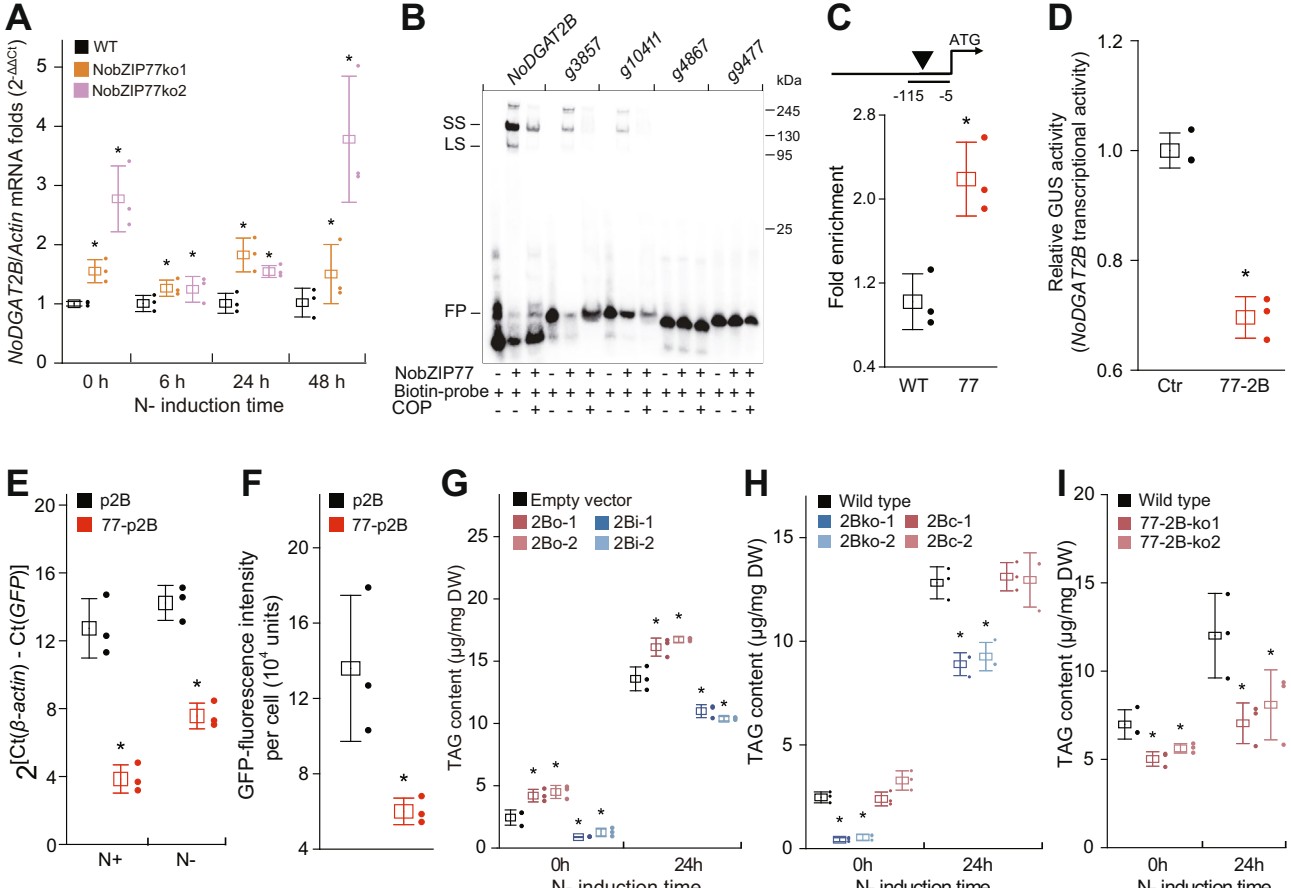

**Fig. 2 NobZIP77 inhibits *NoDGAT2B* transcription by binding to its promoter. A** Transcript level of *NoDGAT2B* in WT, NobZIP77ko-1, and NobZIP77ko-2 at 0 h, 6 h, 24 h, and 48 h under N−. WT, wild type. **B** EMSA validation of specific binding between NobZIP77 and the promoters of its targeted genes (i.e., *NoDGAT2B*, *g3857*, and *g10411*). *g3857*, TAG-lipase; *g10041*, lipase; *g4867*, hsp70; *g9477*, actin. Unlabeled DNA of the promoters in 100-fold molar excess was treated with the NobZIP77 protein. LS, lower shift complex (a 1:1 complex of DNA and the bZIP dimer); SS, super shift complex; FP, free-DNA probe; COP, competitor oligonucleotide primer. The experiments were repeated three times. **C** ChIP-qPCR analysis of NobZIP77 binding to the promoter of *NoDGAT2B* (p*NoDGAT2B*) in vivo. In p*NoDGAT2B*, the region used for ChIP-qPCR is marked (transverse line, top panel). The bZIP binding motif is indicated as arrowheads. The numbers in fragment indicate positions of the nucleotides at the 5′ or 3′ end of the fragment relative to the translation start site. β-actin was used as an internal reference gene. WT, wild type; 77, *gfp:NobZIP77*-overexpression line. **D** The enzymatic activities of glucuronidase (GUS) between NobZIP77-p*NoDGAT2B* co-transfected or p*NoDGAT2B* transfected *Arabidopsis* leaf protoplasts. The luciferase (LUC) was used as a control. Ctr, p*NoDGAT2B* transfected *Arabidopsis* leaf protoplasts; 77-2B, NobZIP77-p*NoDGAT2B* co-transfected *Arabidopsis* leaf protoplasts. **E** Comparison of *GFP* transcript level between the *NoDGAT2B*-promoter transformed (p2B) and the NobZIP77-*NoDGAT2B*-promoter transformed (NobZIP77-p2B) lines of *N. oceanica*, under N + or N− (24 h). **F** Quantification of GFP fluorescence (average fluorescence intensity per cell) in the p2B and NobZIP77-p2B lines. **G** TAG content of the *NoDGAT2B* overexpression and knockdown lines in *N. oceanica*. **H** TAG content of the *NoDGAT2B* knockout and complementation lines. **I** TAG content of the *NobZIP77* and *NoDGAT2B* double-knockout lines. Data are represented as mean ± SD ($n = 3$ biologically independent samples). *: significant change ($p \leq 0.05$) by one-sided Student's *t*-test versus WT (**A**, **C**, **H**, **I**), LUC (**D**), p2B (**E**, **F**), or EV (**G**). Source data are provided as a Source data file.

***NoDGAT2B* is a key target of NobZIP77.** To identify the binding sites of NobZIP77, a phylogenomic footprinting pipeline was developed via MERCED[24], revealing seven LRGs (Supplementary Data 1). These LRGs encode multiple TAG/FA metabolism-related enzymes, including DGAT, phosphatidic acid phosphohydrolase (PAP), fatty acid elongase (FAE), and lipase.

To validate such regulatory links, the transcript levels of all seven LRGs were quantified by RT-qPCR in the *NobZIP77* over-production, knockdown, and knockout lines (Fig. 2A; Supplementary Fig. 7; Supplementary Fig. 8). Notably, *NoDGAT2B* (*g6725*; Genbank ID KX867957), a member of the *N. oceanica* type-2 DGAT family[21,22], is the only LRG that exhibits a

temporal transcription pattern precisely opposite to *NobZIP77* (consistent with the transcriptome data; Fig. 2A; Supplementary Fig. 7; Supplementary Data 1). The *NoDGAT2B* transcript is 50.4% higher in NobZIP77ko-1, 128.9% higher in NobZIP77ko-2, 127.5% higher in NobZIP77i-2, while 26.0% lower in NobZIP77o-2 (average fold change over 0 h, 6 h, 24 h, and 48 h under N−; versus EV; Fig. 2A; Supplementary Fig. 8), which suggests the repression of *NoDGAT2B* by NobZIP77. Such a trend of transcript fold change is also shared by the PAP of g2171 and the lipase of g4187 at the early phase of N− (Supplementary Fig. 8). Thus, *NoDGAT2B* is one key target of NobZIP77.

To probe this hypothesis and test whether NobZIP77 directly interacts with the promoter of *NoDGAT2B* (p*NoDGAT2B*; and of additional predicted target genes or PTGs), electrophoretic mobility shift assays (EMSAs) were used to identify specific band shifts of ACGT-harboring promoter probes (the binding sites of bZIP-type TF genes[25]). Recombinant maltose-binding protein (MBP)—NobZIP77 was expressed in *Escherichia coli* and purified for EMSA ("Methods"). ACGT-harboring probes of the NobZIP77-PTG promoters [g3857 (*TAG-lipase*), g2171 (*PAP*), g10041 (*lipase*), *NoDGAT2B*, and g5641 (*FAE*)] result in mobility shift, indicating the direct binding between the NobZIP77-PTG promoters and NobZIP77 (Supplementary Fig. 9). Notably, the degree of mobility shift is greatly reduced by adding unlabeled fragment of NobZIP77-PTG promoter, yet no binding between NobZIP77 and the negative controls of g4867 (*hsp70*) and g9477 (*actin*) was detected (Fig. 2B), which supports the specificity of binding between NobZIP77 and p*NoDGAT2B*.

Next, we tested the in vivo binding between NobZIP77 and p*NoDGAT2B*, by chromatin immunoprecipitation (ChIP) qPCR assays in *N. oceanica*. A GFP gene fused immediately downstream of the full-length *NobZIP77* cDNA was transformed into *N. oceanica* and positive expression in transgenic lines (OE1) was confirmed with laser-scanning confocal microscopy (Supplementary Fig. 10). Under N+, OE1 and WT cells (cell density of $2 \times 10^7$ cells/mL) were subject to chromatin extraction and immunoprecipitation with anti-GFP antibody ("Methods"). The ChIP products were analyzed by qPCR and the fold change of p*NoDGAT2B* amount calculated via $2^{-\Delta\Delta CT}$ (*β-actin* as internal reference). Notably, the amount of p*NoDGAT2B* in OE1 is 2.2-fold of that in WT (Fig. 2C), supporting the direct binding of NobZIP77 to the ACGT-harboring p*NoDGAT2B* sequence in vivo.

Furthermore, to probe the functional consequence of the binding between NobZIP77 and p*NoDGAT2B*, transient *NoDGAT2B* transcription level was measured via a reporter-effector system in *Arabidopsis* mesophyll protoplast (TEAMP; "Methods")[26]. *Arabidopsis* mesophyll protoplasts were employed in transient analysis of promoter activity in vivo, and β-glucuronidase (GUS) used as reporter for interaction with effector. Specifically, the 900 bp p*NoDGAT2B* was linked to the GUS reporter gene to create the reporter construct (p*NoDGAT2B* promoter-:*GUS*), and the full-length *NobZIP77* coding sequence (CDS) was ligated downstream of the 35S promoter to generate the effector construct (*CaMV35S* promoter-:*NobZIP77*). The two constructs were co-transfected into *A. thaliana* leaf protoplasts and the GUS activity measured ("Methods"). These ex vivo experiments revealed that relative GUS activity of the reporter-effector co-transfected protoplasts was 31.9% lower than the protoplasts transfected with the reporter construct alone, indicating an inhibitory effect on *NoDGAT2B* transcription due to the binding of NobZIP77 to p*NoDGAT2B* (Fig. 2D).

To test whether the NobZIP77-p*NoDGAT2B* binding inhibits *NoDGAT2B* transcription in vivo, a GFP reporter system was designed for quantifying p*NoDGAT2B*-driven gene expression ("Methods"). To avoid potential interference of the inhibition by

the endogenous NobZIP77, this experiment was run in a *NobZIP77*-knockout line of *N. oceanica* (NobZIP77ko1). A control vector pXJ545 was constructed with a core cassette of p*NoDGAT2B-GFP-Ttub-Ptub-ble-Tvcp* (Supplementary Fig. 3F) and a reporter vector pXJ546 designed with a core cassette of p*NoDGAT2B-GFP-Ttub-Ptub-NobZIP77-TpsbA-Ptub-ble-Tvcp* (Supplementary Fig. 3G). In all NobZIP77ko1 lines transformed with pXJ545 and pXJ546, both GFP transcription level and fluorescence intensity were measured. In NobZIP77-p2B (NobZIP77-p*NoDGAT2B*-transformed *N. oceanica*) and versus p2B (p*NoDGAT2B*-transformed line), the *GFP* transcript level was reduced by 69.7% under N+ and 58.8% under N− (Fig. 2E), whereas fluorescence intensity of GFP was reduced by 55.8% (under N+; Fig. 2F and Supplementary Fig. 11). These results support the inhibitory effect of NobZIP77-p*NoDGAT2B* binding on *NoDGAT2B* transcription in vivo. Collectively, these in vitro, ex vivo, and in vivo evidence suggest that, in *N. oceanica*, NobZIP77 inhibits *NoDGAT2B* expression by binding to its promoter via the ACGT core sequence.

The 346-aa NoDGAT2B is encoded via two exons and contains two transmembrane domains. We have validated the TAG-synthetic function of five NoDGAT2s via ex vivo assays in *Saccharomyces cerevisiae* strain H1246, a TAG-deficient quadruple knockout mutant[21,22], yet TAG was not detected in *NoDGAT2B*-containing H1246, probably due to the absence in yeast of proper subcellular structure or cofactor for TAG synthesis. Meanwhile, in vitro assay of NoDGAT2B was unavailable due to the failure of microsome acquirement, thus we turned to the direct characterization of NoDGAT2B function in vivo. *NoDGAT2B* was overexpressed or knocked down (via RNAi) in *N. oceanica* ("Methods"). Compared to EV, *NoDGAT2B* transcripts increased by 4.8–8.7-fold in its overexpression lines 2Bo1 and 2Bo2, while reduced by 41.9–59.0% in the knockdown lines of 2Bi1 and 2Bi2, under N− (Supplementary Fig. 12A). Moreover, under N+ and at the early phase of N−, TAG content is 18.7–71.9% and 23.2–83.9% higher in 2Bo1 and 2Bo2, respectively, while 19.1–64.3% and 23.6–48.9% lower for 2Bi1 and 2Bi2 (versus EV; Fig. 2G, Supplementary Fig. 12B, C). Notably, such difference in TAG content versus EV disappears at the late phase of N− (after 48 h), which indicates the role of other NoDGAT2s in this process. Furthermore, in the *NoDGAT2B*-overpression lines, under N+, TAG-associated C16:1 is 29.3% and 30.0% higher, while C18:0 and C20:5 are 52.9–53.0% and 29.6–29.8% lower than EV. On the opposite, in the *NoDGAT2B*-knockdown lines, TAG-associated C16:1 is 32.3% and 45.1% lower, while C18:0 and C20:5 are 82.1–172.0% and 49.9–106.5% higher than EV (Supplementary Fig. 12D). Thus, the phenotypes of *NobZIP77* and *NoDGAT2B* manipulation are opposite, further supporting the negative regulation of *NoDGAT2B* by NobZIP77.

To further pinpoint its in vivo role, *NoDGAT2B* was knocked out via CRISPR/Cas9 (Supplementary Fig. 13) and complemented in *N. oceanica* (Supplementary Fig. 3; "Methods"). No effects on growth were observed (Supplementary Fig. 14A), yet TAG content changed upon knockout and recovered upon complementation. Specifically, under N−, TAG content is 30.6–82.6% lower in 2Bko-1, and 27.8–78.2% lower for 2Bko-2 (versus WT; Fig. 2H; Supplementary Fig. 14B). As for TAG yield, *NoDGAT2B* knockout led to 26.8–56.5% and 29.5–58.4% reduction, respectively, under N− (Supplementary Fig. 14C). Moreover, the knockout resulted in 2.4–3.8% and 51.5–60.1% lower TAG-associated C16:0 and C16:1, plus 42.4–48.8%, 403.0–571.9%, 124.2–135.2% and 124.3–189.6% higher TAG-associated C18:0, C18:3, C20:4 and C20:5 (under N+, Supplementary Fig. 14D). As for TFA content, no changes were found (Supplementary Fig. 14E). These results are consistent with the *NoDGAT2B* knockdown and overexpression experiments.

Moreover, to prove that differential regulation of *DGAT2B* is relevant for the lipid accumulation phenotypes of the *NobZIP77*-knockout and overexpression lines, we genetically knocked out *NoDGAT2B* via CRISPR/Cas9 in the *NobZIP77*-knockout line of NobZIP77-ko1 (Supplementary Fig. 15; "Methods"). As compared to WT, the resulted double-knockout lines of NobZIP77-2B-ko1 and NobZIP77-2B-ko2 both exhibit no change in growth rate (Supplementary Fig. 16A), yet significant decrease in both TAG content and TAG yield. Specifically, at 0 h and 24 h under N−, the TAG content is 28.1% and 41.3% lower in NobZIP77-2B-ko1 (19.4% and 32.6% lower for NobZIP77-2B-ko2; versus WT; Fig. 2I and Supplementary Fig. 16B). As for the TAG yield, at 0 h and 24 h under N−, the double knockout led to 27.6% and 46.7% reduction in NobZIP77-2B-ko1 (21.8% and 42.6% in NobZIP77-2B-ko2; versus WT; Supplementary Fig. 16C). In addition, for NobZIP77-2B-ko1 and NobZIP77-2B-ko2, under N+, TAG-associated C16:1 is 16.6 and 29.3% lower, yet TAG-associated C18:0, C18:3, C20:4 and C20:5 are 29.6 and 40.7%, 58.3 and 105.3%, 67.9 and 111.0%, and 34.5 and 55.9% higher, respectively (Supplementary Fig. 16D). As for TFA content, no changes were found in either double-knockout lines (except for 48 h under N−; Supplementary Fig. 16E). These TAG-related phenotypes due to the *NobZIP77* and *DGAT2B* double-knockout are like those resulted from *NoDGAT2B* knockout (Supplementary Figs. 14 and 16), while opposite to those from *NobZIP77* knockout (Supplementary Figs. 6 and 16).

**NobZIP77 as a blue-light sensor that releases the *NoDGAT2B* promoter under blue light.** NobZIP77 harbors a conserved light–oxygen–voltage (LOV) domain, which belongs to the PAS superfamily that specifically senses blue light (BL) by a non-covalently bounding to flavin cofactor [FMN (flavin mononucleotide); Supplementary Fig. 17][27]. For NobZIP77-LOV, a series of key residues are conserved among its homologs in higher plants, fungi, and microalgae (Supplementary Fig. 18A, B). Together, the features of the bZIP and LOV domains and the NCR-Q-V-IR-N-N-F-Q conserved residues (Supplementary Fig. 18C) support NobZIP77 as an AUREO1-subfamily member in AUREOCHROME (AUREO), which is a type of BL receptors in stramenopiles[28], and raise the hypothesis that it regulates TAG assembly in response to BL. Moreover, its characteristic order of the sensor and effector domains, with the sensor domain (LOV) at the C-terminus and the effector domain (bZIP) at the N-terminus, strongly suggests NobZIP77 as an aureochrome[28].

To probe the effect of BL on NobZIP77-*NoDGAT2B* interaction, EMSA was performed in the dark, or under continuous illumination of white light (WL) or BL (Supplementary Fig. 19; "Methods"). The shifting of NobZIP77-p*NoDGAT2B* band was observed starting at the NobZIP77 concentration of 0.012 mM under the dark, at 1.5 mM under BL, and at 0.75 mM under WL (Fig. 3A). Thus, the binding between NobZIP77 and p*NoD-GAT2B*, which shuts down *NoDGAT2B*-mediated TAG assembly, can be relieved by BL (and to a lesser degree, by WL).

To probe the in vivo effect of BL on the NobZIP77-*NoDGAT2B* interaction, multiple light wavelengths were tested in cultivating *N. oceanica* WT and NobZIP77ko-1 (Fig. 3B). For WT, under BL (but not green light (GL)), the *NoDGAT2B* transcript is 1.6- and 13.3-folds higher than under the dark (for both N+ and N−; quantified via qRT-PCR; Fig. 3C). Intriguingly, for NobZIP77ko1, no change of *NoDGAT2B* transcript was detected under either BL or GL versus the dark (Fig. 3E). Therefore, BL but not GL can specifically induce NobZIP77-inhibited *NoDGAT2B* expression.

To probe the consequence of such BL-specific induction, the TAG content of *N. oceanica* quantified via single-cell Raman spectra (SCRS[29]; "Methods"). In WT, under N−, TAG content is

65.89% higher under BL, 25.34% higher under GL and 3.82% lower in the dark (versus N+, Fig. 3D), suggesting BL/GL (but not the dark) promotes TAG production under N−, with BL exerting much stronger stimulatory effect than GL. Moreover, in NobZIP77ko-1, TAG content is reduced by 11.78% under BL plus N−, revealing a temporal lag of TAG content in response to BL under N− versus WT (p < 0.05; Fig. 3D–F), despite a similar light-quality-responsive pattern, i.e., stronger stimulation of TAG synthesis by BL than by GL (for NobZIP77ko-1, under N−, TAG content is 19.69% and 49.83% higher under GL and BL, respectively, versus the dark; Fig. 3F). Therefore, NobZIP77 mediates N− induced TAG synthesis via its sensing of BL.

**A working model of stress-induced TAG synthesis mediated by NobZIP77 in *N. oceanica*.** To probe how the interactions among BL, NoZIP77, and TAG synthesis take place, we probed the subcellular locations of NobZIP77 and NoDGAT2B by fusing the GFP-coding gene (*gfp*) immediately downstream of the full-length *NobZIP77* or *NoDGAT2B* cDNA and transforming them into *N. oceanica*, respectively. The NobZIP77:gfp fusion protein is colocalized with the DAPI-stained nuclei, indicating nucleus targeting (Fig. 4A), while the NoDGAT2B:gfp fusion protein is colocalized with the Nile red-stained lipid droplet (LD), suggesting LD targeting (Fig. 4B). The nucleus is physically surrounded by the plastid[30] (Fig. 4A, B), which as the dominating sink of chlorophyll *a* (*N. oceanica* does not harbor chlorophyll *b*) can shield nucleus from BL by absorbing the 420–663 nm wavelength[31]. Thus, when chlorophyll *a* dramatically declines under N− (Fig. 4C), the nucleus would be exposed to more BL, which via the LOV domain of NobZIP77 triggers release of this TF (via allosteric effects such as de-dimerization[32]) from p*NoDGAT2B*.

Taken together, we propose an in vivo mechanistic model of N− induced TAG synthesis in *N. oceanica* (Fig. 4D). NobZIP77 normally (i.e., N+) represses *NoDGAT2B* transcription by directly binding to the latter's promoter; yet under N−, the plastid is no longer able to shield the NobZIP77-located nucleus from BL (due to the degradation of chlorophyll *a* in plastid), and the resulted exposure of NobZIP77 to BL reduces its binding to p*NoDGAT2B* and elevates expression of the enzyme for TAG assembly preferably from C16:1. Moreover, the level of *NobZIP77* transcript is downregulated under N− (Fig. 1A), which further facilitates expression of its downstream TAG-synthetic genes. Although additional mechanisms might be present, this model can explain how both BL exposure and *NobZIP77* downregulation eventually lead to elevated expression of the enzyme that assemblies TAG preferably from C16:1.

**A rational approach to elevate the peak productivity of oil via blue light.** The above discoveries inspired us to devise a strategy for superior TAG production, by simultaneously tuning four key factors: light quality (WL/ BL), *NobZIP77* expression (WT/ NobZIP77ko-1), nitrogen supply (N+/N−), and time of TAG harvest ("Methods"). For the light-quality factor, WL and BL were adopted in wild-type cultivation. Compared to WL, (1) under BL and N+, TAG content increased by 33.7% at 24 h (Supplementary Fig. 20A). (2) under BL and N−, TAG content increased by 6.9–42.9% from 12 h to 168 h (the highest improvement at 72 h), and TAG yield increased by 3.6–48.3% from 6 h to 96 h (the highest improvement at 48 h; Fig. 5A). Meanwhile, microalgal growth rate remained unchanged under BL, versus WL (Supplementary Fig. 20A–C). Therefore, consistent with existing results (Fig. 3D), BL is a key factor in TAG improvement without inhibiting growth, especially under N−.

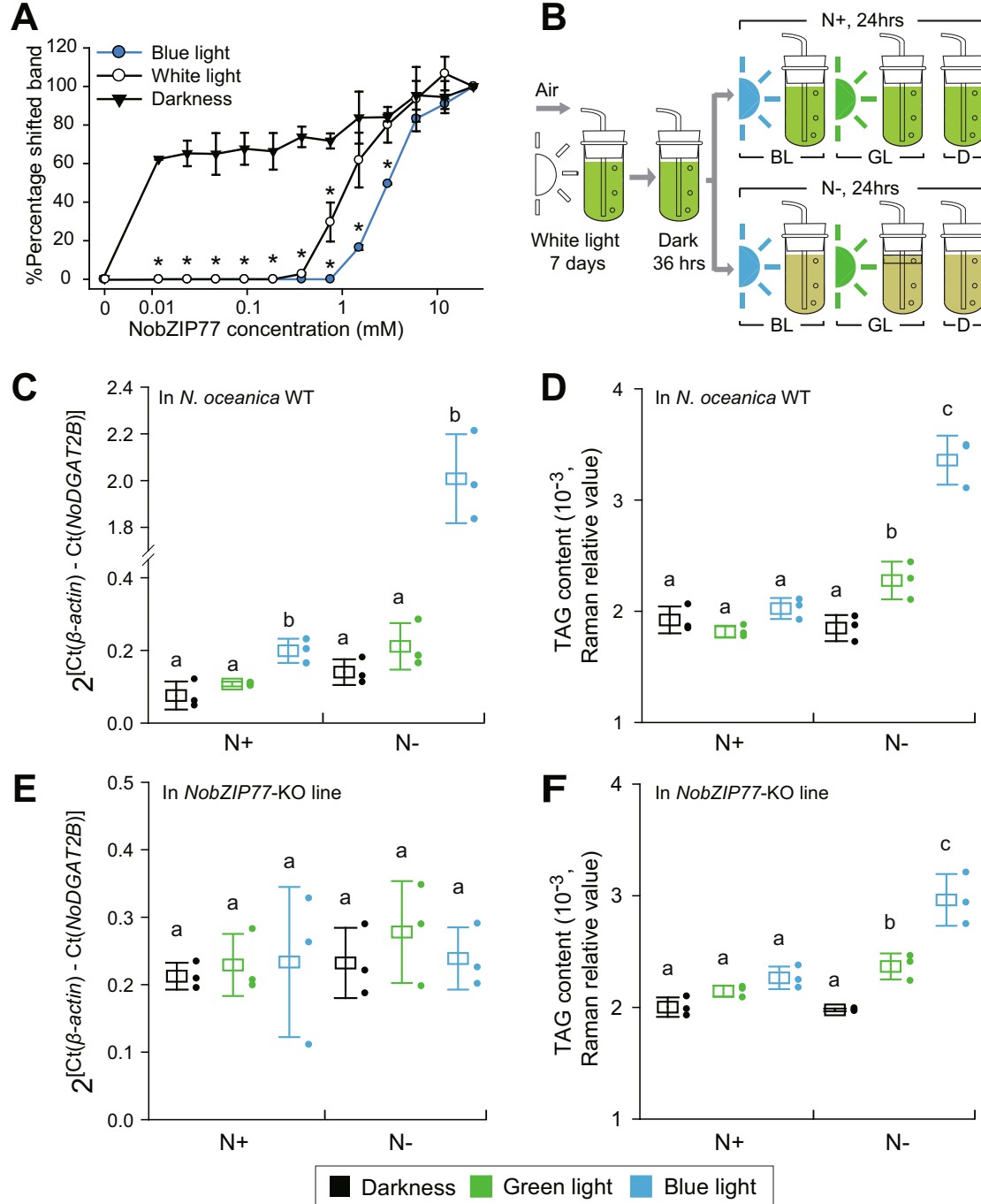

**Fig. 3 Blue light reduces the binding of NobZIP77 to the *NoDGAT2B* promoter. A** DNA-binding curves of NobZIP77 under the dark, white light (WL) or blue light (BL). The curves were generated by quantifying the relative amount of shifted bands of NobZIP77 bound to p*NoDGAT2B* in EMSA. The samples derive from the same experiment and that gels/blots were processed in parallel. Data are represented as mean ± SD ($n = 2$ biologically independent samples). *: significant change ($p \le 0.05$) by one-sided Student's *t*-test versus darkness. **B** Experimental design that probes the specificity of BL to the NobZIP77-mediated *NoDGAT2B* expression and TAG synthesis in vivo, under N+ or N−. BL, blue light; GL, green light; D, darkness. **C, E** Transcript level of *NoDGAT2B* in WT (**C**) and in the knockout mutant of NobZIP77ko-1 (**E**). **D, F** TAG content of WT (**D**) and NobZIP77ko-1 (**F**). Data are represented as mean ± SD ($n = 3$ biologically independent samples). Letters above the bars indicate significant difference ($p \le 0.05$), based on one-way analysis of variance (ANOVA) and Tukey's honestly significant difference (HSD) test. Source data are provided as a Source data file.

For the *NobZIP77*-expression factor, under WL and N+, compared to WT, TAG content of NobZIP77ko-1 increased by 72.5–225.7% from 24 h to 264 h (the highest improvement at 264 h; Supplementary Fig. 20B), and TAG yield of NobZIP77ko-1 increased by 60.9–184.5% from 24 h to 264 h (the highest improvement at 216 h) (Fig. 5B). Meanwhile, no slowdown in microalgal growth was apparent. Under N−, the TAG content,

TAG yield, and growth of NobZIP77ko-1 were all equivalent to WT (Supplementary Fig. 20B). Therefore, knocking out *NobZIP77* can greatly promote TAG synthesis yet without inhibiting growth (under N+), consistent with the above findings (Fig. 3F).

We next tested the link between BL and *NobZIP77*-knockout, by comparing NobZIP77ko-1 under BL (BL-KO) to WT under WL (WL-WT) (Supplementary Fig. 20C). Under N+, the TAG

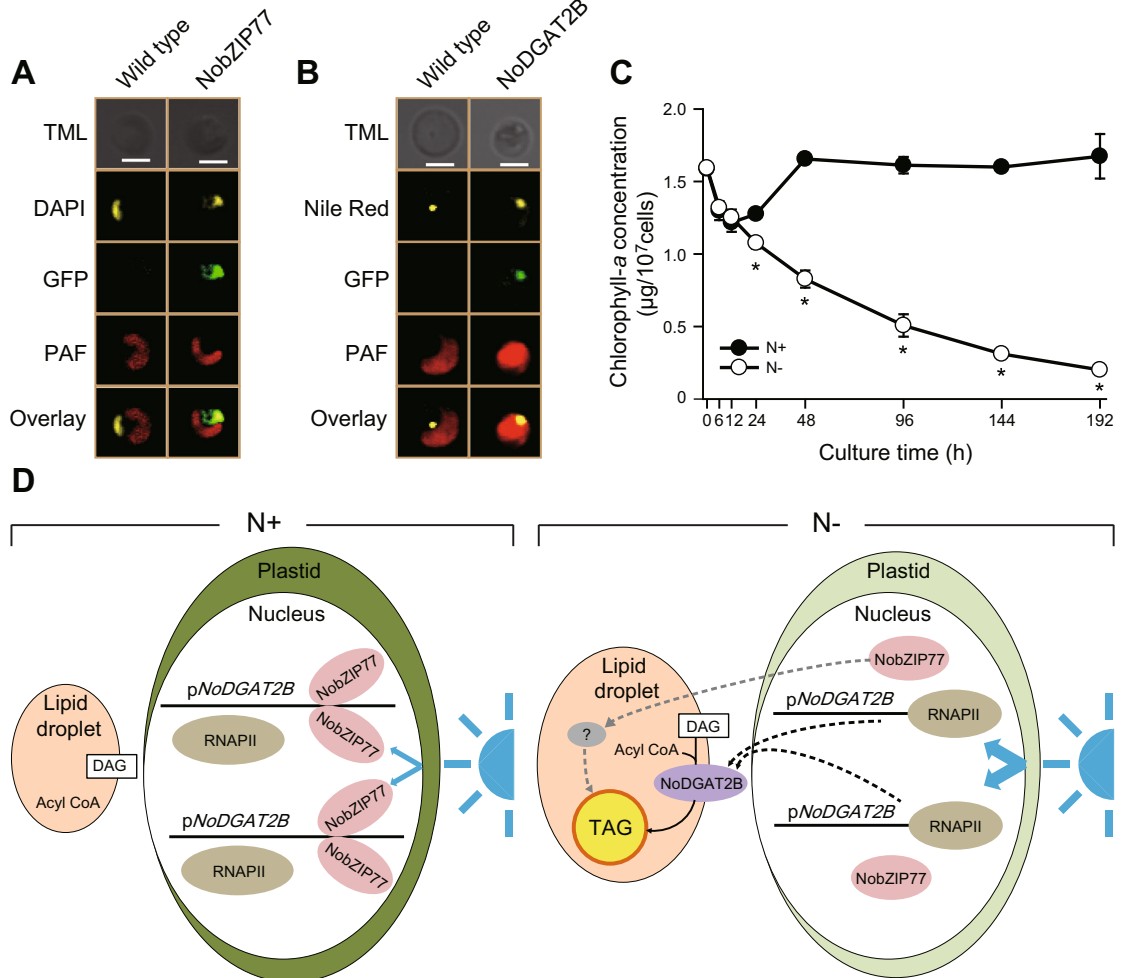

**Fig. 4 A working model of nitrogen-depletion-induced TAG synthesis in *Nannochloropsis oceanica*. A, B** Subcellular localization of NobZIP77 (**A**) and NoDGAT2B (**B**). GFP, green fluorescent protein; TML, transmission light; DAPI, 4',6-diamidino-2-phenylindole; PAF, plastid autofluorescence; scale bar, 2 μm. The experiments were repeated three times. **C** Chlorophyll *a* levels in the *N. oceanica* WT under N+ and N−. Data are represented as mean ± SD (*n* = 3 biologically independent samples). *: significant change (*p* ≤ 0.05) by one-sided Student's *t*-test versus N+. **D** The proposed model, in which blue light promotes TAG production under N−, by inhibiting the binding between NobZIP77 and p*NoDGAT2B* (details described in "Results"). RNAPII, RNA polymerase II; DAG, diacylglycerol; p*NoDGAT2B*, the *NoDGAT2B* promoter. Source data are provided as a Source data file.

content of BL-KO increased by 83.6% and 98.4% at 24 h and 144 h, and the TAG yield increased by 73.4% and 50.5% at 144 h and 216 h. Under N−, the TAG content of BL-KO increased by 17.2−35.7% from 72 h to 120 h (the highest improvement at 72 h), and the TAG yield increased by 35.3% and 21.1% at 72 h and 96 h (Fig. 5C). Notably, the growth rate of BL-KO remained equivalent to WL-WT, under both N+ and N−. Moreover, versus BL-WT or WL-KO, the TAG content, the TAG yield, and growth rate of BL-KO exhibited no change (under N+ and N−). Since BL and *NobZIP77*-knockout can relieve the repressive effect of NobZIP77 and activate TAG synthesis, yet *NobZIP77*-knockout eliminates such an effect of BL (i.e., they are not additive) under N+, BL and NobZIP77 are likely orthogonal in this oleaginous signaling (Fig. 4D).

Finally, the observations that elevation of TAG by BL under N− yet by *NobZIP77*-knockout under N+ allowed us to propose a BL-induced oil-production (BLIO) strategy (Fig. 6A). Specifically, the NobZIP77ko-1 line was cultivated under WL for nine days (corresponding to peak yield of TAG under N+; Fig. 5B) and then with BL for another seven days (i.e., when N in medium is gradually naturally depleted). As a control, WT was cultivated for 16 days under WL (Fig. 6A). Microalgal growth rate for BLIO

is equivalent to the control (Supplementary Fig. 21), yet the TAG yield is elevated in a rather consistent manner along the cultivation course (e.g., increases by 103.5% on Day 16; Fig. 6B). Moreover, the TAG productivity of BLIO doubles that of the control (on average 91.1% higher; Fig. 6C). Therefore, via both genetic and process engineering, we have established a rational approach that doubles the peak productivity of oil in *N. oceanica*.

## Discussion

The BL-NobZIP77-DGAT2B working model is unexpected, since among the eleven putative type-2 DGATs in *N. oceanica*, extensive ex vivo and in vitro assays revealed only NoDGAT2A, 2C, 2D, 2K, and 2J as TAG-synthetic[16,21,22], but not 2B. However, clearly, NoDGAT2B is a potent gene of oil traits (as its overproduction elevated TAG content by 163.0% while knockdown reduced it by 70.2%; Fig. 2D). Moreover, NoDGAT2B's fatty acid (FA) preference is unusual, featuring strong carbon chain−length specificity: for C16:1 yet against C18:0, C18:2, and C20:5 (Supplementary Fig. 12D); in contrast, 2A, 2C, 2D, 2K, and 2J distinguish FA substrates based on degree of unsaturation[21,22]. Therefore, while raising the question of why it (and its TAG

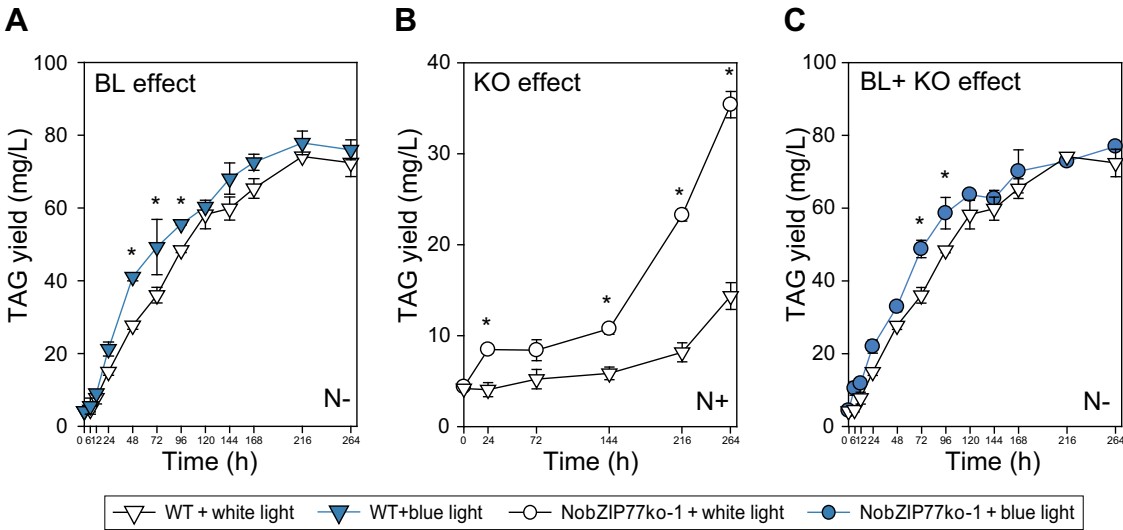

**Fig. 5 A rational approach to elevate oil productivity by exploiting the NobZIP77 signaling.** Three independent cultures were sampled for a period of 11 days. For each of the three independent sets of samples, TAG contents and yields were quantified by single-cell Raman microspectroscopy, based on the protocols that we have published. The results yield very small error bars. TAG yield (mg/L) was compared between wild-type (WT) under white light (WL) and blue light (BL) plus N− (**A**), WT and NobZIP77ko-1 (KO) under WL and N+ (**B**), or WT-BL and KO-BL under N− (**C**). Data are represented as mean ± SD ($n = 3$ biologically independent samples). *: significant change ($p \leq 0.05$) by one-sided Student's $t$-test versus WT-WL. Source data are provided as a Source data file.

products) is specifically induced by BL, NoDGAT2B adds one dimension to the generation of "designer oils"[33].

However, this BL-induced oleaginousness mechanism can potentially inform how microalgae adapt to aquatic ecosystems. Stramenopiles have no flagella yet can exhibit BL-induced movement (e.g., in the xanthophyte *Vaucheria frigida*[32,34]), and the mechanism is unknown. Considering that generally, only BL (among the bio-exploitable wavelength) can penetrate substantial depths in water[35], aureochromes are only found in stramenopiles so far[36] and TAG content and composition can influence cellular buoyancy[37], it is possible that the BL-NobZIP77-NoDGAT2B working model might underpin such phototropic behaviors by controlling microalgal floating depth (via modulating TAG content), to maximize solar-energy harvest or minimize photo-destructive effects (e.g., UV-induced DNA damage).

A TF such as NobZIP77 that responds to both light and N− while directly regulates TAG assembly is extraordinary. In microalgae, a large number of TFs potentially responsive to N− have been identified via transcript-level change under N−[11,15,18], however, none of them were known to directly control LRGs[38]. For example, in *N. gaditana*, a TF of the $Zn_{(II)2} Cys_6$ family was found implicated in N− induced lipid production, and its knockdown by RNAi resulted in double lipid production yet decreased microalgal growth rate[15]; however, neither its downstream target nor the signaling mechanism for its response to N depletion is clear. On the other hand, light-quality-sensing TFs have been reported in *Neurospora*[39] yet none were known to directly regulate TAG synthesis. Thus, NobZIP77 can directly convey light-related signals (i.e., not just light quality) to the transcription of metabolite-synthetic enzymes. As importantly, NobZIP77 appears to be a major master regulator that is lipid-specific, as (1) the promoters of at least seven LRGs carry its binding motifs, (2) its knockout nearly triple TAG under N+, and (3) its loss fully preserves microalgal growth rate. Therefore, illustration of the NobZIP77 regulatory network may present ample opportunities for feedstock development.

Notably, BL greatly promotes microalgal TAG production: In fact, *NobZIP77*-knockout *N. oceanica* exposed to WL first and then to BL more than doubles the peak productivity of TAG (Fig. 6B). Such light-quality-based oil harvesting strategies are perhaps widely applicable, as homologs of the full-length Nob-ZIP77 are widely found in stramenopiles[36]. Light-quality control, with its fast excitation, high temporal and spatial precision, reversibility, expandability, and energy efficiency[40], presents an under-explored yet highly attractive opportunity for bioreactor and bioprocess design.

In animal, plant, and microbes, known pathways that regulate metabolite synthesis in response to light quality usually consist of multiple layers of signal transduction[41]. On the other hand, aureochromes, although known to regulate photomorphogenesis, sexual cycle, and biological rhythm[28], have rarely been implicated in the synthesis of carbon-storage compounds (although strong metabolic shift takes place upon red/blue-light transitions in diatoms[42,43]). Therefore, unveiling of the BL-NobZIP77-NoDGAT2B working model, which directly links BL to metabolite synthesis, advocates *Nannochloropsis* spp. as an advantageous model system for optogenetics and should enable the rational design of opto-controlled algal and plant cell factories.

## Methods

**Strains and culture conditions.** *Escherichia coli* strain transetta (DE3) was grown in Luria–Bertani (LB) medium at 37 °C with shaking at 200 rpm. *Arabidopsis thaliana* Col-0 was grown in a glasshouse under standard conditions (temperature range 20–24 °C with a 16 h day from 06:00 h to 22:00 h and 55% relative humidity)[26]. Unless specified, *Nannochloropsis oceanica* strain IMET1 was cultivated and induced via nitrogen depletion[23,44,45]. It was cultured in modified f/2 liquid medium containing 35 g/L sea salt, 1000 mg/L $NaNO_3$, 66.6 mg/L $NaH_2PO_4 \cdot H_2O$, 3.65 mg/L $FeCl_3 \cdot 6H_2O$, 4.37 mg/L $Na_2EDTA \cdot 2H_2O$, 0.0196 mg/L $CuSO_4 \cdot 5H_2O$, 0.0126 mg/L $Na_2MoO_4 \cdot 2H_2O$, 0.044 mg/L $ZnSO_4 \cdot 7H_2O$, 0.0109 mg/L $CoCl_2 \cdot 6H_2O$, 0.036 mg/L $MnCl_2 \cdot 4H_2O$, 5 µg/L VB12, 5 µg/L biotin, and 0.1 mg/L thiamine HCl. Cells were cultivated in liquid cultures under continuous light (approximately $50 \pm 5$ µmol photons m$^{-2}$ s$^{-1}$) at 25 °C. For induction of oil production via nitrogen deficiency, the *N. oceanica* cells were harvested by centrifugation (3500 × g for 5 min) and then resuspended in the N− medium (modified f/2 medium yet without $NaNO_3$). Cell growth was determined based on both cell density and optical density ($OD_{750}$).

To characterize the *NobZIP77* and *NoDGAT2B* transgenic lines, mid-logarithmic phase algal cells ($OD_{750}$ of 2.6) were collected for validating successful transformants via PCR amplification and Sanger sequencing (Supplementary

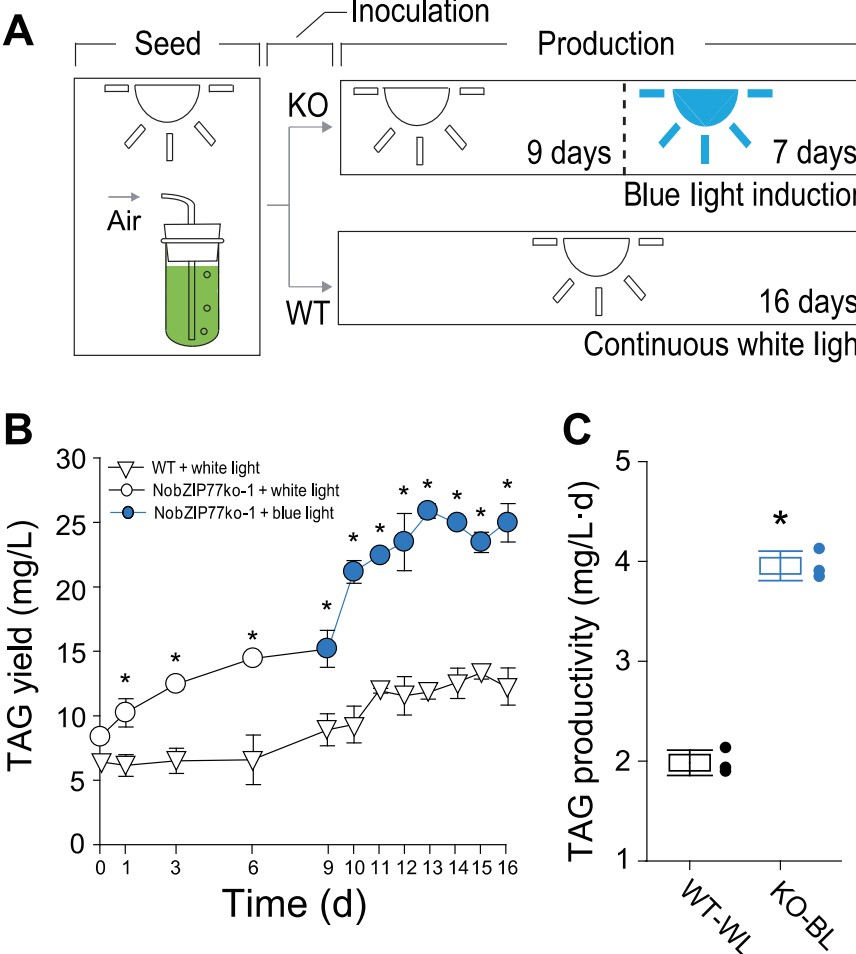

**Fig. 6 The blue-light-induced oil-production strategy.** This experiment was conducted to present the advantage of BL plus *NobZIP77*-KO-approach, as compared to continuous WL plus WT (the conventional traditional way of microalgal cultivation). Data for BL induction of WT under both N+ and N− are presented in both Fig. 5 and Supplementary Fig. 19. **A** Experimental design of BLIO: the NobZIP77ko-1 line of *N. oceanica* was cultured under WL for 9 days and then under BL for 7 days. WT grown under WL for 16 days (WT-WL) was used as the control. **B**, **C** Comparison of TAG yield (mg/L) (**B**) and average TAG productivity (mg/L·d) (**C**) between the two approaches. Data are represented as mean ± SD ($n = 3$ biologically independent samples). *: significant change ($p \leq 0.05$ by one-sided Student's *t*-test versus the control. Source data are provided as a Source data file.

Data 2). Positive lines were then cultured for further verification of transcript change. For phenotyping, the transgenic lines (plus the controls, e.g., WT transformed with empty vector or EV) were grown to OD$_{750}$ of $4.5 \pm 0.5$ for five days, and then the content and FA profiles of both TAG and total lipids were tracked at 0 h, 24 h, 48 h, 72 h, and 96 h under N−.

Three series of experiments that involved light control were performed: (1) The first series investigated the impact of NobZIP77-mediated regulation of TAG synthesis by blue light (BL) in vivo. *N. oceanica* cells (WT and knockout mutant NobZIP77ko-1) were cultured in modified f/2 medium under continuous white light till log phase. Cells were kept in darkness for 36 h, and then harvested by centrifugation ($3500 \times g$, 5 min) and resuspended in fresh modified f/2 medium (N+/N−). The resuspended cells were transferred into bio-reaction tubes and then cultured under darkness, green light (550 nm, $50 \pm 5$ μmol photons m$^{-2}$ s$^{-1}$) or BL (445 nm, $50 \pm 5$ μmol photons m$^{-2}$ s$^{-1}$), by exposing the samples to a customized LED illumination system that allows modulation of the spectral composition and intensity of lights. Each group included three replicates. Cell aliquots were taken at 24 h from each of the columns for Raman spectra measurement and TAG content analysis.

(2) The second experimental series probed the potential of exploiting the NobZIP77-mediated signaling pathway in elevating microalgal TAG productivity. *N. oceanica* cells were cultured in modified f/2 medium under continuous white light (WL) till log phase. Cells were harvested by centrifugation ($3500 \times g$, 5 min) and resuspended in the N+ or N− medium (modified f/2 medium without NaNO$_3$). Then the resuspended cells (OD$_{750} = 4$) were cultured under WL ($50 \pm 5$ μmol photons m$^{-2}$ s$^{-1}$) or BL (445 nm, $50 \pm 5$ μmol photons m$^{-2}$ s$^{-1}$), by exposing samples (in triplicates) to a customized LED illumination system. Cell aliquots were taken from each column (0 h, 24 h, 72 h, 144 h, 216 h, and 264 h for N+; 0 h, 6 h, 12 h, 24 h, 48 h, 72 h, 96 h, 120 h, 144 h, 168 h, 216 h, and 264 h for N−) for growth measurement, Raman spectra acquisition, and TAG content analysis.

(3) The third experimental series assessed the blue-light induction strategy that employs both *NobZIP77* knockout and BL for elevating oil productivity. Both NobZIP77ko-1 and the WT algal strains were cultured in modified f/2 medium under continuous WL till log phase. Then cells were harvested by centrifugation ($3500 \times g$, 5 min) and resuspended in the N+ medium. The subsequent cultivation consisted of two phases and lasted 16 days in total. At the first phase (nine days), the resuspended cells (OD$_{750} = 4$), i.e., both NobZIP77ko-1 and WT, were cultured for nine days under WL ($50 \pm 5$ μmol photons m$^{-2}$ s$^{-1}$). At the second phase (seven days), the NobZIP77ko-1 strain was moved to BL (445 nm, $50 \pm 5$ μmol photons m$^{-2}$ s$^{-1}$) and the culture went on for seven days, while the WT strain continued its culture under WL for seven days. Cell aliquots were taken from each column (0 d, 1 d, 3 d, 6 d, and 9 d for the first step; 10 d, 11 d, 12 d, 13 d, 14 d, 15 d, and 16 d for the second step) for growth measurement, Raman spectra acquisition, and TAG content analysis. All experiments are in triplicates.

For all *N. oceanica* experiments in this work, the culture was centrifuged in $3500 \times g$ for 5 min and then divided into three individual 150 mL aliquots, i.e., in three separate biological replicates. The time point of these aliquots was regarded as 0 h. These aliquots were then separately cultured under various nitrogen levels or light wavelength conditions, and then sampled at 6 h and up to 16 days after the above treatments. For each time point, 1 mL or 30 mL aliquots were sampled for growth measurement, Raman microspectroscopy, lipid profiling, or RNA extraction.

**Identification of TFs that respond to nitrogen depletion and their target genes.** To identify the N-depletion-responding TFs in *N. oceanica* IMET1, expression profiles of 35 TF-coding genes from our previously published temporal transcriptome datasets under N− were analyzed[23], based on the predicted TF-TFBS network that we constructed[11]. Gene expression between N+ and N− was

considered different if the FPKM value showed at least 1.5-fold change while the false discovery rate (FDR)-corrected *p*-value was <0.05[23]. The lipid-related target genes predicted for each of the above TFs were tabulated and classified based on annotated function[11].

**Cloning and phylogenetic analysis of genes**. *N. oceanica* genomic DNA was used as PCR templates (all primers used are listed in Supplementary Data 2). PCR products were then sequenced to obtain the full-length *NobZIP77* and *NoDGAT2B* protein-coding sequences (Genbank ID MT273120 and KX867957). In addition, the extracted genomic DNA was used as template to amplify the genomic *NobZIP77* and *NoDGAT2B* sequences that include both introns and exons. Then the gene structure (i.e., exons, introns, and untranslated regions) was verified by alignment and comparison between the protein sequence and the genome sequence. Conserved motifs in NobZIP77 were identified by MEME version 5.3.3 (http://meme-suite.org/tools/meme).

The encoded protein sequence of *NobZIP77* was aligned with MUSCLE version 3.8.31[46] and adjusted manually using BioEdit version 7.0.5.3[47] before phylogenetic analysis. The optimal substitution model of amino acid substitution was selected using ModelGenerator version 0.84[48]. The curated alignment was then used to construct a phylogenetic tree via the neighbor-joining (NJ) method in MEGA4.1[49], with the tree tested by bootstrapping with 1000 replicates.

**Bioinformatics analysis and structure modeling of the NobZIP77 protein**. For homology modeling, the sequence of NobZIP77 (Genbank No. MT273120) was used and the template was identified using PSI-BLAST against the RCSB protein data bank (PDB). The primary sequence of NobZIP77 consists of 321–448 showed 77.34% identity to LOV domain-chain A from *Phaeodactylum tricornutum* (PDB: 5DKK). And the 3D-structure was built using Swiss-Model server in template mode. Energy minimization of homologous structures was carried out with Molecular Operating Environment (Chemical Computing Group Inc., Quebec, Canada). All atoms of the modeled NobZIP77 were subject to the calculation. All energy minimization was performed under the forcefield of Amber10:EHT. If there were $H_2O$, all $H_2O$ molecules were treated as rigid bodies. Energy minimization was terminated when the root mean square gradient falls below the 0.1 kcal/mol/Å$^2$ value. The final model was evaluated by Ramachandran plot and computed with PROCHECK version 3.5.4 (http://services.mbi.ucla.edu/SAVES), which contains 83.2% of residues in most favored regions, 16.8% residues in additional allowed regions, and 0.0% residues in disallowed regions. The structural figures were generated by PYMOL version 2.5 (http://www.pymol.org/).

**The EMSA experiments**. The coding sequence of *NobZIP77* was inserted into the pMAL-p4X vector (NEB, USA) and then expressed in *Escherichia coli* strain Transetta (DE3)[50]. The recombinant protein was purified using the Amylose resin (NEB, USA), before use in EMSA. As for probes, the promoters of predicted NobZIP77 target genes and the negative controls were amplified with primers labeled with biotin at the 5′ end (Supplementary Data 3). To test the direct bindings between NobZIP77 and above-mentioned promoters, the EMSA assays were conducted via a LightShift® Chemiluminescent EMSA Kit (Thermo Fisher, USA). Briefly, the labeled DNA probes and purified NobZIP77 were incubated for 30 min in the binding buffer, and then analyzed on 6% native polyacrylamide gels. The gel electrophoresis was performed at 4 °C for 60 min (100 V). Finally, the DNA was electroblotted onto a nitrocellulose membrane and detected via chemiluminescence. In all these procedures, unlabeled probes were used as competitors, while the biotin-labeled probe alone was used as negative control.

To probe the effect of WL and BL on the NobZIP77-*NoDGAT2B* interaction, EMSA was performed either in the dark or under continuous WL or BL illumination. The labeled DNA probes (5 nM) and varying amounts of purified NobZIP77 (a 1:2 dilution series of NobZIP77, with the maximal concentration of 24 mM) were incubated in the binding buffer. The protein DNA mixtures were incubated at 25 °C for 60 min under darkness or continuous WL (400 μW cm$^{-2}$) or BL illumination (400 μW cm$^{-2}$ at 450 nm), and then analyzed on 6% native polyacrylamide gels. Gel electrophoresis was performed at 4 °C for 60 min (100 V) under darkness or continuous illumination of WL (30 μW cm$^{-2}$) or BL (30 μW cm$^{-2}$). Finally, the DNA was electroblotted onto a nitrocellulose membrane and detected via chemiluminescence. Super shifted (SS) bands of NobZIP77 bound to p*NoDGAT2B* were used for optical density quantification with Image-Pro Plus 6.0. For each gel, SS optical density of 24 mM NobZIP77 was defined as 100%.

**Construction of genetic vectors**. To construct the vectors for protein overexpression in *N. oceanica* IMET1, *NobZIP77*, and *NoDGAT2B* cDNA were amplified (Supplementary Data 2) and subcloned into pXJ450 vector (Supplementary Fig. 3A), respectively, forming pXJ459 (containing the *NobZIP77* fragment flanked by *Nde*I and *Bam*HI sites) and pXJ452 (containing the *NoD-GAT2B* fragment flanked by *Sal*I and *Bam*HI sites), respectively (Supplementary Fig. 3B, C).

To construct the vectors for RNAi-based knockdown of *NobZIP77* and *NoDGAT2B*, a 232-bp or 212-bp short fragment and a 447-bp or 437-bp long fragment were amplified from the cDNA of *N. oceanica* IMET1, respectively (Supplementary Data 2). The fragments were digested with *Eco*RI and *Xba*I and

ligated to the *Eco*RI-digested pXJ431 plasmid[21] to form pXJ448 (containing *NobZIP77* fragments) and pXJ432 (containing *NoDGAT2B* fragments), respectively (Supplementary Fig. 3D, E).

To construct the CRISPR/Cas9 vectors for *NobZIP77* and *NoDGAT2B* knockout (KO), two DNA fragments with the hammerhead ribozyme and gRNA target sequence (Supplementary Data 2) were designed and formed a primer dimer via annealing. 5′-ACGGTGGTGAACAATGGAGG-3′ (501–521 bp, for *NobZIP77*) and 5′-ATGTCCGCAGCGCAATGGCG-3′ (227–246 bp, for *NoDGAT2B*) were used as the targeted sequence with a PAM sequence of TGG. The primer dimer was ligated to the *Bsp*QI-digested pNOC-ARS-CRISPR-v2 vector (with hygB resistance)[38] to form the knockout vector. This vector expresses Cas9 and gRNA via the bidirectional *Ribi* promoter (using the *CS* and *LDSP* terminators, respectively). Moreover, the primer dimer of *NoDGAT2B* was ligated to a revised pNOC-ARS-CRISPR-v2 vector with a *ble* rather than *hyg* marker.

**Transformation of *Nannochloropsis oceanica* cells**. Nuclear transformation of *N. oceanica* was performed for linearized overexpression and RNAi vectors (or circular Cas9 vector) using the high-voltage (11,000 V/cm) electroporation method[51]. The transformant with empty pXJ450 vector was used as control (Supplementary Fig. 3A). Mid-logarithmic-phase algal cells (OD$_{750}$ = 2.6) were collected for validation of successful transformants via PCR amplification (Supplementary Data 2). For *NobZIP77*- and *NoDGAT2B*-CRISPR/Cas9, twelve PCR-positive monoclonies were identified via 3730-sequencing, respectively. Among *NobZIP77*-CRISPR/Cas9 lines, two types of knockout events were identified, including 5′-ACGGTGGT-GAACAATG-AGG-3′ (NobZIP77ko-1) and 5′-ACGGTGGTGAACAATAGG-3′ (NobZIP77ko-2). Among the *NoDGAT2B*-CRISPR/Cas9 lines, two types of successful knockout were identified, including 'ATGTCCGCAGCGCAATG-CG' (2Bko-1) and 'ATGTCCGCAGCGCAATGGGGCG' (2Bko-2). Moreover, nuclear transformation of *N. oceanica* NobZIP77ko-1 was performed for *NoDGAT2B*-CRISPR/Cas9 vector using the same electroporation method. Among *NobZIP77*-*NoDGAT2B*- CRISPR/Cas9 lines (double-knockout mutants), knockout events were identified, including 5′-ATGTCCGCAGCGCAATG-CG-3′ (NobZIP77-2B-ko1) and 5′-ATGTCCGCAGCGCAATG-CG-3′ (NobZIP77-2B-ko2) (Supplementary Fig. 14). Considering the pNOC-ARS-CRISPR-v2 vector can be automatically lost in transgenic lines[37], complemented strains for the *NobZIP77* and *NoDGAT2B*-knockout lines were produced via transforming the overexpression vectors of pXJ459 and pXJ452 (Supplementary Fig. 3B, C) into *NobZIP77*- and *NoDGAT2B*-knockout lines, respectively.

**Lipid isolation and quantification via TLC and GC-MS**. Total lipids were extracted from dried samples using chloroform:methanol (2:1 [v/v]) with 100 mM internal control of tri13:0 TAG and separated on a silica TLC plate using a mixture of solvents consisting of petroleum ether, ethyl ether, and acetic acid (70:30:1, by volume). To quantify the amount of TAG accumulated in *N. oceanica* IMET1 strains that express the *NobZIP77* or *NoDGAT2B* constructs, TAG bands were scraped from the TLC plate. Fatty acid methyl esters (FAMEs) were prepared by acid-catalyzed transmethylation of the TAG bands and then analyzed by GC-MS[52]. Mixed analytical standards of FAMEs and pentadecane were used as external and internal standard, respectively. The amounts of TAGs and the profiles of TAG-associated FA were calculated based on the results derived from GC-MS. The chemicals used as standards were purchased from Sigma, USA.

**Quantification of oil content in *Nannochloropsis oceanica* via single-cell Raman spectra**. A ramanome is defined as the collection of single-cell Raman spectra (SCRS) randomly sampled from an isogenic cellular population[53]. Based on ramanome, we have developed a method to quantify oil content of a given *N. oceanica* sample.

TLC-GC-MS based methods were generally used for quantification of TAG content, however, they typically consume 30 mL culture for each measurement. As the maximal culture volume is frequently limited, these methods are not practical for tracking cellular oil content with high temporal density or over prolonged microalgal cultivation. Moreover, they are quite tedious and slow (3–4 days for a complete process of experimental analysis). We have developed a rapid, non-destructive, single-cell-resolution method for quantifying TAG content that reduces sample volume by at least 20 folds[29,54,55]. Typically, 1.5 mL of each algal culture (the volume can be even lower) was centrifuged at 3000 × *g* and cell pellets preserved in −80 °C until use. Aliquots of the wet algal paste thawed at 4 °C were resuspended in 0.2 mL ddH$_2$O by gentle shaking, and then SCRS of individual algal cells were acquired on a Raman Microspectrometer (Single-Cell Biotech. Ltd, Qingdao, China) with excitation wavelength of 532 nm[29,54,55]. Briefly, cells were washed by ddH$_2$O three times after centrifugation and loaded into a capillary tube (50 mm length × 1 mm width × 0.1 mm height; Camlab, UK). Individual cells were trapped and measured after photobleaching by a 532 nm laser with about 50 mW output power. For the whole spectra (398.435–3344.56 cm$^{-1}$), the acquisition time for each SCRS is 1 s. The SCRS of 15–20 cells and four background spots were randomly recorded for each of three biological replicate cultures (i.e., 45–60 cells per time point). The raw SCRS were pre-processed for background subtraction, baseline correction, and normalization with Labspec 5 software (Horiba JobinYvon Ltd., UK). The information-rich region of SCRS (951.063–3050.17 cm$^{-1}$) was

extracted and normalized via division by its area. Partial least squares regression (PLSR), which develops multivariate calibration models that correlate between investigated properties and spectroscopic data, was employed to construct a model to predict the oil content for a given sample based on its ramanome.

For calculating oil content, both the ramanome dataset and its corresponding TAG content/yield dataset of the TAG-producing process of wild-type *N. oceanica* IMET1, which consists of the physiological state at each of five timepoints in triplicate cultures (0 h, 24 h, 48 h, 72 h, 96 h) under N−, were collected as a reference dataset. TAG content (μg/mg DW) and yield (mg/L) quantification models were, respectively, established based on this reference dataset. Two of the triplicate cultures were used as the training dataset and the remaining one as the test dataset for model validation. PLSR models were established using the averaged SCRS and the corresponding TAG content or yield by the training dataset and then validated by the test dataset[29,54,55]. The PLSR model for TAG content (μg/mg DW) feature coefficient ($R^2$) of 1 for the training dataset and 0.9662 for the test dataset, with the overall coefficient of 0.9801 (Supplementary Fig. 22A). Similarly, the PLSR models for TAG yield feature overall $R^2$ of 0.9801 (Supplementary Fig. 22B). These models were applied to quantify the TAG content and TAG yield for new samples.

**Chlorophyll *a* extraction and analysis**. Chlorophyll *a* in fresh algal cells (5 mL culture, centrifugation 8000 × *g* for 10 min) were extracted with 5 mL methanol for 5 h at 4 °C. By measuring the optical density, respectively, at 665 nm, 652 nm, and 750 nm with a spectrophotometer (Gene Quant 1300; GE Healthcare Ultrospec 500 Pro, USA), the pigment concentrations were calculated[56]. Absorbance at 652 nm and 665 nm was corrected by subtracting the absorbance at 750 nm. The concentration of Chlorophyll *a* was calculated with the formula:

$$[\text{Chlorophyll } a](\text{mg/L}) = 16.5169 \times A_{665} - 8.0962 \times A_{652} \quad (1)$$

**Fluorescence microscopy for protein subcellular localization**. Localization of NobZIP77:GFP and NoDGAT2B:GFP fusion proteins in *N. oceanica* IMET1 cells was carried out with a laser-scanning confocal microscope, FluoView FV1000 (Olympus, Japan). Chlorophyll autofluorescence was excited at 559 nm and detected at a bandwidth of 650–750 nm. Fluorescence of GFP was excited at 488 nm and detected at a bandwidth of 500–525 nm.

To validate the nucleus-localization of NobZIP77, the NobZIP77:GFP line and WT were stained with 0.5 μg/mL 4′,6-diamidino-2-phenylindole (DAPI, Solarbio) for 5 min at room temperature and then twice-washed with Tris-buffered saline Tween (TBST, Solarbio) for 4 min at room temperature. Fluorescence of DAPI was excited at 405 nm and detected at 450–464 nm.

To confirm the subcellular localization of NoDGATA2B, the NoDGAT2B:GFP line and *N. oceanica* WT were stained with 0.25 mg/mL Nile red (Solarbio) for 30 min at room temperature and then twice-washed with distilled water. Fluorescence of Nile red was excited at 488 nm and detected at 560–590 nm.

**Validation of the NobZIP77-mediated *NoDGAT2B* inhibition ex vivo in *Arabidopsis thaliana***. The transcription activity assay was carried out in the transient-transformed protoplast of *A. thaliana*[26]. It consists of the following steps: preparation of reporter and effector constructs and Arabidopsis mesophyll protoplasts, performing the transient transfection, lysing the transfected protoplasts, and analyzing the expression level of *GUS* in the soluble extracts after 18 h of incubation. Here, both effector and reporter were constructed using the pBI221 (Clontech, USA) vector. For detecting the inhibitory activity of *NobZIP77* to the tested promoter, the promoter sequence of *NoDGAT2B* was amplified from *N. oceanica* genomic DNA (Supplementary Data 2) and ligated to the upstream of the *GUS* reporter after removing the 35S promoter in pBI221, so as to create the reporter constructs. The *NobZIP77* coding region was amplified from the *N. oceanica* cDNA library (Supplementary Data 2) and ligated between the *35S* promoter and the *NOS* terminator after removing *GUS* from the pBI221 vector, so as to create the effector construct. The reporter and effector constructs as well as the 35S:LUC vector (internal control) were co-transfected into Arabidopsis leaf protoplasts. All the samples were incubated overnight in darkness, and LUC luminescence and 4-methylumbelliferone fluorescence (products of GUS activity) measured by a luminometer (Promega). The expression level of reporter gene was determined as the relative ratio of GUS to luciferase activity. The experiments were conducted at least three times. In each experiment, expression level of the *GUS* reporter in the protoplasts transfected with the reporter construct alone was measured, to normalize the results.

**Validation of NobZIP77-mediated *NoDGAT2B* inhibition in vivo in *Nannochloropsis oceanica***. To validate the NobZIP77-mediated *NoDGAT2B* inhibition in *N. oceanica*, a green fluorescent protein (GFP) reporter system was designed and constructed. First, the *NoDGAT2B*-promoter sequence was isolated as above-mentioned (Supplementary Data 1). Second, p*NoDGAT2B* was inserted into a backbone vector of pXJ53 (containing a core cassette of GFP-α-tubulin terminator-β-tubulin promoter-*ble*-violaxanthin/chlorophyll *a* binding protein (*VCP*) terminator) to produce the control vector pXJ545 (Supplementary Fig. 3F). Third, the cassette of *β-tubulin* promoter-NobZIP77-*psbA* terminator was isolated from

pXJ459 (Supplementary Fig. 3B) and subcloned into the interspace between *α-tubulin* terminator and *β-tubulin* promoter in pXJ545, to produce the reporter vector pXJ546 (Supplementary Fig. 3G). Then pXJ545 and pXJ546 were transformed into *N. oceanica*, respectively. For the PCR-positive lines, the level of *GFP* transcript was measured by qRT-PCR, and the fluorescence of GFP detected via a laser-scanning confocal microscope. GFP intensity was then quantified by Olympus Flowview Ver 4.0b. For each of the lines, GFP intensity of ten cells was quantified and the average intensity per cell was used for comparison across lines.

**ChIP-qPCR analysis**. To validate the direct binding between NobZIP77 and p*NoDGAT2B*, a ChIP-qPCR experiment was performed[57,58]. Briefly, 400 mL of algal cells from WT and transgenic *N. oceanica* overexpressing GFP-NobZIP77 (OE1; Supplementary Fig. 10) were harvested and cross-linked with 1% formaldehyde. Then algal cells were ground into fine powder with liquid nitrogen, and thereafter, chromatin was extracted and sheared into 200- to 2000-bp fragments by sonication. Subsequently, the crude chromatin lysates served as the input and were incubated overnight at 4 °C with or without anti-GFP monoclonal antibody (Abmart M20004, clone7G9, 1/50; Shanghai, China). The immunoprecipitated complexes were then pre-cleared with protein-A agarose beads (Cell Signal Technology, Danvers, MA, USA). Then the precipitated DNA was extracted with phenol and chloroform, and then purified with ethanol.

For ChIP-qPCR experiments, immunoprecipitated DNA and control DNA, with three biological replicates for each, were mixed with primers and LightCycler 480 SYBR Green I master mix (Roche), respectively. Products were amplified and fluorescent signals acquired with a LightCycler 480 detection system. The fold enrichment of target in the immunoprecipitated DNA relative to the control was calculated from an average of three replicate qPCRs. Relative enrichment of the target region of *NoDGAT2B* was normalized against *β-actin*. The fold change was calculated via the $2^{-\Delta\Delta Ct}$ method between mutant and WT, as below. Sequences of the primers are listed in Supplementary Data 2.

$$\Delta\Delta Ct = (\Delta Ct\,(Ct_{[\text{ChIPDNA}]} - Ct_{[\text{InputDNA}]});\ \text{mutant}) - (\Delta Ct\,(Ct_{[\text{ChIPDNA}]} - Ct_{[\text{InputDNA}]});\ \text{WT})$$
$$(2)$$

**Statistical analysis**. All experiments were in triplicates, with results presented as mean ± standard deviation (SD). Statistical analysis was performed using Graphpad Prism 5 (GraphPad, USA). The *p*-values were calculated via one-way analysis of variance (ANOVA).

**Reporting summary**. Further information on research design is available in the Nature Research Reporting Summary linked to this article.

## Data availability
The coding sequences of NobZIP77 and NoDGAT2B are deposited in GenBank under MT273120 and KX867957, respectively. Source data are provided with this paper.

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

## Acknowledgements

This work was supported by grants from National Key Research and Development Program (2018YFA0902500), Natural Science Foundation of China (31600059, 31425002, 31900074, 31900047, and 31800071), Postdoctoral Science Foundation of China (2018M642716), Natural Science Foundation of Shandong, China (ZR2019BC016 and ZR201709180185), and Strategic Priority Research Program of the Chinese Academy of Sciences (XDPB18). We acknowledge Eva M Farré for helpful discussions, and thank Yanbin Feng for structural modeling of NobZIP77.

## Author contributions

J.X., Y.X., and P.Z. designed research; P.Z., Y.X., and X.T. performed the EMSA assay; P.Z., and Y.X. performed the GFP-reporting assays in N. oceanica; P.Z., Y.X., Q.W., and N.L., generated and screened transgenic lines of N. oceanica; P.Z., Y.X., and Y.L. conducted the GC-MS assay; Y.H., and P.Z. performed Raman sampling, measurement, and analysis; C.S., P.Z., and Y.X. performed the protein localization assay. Y.X. and P.Z. performed phylogenetic analysis; Q.H. provided inputs. J.X., Y.X., and P.Z. analyzed data and wrote the paper.

## Competing interests

The authors declare no competing interests.
