## [Peer Review File · Nature Communications]

Exploring a blue-light-sensing oleaginous pathway to double the peak productivity of oil in *Nannochloropsis oceanica*Reviewers' Comments:

Reviewer #1:

Remarks to the Author:

The paper by Zhang et al., A blue-light-inductive oleaginous pathway doubles peak oil productivity in industrial microalgae, is well written and for the most part the data supports the main conclusion of the paper, that a transcription factor likely involved in lipid metabolism has been identified. This is certainly not the first TF identified in algae lipid biosynthesis, so it is unclear why the authors make that claim so many times in the paper. There are several other major concerns that need to be addressed before this paper is suitable for publication.

- 1) The authors consistently state that this is the first signaling pathway identified and the first transcription factor identified that functions in a role to increase lipid accumulation in algae. This is simply not correct, there are many examples of previously identified lipid signaling pathways, and signal transduction cascades, and even cloned and over and under-expressed transcription factors - shown to function in lipid accumulation in algae. This may be the first lipid pathway that has a blue light response, but that does not in my mind trump others previous work, which gets almost no discussion in this paper.
- 2) Chief among the previous publications that need to be discussed is the paper by Ajjawi et al from 2017; Lipid production in *Nannochloropsis gaditana* is doubled by decreasing expression of a single transcriptional regulator. That paper has significant overlap with the work here, and in fact many of the conclusions in the present paper, were previously reported in the Ajjawi paper, so why is there so little discussion, especially between the differences between the transcription factor identified in this work and the one described in the Ajjawi paper?
- 3) The paper by Thiriet-Rupert et al., 2018, Identification of transcription factors involved in the phenotype of a domesticated oleaginous microalgae strain of *Tisochrysis lutea*, should also have been included in any discussion.
- 4) The authors also state in the discussion: "Finally, such a concise pathway that directly links light sensing to metabolite synthesis was not previously considered possible." I am not sure what the authors are talking about here, almost ALL metabolic pathways in plants and algae are somehow tied to light sensing. Same with this statement: "however, despite the many studies profiling transcriptomics, proteomics and metabolomics of N- induced TAG synthesis in various microalgae, signaling pathways linking the stress and TAG synthesis remain elusive." Perhaps they mean that specific molecular interactions have not been documented, but certainly the pathways are well known and documented.
- 5) The statement "Here in the industrial oleaginous microalgae *Nannochloropsis oceanica*, we discovered the first signaling pathway underlying N- induced oleaginousness." The paper by Ajjawi et al identified a transcription factor associated with nitrogen induced lipid metabolism pathway previously, so not sure what this statement is claiming, perhaps a light induced pathway?
- 6) Statement: "Consistently, growth inhibition has frequently been a side effect of metabolic engineering that boosts cellular TAG content (e.g., resulting in ~25% biomass decrease." Yes N starvation slows growth rate, but many have published on algae with increased growth rates and no loss of biomass accumulation.
- 7) The statement "Therefore, NobZIP77 expression level exerts no significant effects on microalgal growth rate." Seems highly unlikely and has not been demonstrated here. That a small increase or decrease in NobZIP77 expression does not dramatically reduce biomass accumulation under a short laboratory experiment does not allow one to reach the former conclusion.

8) To demonstrate NobZIP77 function the authors use a CaMV35S promoter:NobZIP77 in Arabidopsis mesophyll protoplast, and associate the relative GUS activity as proof that NobZIP77 inhibits NoDGAT2B transcription. I do not understand why this experiment was run, the authors know how to make *N. ocellata* transgenic lines, so why use a plant system with a CaMV promoter to assay for this, the plant system and associated assay is simply too far away from algae and what is really needed to demonstrate NobZIP77 function.

9) A recombinant maltose-binding-NobZIP77 fusion protein was expressed in *Escherichia coli* and purified for gel shift assays to identify DNA binding activity of NobZIP77. Although DNA binding assays are notoriously unreliable, the figure presented here (Fig. 2C) is just not convincing. A standard protein dilution generally results in increasing amounts of DNA shifted, and the DNA/protein is often a monomer at low protein concentrations and then a protein dimer at higher concentrations. Unlabeled DNA is then used to outcompete the labelled fragment. None of that is observed in figure 2C, making it difficult to tell if the protein is really specific for that DNA fragment or not. All DNA binding proteins have some non-specific affinity for DNA.

Reviewer #2:

Remarks to the Author:

This manuscript described a transcription factor, NobZIP77 (g77), as a blue light sensor and a regulator of NoDGAT2B, leading to improved TAG production. There were merits in the manuscript, including description of the new TF and its possible functions for lipid metabolism. However, its direct interaction with the promoter of NoDGAT2B is overall weak: the *in vitro* EMSA assay is incomplete, and no *in vivo* evidence (such as ChIP assays). There were conflicts between its expression and that of the target, leaving an open question for different targets (such as lipases listed in Table S1). Moreover, its localization as a TF in the nucleus is not complete. There were many other points to be addressed for acceptance in *Nature Communications*.

- Table S1: The data presented in this table look interesting, but lack practical information, such as FPKM for both TFs and targets (LRGs), to show how much TF expression was induced and how target expression is affected.

In addition, targets may have different (and opposite) impacts on lipid metabolism. For example, targets of g77 include DGATs and lipases, which warrants explanation.

- Figure 1B: If g77 is a blue light sensor, the structure should be drawn with and without its chromophore.

- Figures 2A and 2D: Expression of NoDGAT2B and TAG analyses were done in different time scale. Why?

- Figure 2B: The reporter assay could have been better if performed in *Nannochloropsis*.

- Figure 2C: In my understanding, the super shift should be caused by adding an antibody against g77. I think that the different sizes of shifted bands may represent formation of oligomers or dramatic conformational changes of g77.

However, effects of unlabeled DNA are weak (even with 10 times more of it, as shown lanes 2,3,4), which may suggest non-specific binding of the promoter sequence of NoDGAT2B. This should be considered critically, because this data is the only evidence for direct interaction between g77 and NoDGAT2B promoter. It is also recommended to perform the same EMSA with non-specific cold DNA (e.g., same amount of unrelated genomic DNA fragments) and compare to that of specific cold probe. In addition, bZIPs are usually transcriptional activator, not repressors (Figure 2A). With these possible conflicts, it is recommended to run EMSA with different promoters, such as one of the lipase genes (Table S1), which may have opposite impact on lipid metabolism, compared to DGATs.

- Figure 2D: In addition to the % TAG (as shown here), it would be interesting to see TAG contents, since many DGATs are known to increase TAG contents and yields.

It is expected to have opposite effects between NoDGAT2B ox and kd on TAG contents, which was shown only on time 0. Effects of ox disappeared at later times, which requires explanation. One suggestion may be that NoDGAT2B may not be related to N-, but with N+ conditions.

Figure 3A-C: Quantitation of DNA binding should be measured with the shifted band, rather than the saturated signals of the unbound. Actually, shifted bands appeared faster under the dark. Another question is about the same EMSA assay with the white light (rather than dark), since all other experiments were done with white lights as a control.

Figure 3C: Were EMSA assay repeated to reveal statistic significance? Binding with dark and blue light doesn't look significantly different. It would also be better to see blue light effects with N+ conditions. Figures 3E,F: It is good to see employing Raman technology for measuring relative TAG contents. Can a standard curve be created using cells with known TAG contents?

It is also interesting to see effects of the green light: It is generally thought that green light is not used for photosynthesis (and I don't recall any green light receptors), and can be assumed no effects on lipid metabolism; however, it has effects under N- condition. How could it be?

Another point to note: Blue light still has positive effects in g77 KO line under N- condition, which leaves a question for additional blue light sensors.

Figure 4A: Nuclear localization for g77 should be confirmed by nuclear staining using DAPI. Localization of NoDGAT2B is not clear. It should be co-localized with sDer1:RFP fusion. If it is associated with lipid droplets, it can also be co-stained with Bodipy or Nile red.

Figure 5: Experiments were done with different time schedules, and it is hard to directly compare data directly. Time values are misplaced in Figure 5A.

Figure 6: Explain why blue light induction was done for KO mutants only. To claim its effect, the same treatment should be applied to WT.

* The letters (B and C) are misplaced.

- Figure S1A: Actually, expression patterns for g77 transcripts and protein are not consistent. Protein level increased at six hours, and then dramatically decreased at 12 hrs; however, transcripts levels were relatively consistent. Explain why. More importantly, expression patterns for NoDGAT2B and other targets (also pointed out in Table S1). True interacting targets should be selected from this.

- Figure S2: There are multiple bZIPs in algae and many more in plants. Explain how the representative bZIPs were chosen. Are they all related to blue light sensing?

- Figure S4: Expression levels of g77 were analyzed up to 48 hrs, but TAG was analyzed up to 96 hrs. Explain why.

- Figure S4C showed only TAG per total lipids. It is recommended to show a graph for total lipid contents.

In addition, expression patterns of g77 is more or less consistent as shown in Figure S4A, but its impact on TAG content (S4C) and TAG yields (S4D) are different (mostly at 0 and 24 hrs). Explain why.

*The gene name was initially defined as g77, but NobZIP77 was also used later. It would be better to use consistent names.

- Figure S5: Please indicate gene names together with their id. Induction patterns for different targets are different, and some are opposite. Explain why.

- Figure S7: LOV domains can be found in different types of proteins, and it would be interesting to reveal identities of proteins that contains the domain.

- Overall English writing needs improvement by professional writers, since many statements were not clear and can be misleading; for example, in Abstract: "a blue-light-oil-induction strategy that exposes NobZIP77-knockout *N. oceanica* to white light and then to blue light doubles peak TAG productivity" What does it mean?

It is not clear (and actually confusing) how the blue and white lights are used and how they are related, leaving a room for confusion and misunderstanding.

- In my version of the manuscript, references 1-38 are missing.

- Line 83: Totally?

Reviewer #3:

Remarks to the Author:

It is well known that microalgae produce a high amount of TAGs upon nitrogen depletion, *Nannochloropsis* is especially famous for this and proposed to be one major algae with interesting industrial potential. However, the signaling mechanisms triggering the metabolic switch from a carbohydrate/protein metabolism to a lipid metabolism are not well understood. In this manuscript, the authors, based on bioinformatics analyses, identified an interesting transcription factor (NobZIP77) which might be involved in regulating lipid content after nitrogen depletion. They produce knockdown, overexpressing and KO mutants of this factor and characterize them with respect to growth and lipid production, also including testing the effect of different light colors. They identify this transcription factor to be an aurochrome, a blue light photoreceptor with transcription factor activity, a peculiarity so far only found in stramenopiles. Moreover, the authors propose a putative target gene of this TF, a diacylglycerol acyltransferase (NoDGAT2B) for which they additionally produce knockdown and overexpressing lines and characterize them with respect to growth and lipid production. Moreover, they conclusively demonstrate an interaction of NobZIP77 with NoDGAT2B promoter by EMSA analyses and GUS reporter assays. They show that NobZIP77 represses NoDGAT2B expression, and this repression is released upon blue light exposure. Interestingly, they show that using this knowledge and the KO mutant, higher lipid yields can be obtained without compromising cell growth, which is normally, using the N-limitation strategy, a major problem for upscaling microalgae oil production to an economically interesting scale.

The results are novel and widely interesting for the field. The experimental strategy is well-thought, the data presentation is clear, the manuscript well written and the interpretation of the data is mostly justified, including an interesting working model.

Besides several major and minor issues specified below, there is one dominant problem which definitely needs to be addressed exhaustively:

Actually there is only a significant differential regulation of NobZIP77 under nitrogen deplete conditions in the range of 12-48 hours, which made the authors initially chose this factor for further analysis. However, comparing the ov. and sil. mutants to the WT (or EV) under these conditions, there is little or no effect on the TAG accumulation (more specified below in the specific comments), the largest effect seems actually always to be at nitrogen replete condition (time point 0), which makes all interpretation regarding the effects of nitrogen limitation on TAG accumulation via NobZIP77 difficult to understand. A major reason for this may be that both, silencing and overexpression of NobZIP77, was very little effective, with reduction of transcript level (not talking about the protein level which has not been shown) only to be in the range of 20 % and overexpression only approximately 2 fold. While the effects of overexpression, especially of TFs, in general can be very ambiguous, a knockdown

and especially a KO clearly provides more reliable results. As the authors have produced KO lines of NobZIP77, I strongly recommend to repeat experiments of Fig.1c, 2a and especially S4 with the KO line. This may clarify to which extent NobZIP77 plays a direct role in regulating TAG and FA synthesis under nitrogen depletion. Based on Suppl. Fig. 8, the effect might be marginal or even not present, in line with Fig. S4, and possibly NobZIP only plays a role under N+, which then would need to be discussed accordingly.

More specific comments below.

Bernard Lepetit

Major comments:

1. Line 100: "Under N+, NobZIP77 overexpression results in 9.03% and 1.23% lower growth rates than EV, while NobZIP77 knockdown lines remained unchanged (Fig. S4B).

How do you calculate a 1.23 % lower growth rate based on this figure, while there is no change in overexpression lines? What is the math behind? And is this at all significant? (By the way: two digits after the comma here is a bit over precise). I am also wondering why the growth is exactly linear during your 8 days incubation period. This means neither is there any exponential increase in growth nor a clear stationary phase. Everyday there are around 3 million cells more than the day before, a somehow weird situation in algae growth experiments.

2. Regarding the amount of TAGs (Fig. S4) I am a bit skeptical (this comment relates to the main comment above): First, there are huge differences between silencing line 2 compared to silencing line 1, despite basically the same reduction in RNA along all time points. And actually only silencing line 2 shows significantly elevated TAG contents compared to the Wt at time point 24, 48 and 72 hours. Second, there is no real difference in TAG content of overexpressing lines compared to the silencing lines at time point 48 and 96 and also at time point 72 Wt, both overexpressing lines and the first silencing line show very similar TAG profiles.

3. I strongly miss any experimental data demonstrating that the generated obZIP77-CRISPR/Cas9 lines are indeed KO lines. It is described in the paper but has to be shown showing e.g. the PCR result including sequencing of the mutated fragment and/or Southern blot and/or (if antibody is available) Western blot. Also, as the observed differential effects on TAG accumulation in the KO line compared to WT are rather moderate, not to say small, also complemented strains should be created to exclude any off-target effect.

4. L199: "In NobZIP77ko-1, the TAG content is reduced by 11.78% under BL plus N-, revealing a temporal lag in TAG content in response to BL under N- as compared to the wild-type (Fig. 3E; Fig. 3F), although their light-quality-responsive pattern appears similar, i.e., stronger stimulation of TAG by BL than by GL (for NobZIP77ko-1, under N-, the TAG content exhibits 19.69% increase under GL and 49.83% increase under BL versus the dark; Fig. 3F). Therefore, N-induced TAG synthesis is mediated by NobZIP77 via its sensing of BL."

I only partly agree: Clearly, TAG synthesis is stimulated by light, as N.oc. needs carbon and reducing equivalents for this. The reason for the much stronger effect of BL than green light is that the authors in both cases applied the same amount of light intensity, i.e. 50 $\mu\text{mol photons m}^{-2} \text{s}^{-1}$. As blue light is much better absorbed than green light by the cell, simply more carbon and reducing equivalents are available for TAG synthesis under BL than under GL. The fact that the KO of NobZIP77 only slightly diminishes the amount of TAG accumulation under BL indicates that it is only little involved in this response, but for sure it is not a prime regulator, otherwise there should be no increase in TAG in BL at all. Hence, please formulate much more carefully.

In future experiments the authors should pay attention to the fact that when comparing the effect of

different light qualities, the differential absorption of the light colors by the cells has to be corrected, e.g. by calculating QPhar (e.g. check Jungandreas et al. 2014, PLoS One). Also, while there is extensive data on gene regulation of NobZIP77 and NoDGAT2B under white light and N-, nothing is provided on BL/GL. Please provide these data for the experimental conditions where you analyzed TAG.

5. L320: "Notably, BL greatly promotes microalgal TAG production: a NobZIP77-knockout *N. oceanica* exposed to WL first and then to BL more than doubles the peak TAG productivity"

Dangerous statement. Again, also here BL had not been adjusted properly to yield the same QPhar as white light. Hence, comparison without a reliable control is difficult. For the BLIO strategy experiment in Fig. 6, I strongly recommend including also the Wt under 9 days WL and 7 days BL. Then the real effect of BL via NobZIP77 will become visible.

6. Suppl. Fig. 8a: Why is there such a little difference in growth between N+ and N- conditions?

Starting from an OD of 4 under N replete conditions the cells reach an OD750 of around 18 after 11 days, while they reach something like 15 under N- conditions – really not much regarding the massive importance of nitrogen for cell growth. Also: Measuring an OD at 750 could include cell debris as well as bacteria, too, easily introduced and propagating during such a long experiment, which may bias the interpretation of the growth curve. By the way: Measured ODs about 1 generally have to be treated with care due to the Lambert-Beer law, but ODs larger than 10 are really not trustful anymore. I strongly recommend showing these growth experiments by really calculating the cell number, either using a growth chamber or a Coulter counter.

Minor comments:

7. Nature Comm limits references to 70 max, but this limit is not at all reached and there are several passages, where more references would be beneficial and actually needed. E.g., line 46: "In oleaginous microalgae, oil accumulation is typically induced by environmental stresses, such as nutrient deprivation, high light or heat 3, 5", more than the relatively old two references should be provided. Similarly, L50: "For example, nitrogen depletion (N-) has been intensively studied and practically used for triggering microalgal TAG accumulation 6", many more works have dealt with this issue in various microalgae besides *Neochloris oleabundans*, the species the only reference provided deals with.

8. Legend Fig 1.: Is it correct to explain FPKM as "absolute abundance of transcript"? I'd rather call it the normalized abundance of transcript (on gene length and total reads).

9. Fig. S1: How has the protein abundance been quantified? I could not find any methodological explanation here.

10. L83: "Totally 32 TFs respond to N-, with eleven upregulated and 14 downregulated."

For me, this sums up to 25?

11. Fig.1a: What about g8021: From the transcription profile it showed by far the strongest response and is actually the only factor clearly up-regulated under N-. Any ideas about its concrete function?

12. Suppl. Fig. 5, Which of these genes is NoDGAT2B? Not possible to identify based on the given gene names.

13. L125: "Among these LRGs, NoDGAT2B (Genbank ID KX867957), a novel member of the N. oceanica type-2 DGAT family 8, 9, exhibited 127.5% higher transcript level in NobZIP77i-2 while 26.0% lower transcript level in NobZIP77o-2 (compared to EV; Fig. 2A), suggesting its repression by NobZIP77."

On which of the depicted time points do these % numbers refer?

14. Legend Fig. 2C. Please explain somewhere that "2B" stands for "NoDGAT2B" promoter

15. Paragraph NobZIP77 as a blue-light sensor that releases the NoDGAT2B promoter under blue light

I absolutely agree that the sequence and domain features support NobZIP77 to be an aureochrome. I just want to add that also the inverse order of sensor and effector domain, with the sensor domain (LOV) at the C-terminus and the effector domain (bZip) rather at the N-terminus strongly support this TF to be an aureochrome. Such an arrangement is strongly atypical for any photoreceptor except the aureochrome.

16. Fig. S7C. I think there is an error in the figure legend:

"Red and blue frames indicate bZIP and LOV domains, respectively. Heptad leucine residues of bZIP domains are in blue, while the eleven conserved amino acid residues that are associated with flavin binding are in red."

It has to be "BLUE and RED frames indicate bZIP and LOV domains, respectively...."

17. Fig.4a: Although based on the fluorescence figure I also consider it likely that NobZIP77 is located in the nucleus, a proof is lacking here. Please also stain your GFP cell lines with DAPI or Hoechst and then check in the overlay the match of NobZIP77-GFP with Hoechst/DAPI fluorescence.

18. Regarding the very small error bars in Fig. 5 and Fig. S8 for the lipid measurements, it is stated that the values are means of three replicates: Are these lipid measurements from the same culture, where three aliquots have been analyzed in GC-MS measurements, or had there been cultured three independent cultures for a period of 11 days from where the lipid analyses had been performed respectively? For a measurement of lipid content, no visible error bars are somehow not to be expected.

19. L331: "On the other hand, aureochromes, although known to regulate photomorphogenesis, sexual cycle and biological rhythm, have not been implicated in the synthesis of carbon-storage compounds."

Agree, nothing is published yet, but please see Jungandreas et al. (2014, PLoS One) and discussion there about the strong metabolic shift upon red/blue light transitions in diatoms and possible signaling triggers.

20. Methods: A detailed section is dedicated to growth of Wt and NobZIP77-KO cultivation experiments. However, a very large part of the manuscript describes data obtained with silencing and overexpressing lines (NobZIP77 and NoDGAT2B). Please provide cultivation conditions for these strains, too.

Dear referees,

Thank you for giving us the opportunity to submit a revision for our original submission
(Manuscript ID NCOMMS-20-28862-T). We are grateful to the very valuable comments. All the
experiments and analyses suggested have now been completed. Below please find our itemized
responses (regular bold fonts) to these suggestions (*italic*).

Notably, all source data are provided in a “Source Data-NCOMMS-20-28862-T” folder (submitted
with this revision package) that accompanies this paper.

**Comments from the Reviewer #1**

*1. The paper by Zhang et al., A blue-light-inductive oleaginous pathway doubles peak oil*
*productivity in industrial microalgae, is well written and for the most part the data supports the*
*main conclusion of the paper, that a transcription factor likely involved in lipid metabolism has*
*been identified. This is certainly not the first TF identified in algae lipid biosynthesis, so it is*
*unclear why the authors make that claim so many times in the paper. There are several other*
*major concerns that need to be addressed before this paper is suitable for publication.*

**Response: We appreciate the constructive comments. We have revised the manuscript as**
**advised. For the question of “the first TF”, please see our response to Comment #2 below.**

*2. 1) The authors consistently state that this is the first signaling pathway identified and the first*
*transcription factor identified that functions in a role to increases lipid accumulation in algae.*
*This is simply not correct, there are many examples of previously identified lipid signaling*
*pathways, and signal transduction cascades, and even cloned and over and under-expressed*
*transcription factors - shown to function in lipid accumulation in algae. This may be the first lipid*
*pathway that has a blue light response, but that does not in my mind trump others previous work,*
*which gets almost no discussion in this paper.*

**Response: In this study, we define “a signaling pathway” to be one with its stress, mediator,**
**signal receptor and signal effector all completely clarified. The table below lists the**
**differences between this work and existing studies in microalgae (“-”: still unknown).**

Microalgae	Stress	Mediator	Signal receptor	Signal effector	Reference
N. gaditana	nitrogen starvation	-	ZnCys	-	1
Haematococcus pluvialis	high light	melatonin	MAPK	-	2
Trebouxia sp.	high salinity	ABA	-	AAO3, RD21A, etc.	3
N. oceanica	nitrogen starvation	blue light	NobZIP77	NoDGAT2B	This work

To clarify this issue, we have revised the manuscript to be more specific about our claim of
 novelty. In Abstract, “In *Nannochloropsis oceanica*, we discovered the first such pathway: ...”
 was now changed to “In *Nannochloropsis oceanica*, we discovered one such pathway: ...”
 (Line 24-25).

 In addition, in Introduction, “we discovered the first signaling pathway underlying N-
 induced oleaginousness” was now revised as: “Here in the industrial oleaginous microalga
 *Nannochloropsis oceanica*, we discovered a blue light (BL) mediated signaling pathway
 underlying N- induced oleaginousness” in Introduction (Line 62-63).

 3. 2) Chief among the previous publication that need to be discussed is the paper by Ajjawi et al
 from 2017; Lipid production in *Nannochloropsis gaditana* is doubled by decreasing expression of
 a single transcriptional regulator. That paper has significant overlap with the work here, and in
 fact many of the conclusion in the present paper, were previously reported in the Ajjawi paper, so
 why is there so little discussion, especially between the differences between the transcription
 factor identified un this work and the one described in the Ajjawi paper?

 **Response:** The table below lists the differences between the transcription factor identified in
 this work and the one discovered by Ajjawi et al.

	This work	Ajjawi’s work
Key TF identified	NobZIP77	ZnCys
Family of the TF	aurochrome	Zn _(II) Cys ₆
Engineering method	CRISPR-Cas9 (knockout)	RNAi (knockdown)
Lipid productivity improvement	3-folds (TAG)	2-folds (total lipid)
Growth retardation	No retardation	5-15% decrease
Target of the TF	NoDGAT2B	unknown
Mechanism of response to N depletion	blue light	unknown

 As advised, we have now inserted the following discussion into Discussion (Line 323-326):

 “For example, in *N. gaditana*, a TF of the Zn_(II)Cys₆ family was found implicated in N-
 induced lipid production, and its knockdown by RNAi resulted in double lipid production
 yet decreased microalgal growth rate¹; however, neither its downstream target nor the
 signaling mechanism for its response to N depletion is clear.”

4. 3) *The paper by Thiriet-Rupert et al., 2018, Identification of transcription factors involved in*
*the phenotype of a domesticated oleaginous microalgae strain of Tisochrysis lutea, should also*
*have been included in any discussion.*

**Response:** As advised, we have now inserted citation and discussion of this nice paper into
**Introduction (Line 45-50):**

**“For example, nitrogen depletion (N-) has been intensively studied and practically used for**
**triggering microalgal TAG accumulation^{4,5}; however, despite the many studies in various**
**microalgae profiling transcriptomics, proteomics and metabolomics of N- induced TAG**
**synthesis⁶, as well as the plethora of transcriptional factors (TFs) predicted to be involved**
**(e.g., *Tisochrysis lutea*⁷ and *N. oceanica*⁸), signaling pathways that link TAG synthesis to**
**stress exposure remain elusive⁹⁻¹¹.”**

5. 4) *The authors also state in the discussion: “Finally, such a concise pathway that directly*
*links light sensing to metabolite synthesis was not previously considered possible.” I am not*
*sure what the authors are talking about here, almost ALL metabolic pathways in plants and algae*
*are somehow tied to light sensing.*

**Response:** As advised, we have revised the statement as **“Finally, such a concise pathway that**
**directly links light sensing to TAG synthesis was not previously known.” (Line 342-343).**

6. 5) *Same with this statement: “however, despite the many studies profiling transcriptomics,*
*proteomics and metabolomics of N- induced TAG synthesis in various microalgae, signaling*
*pathways linking the stress and TAG synthesis remain elusive.” Perhaps they mean that specific*
*molecular interactions have not been documented, but certainly the pathways are well known and*
*documented.*

**Response:** Here we tend to emphasize the clear signaling pathway (i.e., completely clarify the
**signaling stress, media, receptor and effector) remain elusive on N- induced TAG synthesis.**

**To avoid any confusion, we have revised the statement as “however, despite the many studies**
**in various microalgae profiling transcriptomics, proteomics and metabolomics of N- induced**
**TAG synthesis⁹, as well as the plethora of transcriptional factors (TFs) predicted to be**
**involved (e.g., *Tisochrysis lutea*⁷ and *N. oceanica*⁸), signaling pathways that link TAG**
**synthesis to stress exposure remain elusive⁹⁻¹¹.” (Line 47-50).**

7. 6) *The statement “Here in the industrial oleaginous microalga Nannochloropsis oceanica, we*
*discovered the first signaling pathway underlying N- induced oleaginousness.” The paper by*
*Ajjawi et al identified a transcription factor associated with nitrogen induced lipid metabolism*
*pathway previously, so not sure what this statement is clanking, perhaps a light induced pathway?*

**Response:** Please refer to our response to Item #2 above.

8. 7) Statement: “Consistently, growth inhibition has frequently been a side effect of metabolic
engineering that boosts cellular TAG content (e.g., resulting in ~25% biomass decrease.” Yes N
starvation slows growth rate, but many have published on algae with increased growth rates and
no loss of biomass accumulation.

**Response:** As advised, we have inserted a review on this topic ¹². Moreover, we have now
revised the statement as: “Consistently, growth inhibition has frequently (although not
always) been a side effect of metabolic engineering that boosts cellular TAG content ¹² (e.g.,
resulting in ~25% biomass decrease ^{1,11}.” (Line 55-57).

9. 8) The statement “Therefore, NobZIP77 expression level exerts no significant effects on
microalgal growth rate.” Seems highly unlikely and has not been demonstrated here. That a small
increase or decrease in NobZIP77 expression does not dramatically reduce biomass accumulation
under a short laboratory experiment does not allow one to reach the former conclusion.

**Response:** Agree. Now we revised this sentence as “Therefore, under such experimental
conditions, NobZIP77 expression level does not seem to exert significant effects on microalgal
growth.” (Line 99-101).

10. 9) To demonstrate NobZIP77 function the authors use a CaMV35S promoter:NobZIP77 in
Arabidopsis mesophyll protoplast, and associate the relative GUS activity as proof that NobZIP77
inhibits NoDGAT2B transcription. I do not understand why this experiment was run, the authors
know how to make *N. oceanica* transgenic lines, so why use a plant system with a CaMV promoter
to assay for this, the plant system and associated assay is simply too far away from algae and
what is really needed to demonstrate NobZIP77 function.

**Response:** As advised, we have conducted this report-gene experiment in *N. oceanica*. Due to
the difficulty in detecting GUS activity in *N. oceanica*, green fluorescent protein (GFP) was
used for the validation of NobZIP77-mediated NoDGAT2B transcription inhibition. To avoid
potential interference of this inhibition by the endogenous NobZIP77, this experiment was
run in a NobZIP77-knockout line of NobZIP77ko1 (details in Methods, Line 648-662).

**Firstly**, a control vector pXJ545 was constructed with a core cassette of NoDGAT2B
promoter-GFP-T_{tub}-P_{tub}-ble-T_{vcf} (Fig. S3F, shown below). **Secondly**, a reporter vector pXJ546
was constructed with a core cassette of NoDGAT2B-promoter-GFP-T_{tub}-P_{tub}-NobZIP77-
T_{psbA}-P_{tub}-ble-T_{vcf} (Fig. S3G, also shown below). **Thirdly**, pXJ545 and pXJ546 were
successfully transformed respectively into NobZIP77-knockout line NobZIP77ko1 (Fig. S5,
details please refer to Item #46 below).

The qRT-PCR results revealed that in the pXJ545-transformed *N. oceanica*, level of the GFP
transcript was reduced by 69.7% under N+ and by 58.8% under N-, versus their
counterparts in the pXJ546-transformed lines (Fig. 2B, as shown below). Therefore, the
transcription of GFP was interfered by NobZIP77-mediated NoDGAT2B transcription
inhibition, under both N+ and N-. Moreover, results from GFP fluorescence (Fig. 2C and Fig.

S8, shown below) suggested that, in the pXJ545-transformed lines, GFP fluorescence
 intensity (average fluorescence intensity per cell) was reduced by 55.8% under N+, versus
 their counterparts in the pXJ546-transformed lines. Therefore, the NobZIP77-mediated
 inhibition of *NoDGAT2B* transcription was confirmed.

Fig. S3F-3G. Vector maps of pXJ545 (F) and pXJ546 (G). P_{2B} : promoter of *NoDGAT2B*; P_{tub} :
 promoter of β -*tublin*; T_{tub} : terminator of α -*tublin*; *ble*: zeocin resistance gene; T_{psbA} :
 terminator of *psbA*; T_{vcp} : terminator of *violaxanthin/chlorophyll a binding protein*; *GFP*:
 green fluorescent protein.

 Fig. 2. *NoDGAT2B* is a downstream target of NobZIP77 which inhibits *NoDGAT2B*
 transcription by binding to the promoter. (B) Comparison of *GFP* transcript level between
 the *NoDGAT2B*-promoter transformed (p2B) and the NobZIP77-*NoDGAT2B*-promoter
 transformed (NobZIP77-p2B) lines of *N. oceanica*, under N+ or N- (24 h). (C) Quantification
 of *GFP* fluorescence (average fluorescence intensity per cell) in the p2B and NobZIP77-p2B
 lines. Values shown as mean \pm SD (n = 3). *: significant change ($p \leq 0.01$) versus p2B.

**Fig. S8. Confocal GFP-fluorescence images of *NoDGAT2B*-promoter-transformed *N.***
***oceanica* (A) and NobZIP77-*NoDGAT2B*-promoter-transformed *N. oceanica* (B). Three lines**
**of each transformed cassette were presented, which were also used for the quantification of**
**GFP-fluorescence intensity in Figure 2C. Green, GFP-fluorescence; red, plastid**
**autofluorescence.**

**To describe these new experimental results and clarify this topic, we have now inserted into**
**Results (Line 146-153): “To verify the inhibition of *NoDGAT2B* transcription by NobZIP77**
**in *N. oceanica*, a green fluorescent protein (GFP) reporter system that quantifies the level of**
***NoDGAT2B*-promoter-driven gene expression was designed (Methods). In the NobZIP77-**

*NoDGAT2B*-promoter-transformed *N. oceanica* (NobZIP77-p2B), *GFP* transcript level was
reduced by 69.7% under N+ and 58.8% under N-, versus the *NoDGAT2B*-promoter-
transformed lines (p2B; Fig. 2B). Moreover, in the NobZIP77-p2B lines, fluorescence
intensity of GFP was reduced by 55.8% versus the p2B lines (under N+; Fig. 2C and Fig. S8).
Thus NobZIP77 inhibits *NoDGAT2B* transcription *in vivo*.”

Correspondingly, we have inserted Methods (Line 648-662) as: “To validate the
NobZIP77-mediated *NoDGAT2B* inhibition in *N. oceanica*, a green fluorescent protein (GFP)
reporter system was designed and constructed. Firstly, the promoter sequence of *NoDGAT2B*
was isolated as above-mentioned (Table S1). Secondly, p*NoDGAT2B* was inserted into a
backbone vector of pXJ53 (containing a core cassette of *GFP- α -tubulin* terminator- *β -tubulin*
promoter-*ble-violaxanthin/chlorophyll a binding protein (VCP)* terminator) to produce the
control vector pXJ545 (Fig. S3F). Thirdly, the cassette of *β -tubulin* promoter-NobZIP77-*psbA*
terminator was isolated from pXJ459 (Fig. S3B) and subcloned into the interspace between
*α -tubulin* terminator and *β -tubulin* promoter in pXJ545, to produce the reporter vector
pXJ546 (Fig. S3G). Then pXJ545 and pXJ546 were transformed into *N. oceanica*,
respectively. For the PCR-positive lines, the level of *GFP* transcript was measured by
qRT-PCR, and the fluorescence of GFP detected via a laser-scanning confocal microscope.
GFP intensity was then quantified by Olympus Flowview Ver 4.0b. For each of the lines,
GFP intensity of ten cells was quantified and the average intensity per cell was used for
comparison across lines.”

11. 10) A recombinant maltose-binding-NobZIP77 fusion protein was expressed in *Escherichia*
*coli* and purified for gel shift assays to identify DNA binding activity of NpbZIP77. Although DNA
binding assays are notoriously unreliable, the figure presented here (Fig. 2C) is just not
convincing. A standard protein dilution generally results in increasing amounts of DNA shifted,
and the DNA/protein is often a monomer at low protein concentrations and then a protein dimer
at higher concentrations. Unlabeled DNA is then used to outcompete the labelled fragment.
None of that is observed in figure 2C, making it difficult to tell if the protein is really specific for
that DNA fragment or not. All DNA binding proteins have some non-specific affinity for DNA.

**Response:** As advised, we have now performed additional experiments to thoroughly
validate the specific binding of NobZIP77 to the *NoDGAT2B* promoter.

**Firstly**, to validate NobZIP77-targeted genes, we have performed an EMSA assay with five
NobZIP77-targeted genes, including *g3857* (*TAG-lipase*), *g2171* (*PAP*), *g10041* (*lipase*),
*NoDGAT2B* and *g5641* (*FAE*). The results are now presented in a new supplementary figure
of Fig. S9 (shown below). Its legend reads: “Fig. S9. EMSA-based validation of
NobZIP77-targeted genes. EMSA results revealed that NobZIP77 directly binds to the
promoters of *NoDGAT2B*, *g5641* (*FAE*), *g10411* (*lipase*), *g2171* (*PAP*) and *g3857*
(*TAG-lipase*). Unlabelled DNA of the promoters in 100-fold molar excess were treated with
the NobZIP77 protein. PAP, Phosphatidic acid phosphohydrolase; FAE, fatty acid elongase;
LS, lower shift complex (a 1:1 complex of DNA and the bZIP dimer); SS, super shift
complex; FP, free-DNA probe; COP, competitor oligonucleotide primer.”

**Secondly, we performed the EMSAs with different promoters, including three predicted**
 **target genes [*NoDGAT2B*, *g3857* (TAG-lipase) and *g10041* (lipase)] and the negative controls**
 **[*g4867* (*hsp70*) and *g9477* (actin)]. The results revealed that the recombinant NobZIP77 can**
 **bind specifically with promoters of predicted target genes, rather than other promoters, such**
 **as *g4867* and *g9477*. Thus we now provide a new version of Fig. 2D (formerly Fig. 2C), as**
 **shown below. The legend of Fig. 2D now reads: “(D) EMSA validation of specific binding**
 **between NobZIP77 and the promoters of its targeted genes (i.e., *NoDGAT2B*, *g3857* and**
 ***g10411*). *g3857*, TAG-lipase; *g10041*, lipase; *g4867*, *hsp70A*; *g9477*, actin Unlabelled DNA of**
 **the promoters in 100-fold molar excess was treated with the NobZIP77 protein. LS, lower**
 **shift complex (a 1:1 complex of DNA and the bZIP dimer); SS, super shift complex; FP,**
 **free-DNA probe; COP, competitor oligonucleotide primer.”**

Accordingly, in Results, the following was inserted (Line 154-165): “To probe the direct binding between NobZIP77 and the promoter of *NoDGAT2B* (and other predicted target genes; PTGs), electrophoretic mobility shift assays (EMSAs) were used to identify specific band shifts of ACGT-harboring promoter probes (the binding sites of bZIP-type TF genes²⁵). Recombinant maltose-binding protein (MBP) - NobZIP77 was expressed in *Escherichia coli* and purified for EMSA. ACGT-harboring probes of the pNobZIP77-PTGs [g3857 (TAG-lipase), g2171 (PAP), g10041 (lipase), *NoDGAT2B* and g5641 (FAE)] result in mobility shift, indicating the direct binding between the NobZIP77-PTG promoters and NobZIP77 (Fig. S9). Notably, the degree of mobility shift was weakened by adding unlabeled fragment of NobZIP77-PTG promoter (Fig. S9), yet no binding between NobZIP77 and the negative

controls of *g4867* (*hsp70*) and *g9477* (*actin*) was detected (Fig. 2D), which supports the
specificity of binding. Thus *NobZIP77* suppresses *NoDGAT2B* expression by binding to its
promoter via the ACGT core sequence.”

Moreover, in Methods, below was revised (Line 516-520): “As for probes, the promoters of
predicted *NobZIP77* target genes and *NobZIP77* target free genes were amplified with
primers labeled with biotin at the 5’ end (Table S3). To test the direct bindings between
*NobZIP77* and promoters, the EMSA assays were conducted via a LightShift®
Chemiluminescent EMSA Kit (Thermo Fisher, USA).”

**Comments from the Reviewer #2**

*12. This manuscript described a transcription factor, NobZIP77 (g77), as a blue light sensor and*
*a regulator of NoDGAT2B, leading to improved TAG production. There were merits in the*
*manuscript, including description of the new TF and its possible functions for lipid metabolism.*
*However, its direct interaction with the promoter of NoDGAT2B is overall weak: the in vitro*
*EMSA assay is incomplete, and no in vivo evidence (such as ChIP assays). There were conflicts*
*between its expression and that of the target, leaving an open question for different targets (such*
*as lipases listed in Table S1). Moreover, its localization as a TF in the nucleus is not complete.*
*There were many other points to be addressed for acceptance in Nature Communications.*

**Response:** We appreciate these inputs. For *in vitro* EMSA assay, please refer to our revision
in Item #11 above. For *in vivo* evidence, now we had constructed a report-system in *N.*
*oceanica*, as shown in Item #10.

For *NobZIP77* localization in the nucleus, we have confirmed it by DAPI staining as shown
in Fig. 4A (shown below). Specifically, below was inserted into Results (Line 235-237): “The
*NobZIP77:gfp* fusion protein was colocalized with the DAPI-stained nuclei, indicating
nucleus targeting (Fig. 4A).”

Moreover, below was inserted into Methods (Line 639-643): “To validate the nucleus-
localization of *NobZIP77*, the *NobZIP77:GFP* line and the WT strain were stained with 0.5
286 µg/mL 4',6-diamidino-2-phenylindole (DAPI, Solarbio) for 5 min at room temperature and
287 then twice-washed with Tris buffered saline Tween (TBST, Solarbio) for 4 min at room
temperature. Fluorescence of DAPI was excited at 405 nm and detected at 450-464 nm.”

**Fig. 4. A mechanistic model of N- induced TAG synthesis via the BL-NobZIP77-NoDGAT2B**
 **pathway in *N. oceanica*. (A) Subcellular localization of NobZIP77. GFP, green fluorescent**
 **protein; TML, transmission light; DAPI, 4',6-diamidino-2- phenylindole; PAF, plastid**
 **autofluorescence; scale bar, 2 μ m.**

*13. Table S1: The data presented in this table look interesting, but lack practical information,*
 *such as FPKM for both TFs and targets (LRGs), to show how much TF expression was induced*
 *and how target expression is affected. In addition, targets may have different (and opposite)*
 *impacts on lipid metabolism. For example, targets of g77 include DGATs and lipases, which*
 *warrants explanation.*

**Response:** As advised, we have now revised Table S1 (N-depletion induced TFs and their
 **putative LRGs in *N. oceanica*.) to include the information requested (shown in “Source**
 **Data-NCOMMS-20-28862-T” folder).**

**In addition, as advised, we have inserted the following into Results: “Altogether, 32 TFs**
 **respond to N-, with eleven up-regulated and fourteen down-regulated (seven of them**
 **showing distinct trends of regulation among time points). Of them, 16 TFs were predicted to**
 **regulate lipid-related genes (LRGs; Fig. 1A, Table S1) ¹¹, and scaffold00007.g77 (or g77,**
 **Genbank ID MT273120), which is down-regulated by N-, targets seven such genes that**
 **include both diacylglycerol acyltransferases (DGATs) and lipases (the enzymes play opposite**
 **roles on TAG metabolism (Fig. S1A, Table S1). Notably, transcripts in the LRG of g77**
 **exhibit distinct or even opposite fold-changes, e.g., up-regulation of the lipase gene g10411**

yet down- regulation of the DGAT gene of g6725 (Table S1). As the activities of lipases and
DGATs are exactly opposite (oil degradation and assembly respectively), these results
indicate g77 as a negative regulator of TAG synthesis.” (Line 76-85).

14. Figure 1B: If g77 is a blue light sensor, the structure should be drawn with and without its
chromophore.

**Response:** As advised, in Fig. S13, we have now provided the g77 structure with and without
the chromophore, as shown below. Its legend reads: “Fig. S13. Structural models of
NobZIP77 with and without flavin mononucleotide (FMN). The tertiary structure of
NobZIP77 (249-391) was modeled using a Circadian locomotor output cycles protein kaput
as initial template (15.44% sequence identity).”

15. Figures 2A and 2D: Expression of NoDGAT2B and TAG analyses were done in different time
scale. Why?

**Response:** In *N. oceanica*, there is usually ~24 h expression-lag (TAG production for even
longer time) to transcription, thus the sampling of TAG-quantification is 48 h longer than
qPCR. As advised, we have now inserted the following explanation into Fig. S10B legend:
“(B) TAG content ($\mu\text{g}/\text{mg}$ DW) under N+ (0 h) and N- (24 h, 48 h, 72 h and 96 h after
induction, considering ~48 h expression-lag to transcription in *N. oceanica*).” (Line 901-903).

16. Figure 2B: The reporter assay could have been better if performed in *Nannochloropsis*.

**Response:** Please refer to our response to Item #10 above.

17. Figure 2C: In my understanding, the super shift should be caused by adding an antibody
against g77. I think that the different sizes of shifted bands may represent formation of oligomers
or dramatic conformational changes of g77.

However, effects of unlabeled DNA are weak (even with 10 times more of it, as shown lanes
2,3,4), which may suggest non-specific binding of the promoter sequence of NoDGAT2B. This
should be considered critically, because this data is the only evidence for direct interaction
between g77 and NoDGAT2B promoter. It is also recommended to perform the same EMSA with
non-specific cold DNA (e.g., same amount of unrelated genomic DNA fragments) and compare to
that of specific cold probe.

In addition, bZIPs are usually transcriptional activator, not repressors (Figure 2A). With
these possible conflicts, it is recommended to run EMSA with different promoters, such as one of
the lipase genes (Table S1), which may have opposite impact on lipid metabolism, compared to
DGATs.

**Response:** Please refer to our response to Item #11 above.

18. Figure 2D: In addition to the % TAG (as shown here), it would be interesting to see TAG
contents, since many DGATs are known to increased TAG contents and yields.

**Response:** As advised, we have now changed % TAG to TAG content ($\mu\text{g}/\text{mg DW}$) in Fig. 2E
and Fig. S10B (shown below respectively). These results also support our original conclusion
that TAG content increased in 0 h and 24 h after N- induction, as compared to the EV
control (Line 175-178). The original plots of % TAG are now shown in Fig. S10C.

The updated legend of Fig. 2E reads: “(E) TAG content of the NoDGAT2B overexpression
and knockdown lines under N+ (0 h) and N- (24 h) in *N. oceanica*.”

The updated legend of Fig. S10B reads: “(B) TAG content ($\mu\text{g}/\text{mg DW}$) under N+ (0 h) and
 N- (24 h, 48 h, 72 h and 96 h after induction, considering ~ 48 h expression-lag to
 transcription in *N. oceanica*).”

 19. It is expected to have opposite effects between *NoDGAT2B* *ox* and *kd* on TAG contents, which
 was shown only on time 0. Effects of *ox* disappeared at later times, which requires explanation.
 One suggestion may be that *NoDGAT2B* may not be related to N-, but with N+ conditions.

**Response:** To explain this observation, our update version of Results stated: “Taken together,
 we propose an *in vivo* mechanism of N- induced TAG synthesis in *N. oceanica* (Fig. 4D).
 *NobZIP77* normally (i.e., N+) represses *NoDGAT2B* transcription by directly binding to the
 latter’s promoter; yet under N-, the plastid is no longer able to shield the *NobZIP77*-located
 nucleus from BL (due to the degradation of chlorophyll *a* in plastid), and the resulted
 exposure of *NobZIP77* to BL reduces its binding to *pNoDGAT2B* and elevates expression of
 the enzyme for TAG assembly preferably from C16:1. Moreover, the level of *NobZIP77*
 transcript is down-regulated under N- (Fig. 1A), which further facilitates expression of its
 downstream TAG-synthetic genes. Therefore, both BL exposure and *NobZIP77*
 down-regulation eventually lead to elevated expression of the enzyme which assembles TAG
 preferably from C16:1.” (Line 245-254).

As advised, to further highlight this point, we have now inserted the following explanation
 into Results (Line 178-180): “Notably, such difference in TAG content versus EV disappears
 at the late phase of N- (after 48 h), which may indicate the role of other *NoDGAT2s* in this
 process.”

20. Figure 3A-C: Quantitation of DNA binding should be measured with the shifted band, rather
than the saturated signals of the unbound. Actually, shifted bands appeared faster under the
dark.

**Response:** As advised, we have now quantified DNA binding with the shifted band in all the
new EMSA experiments (which replaced the original set of EMSA results). Please refer to
our response to Comment # 21 below.

21. Another question is about the same EMSA assay with the white light (rather than dark), since
all other experiments were done with white lights as a control.

**Response:** As advised, we have now inserted white light conditions in the EMSA assay, as
shown in Fig. 3A and Fig. S15 below.

The legend of Fig. 3A reads: “(A) DNA binding curves of NobZIP77 under the dark, white
light (WL) or blue light (BL). The curves were generated by quantifying the relative amount
of shifted bands of NobZIP77 bound to pNoDGAT2B in EMSA. Values shown as mean \pm SD
(n = 2). *: significant change ($p \leq 0.01$) versus darkness.”

The legend of Fig. S15 reads: “Fig. S15. Blue light reduces the binding of NobZIP77 to the
NoDGAT2B promoter. EMSAs under dark (A), blue light (BL) (B) or white light (WL) (C) in
the presence of 5 nM NoDGAT2B promoter DNA and varying amounts of the purified
NobZIP77 protein (a 1:2 dilution series of NobZIP77 with the maximum concentration of
24mM). SS (super shift complex) was used for gray value quantification, and compared
among dark, BL and WL.”

Correspondingly, the following was inserted into Results (Line 208-213): “To probe the effect
 of BL on NobZIP77-*NoDGAT2B* interaction, EMSA was performed in the dark, or under
 continuous illumination of white light (WL) or BL (Fig.S15; Methods). The shifting of
 NobZIP77-p*NoDGAT2B* band was observed starting at the NobZIP77 concentration of 0.012
 mM under the dark, 0.75 mM under BL, and 0.38 mM under WL (Fig. 3A). Thus, the
 binding between NobZIP77 and p*NoDGAT2B*, which shutdown *NoDGAT2B*-mediated TAG
 assembly, can be relieved by BL (and to a lesser degree, by WL).”

Moreover, into Methods below was inserted (Line 526-537): “To probe the effect of WL and
 BL on the NobZIP77-*NoDGAT2B* interaction, EMSA was performed either in the dark or
 under continuous WL or BL illumination. The labeled DNA probes (5nM) and varying
 amounts of purified NobZIP77 (a 1:2 dilution series of NobZIP77, with the maximal
 concentration of 24mM) were incubated in the binding buffer. The protein DNA mixtures
 were incubated at 25 °C for 60 min under darkness or continuous WL (400 $\mu\text{W cm}^{-2}$) or BL
 illumination (400 $\mu\text{W cm}^{-2}$ at 450 nm), and then analyzed on 6% native polyacrylamide gels.
 Gel electrophoresis was performed at 4°C for 60 min (100 V) under darkness or continuous

illumination of WL ($30 \mu\text{W cm}^{-2}$) or BL ($30 \mu\text{W cm}^{-2}$). Finally, the DNA was electroblotted
 onto a nitrocellulose membrane and detected via chemiluminescence. Super shifted (SS)
 bands of NobZIP77 bound to p*NoDGAT2B* were used for gray value quantification with
 Image-Pro Plus 6.0. For each gel, SS gray value of 24 mM NobZIP77 was defined as 100%.”

22. Figure 3C: Were EMSA assay repeated to reveal statistic significance? Binding with dark and
 blue light doesn't look significantly different. It would also be better to see blue light effects with
 N+ conditions.

**Response:** The EMSA assays in this work were all repeated to reveal statistic significance
 (e.g., in Fig. S15 above). Notably, as shown in both Fig. 3A and Fig. S15 (shown above),
 binding with dark and blue light looks very different.

As for the “blue light effects with N+ conditions”, we have already shown, based on
 comparison between Fig. 3C and Fig. 3E (shown below), that blue light slightly impairs the
 NobZIP77-mediated *NoDGAT2B* inhibition under N+ in *N. oceanica*.

**Fig. 3. Blue light reduces the binding of NobZIP77 to the *NoDGAT2B* promoter. (C, E)**
 **Transcript level of *NoDGAT2B* in WT (C) and in the knockout mutant of NobZIP77ko-1 (E).**
 **Values shown as mean \pm SD (n = 3). Letters above the bars indicate significant difference (p**
 **≤ 0.01), based on one-way analysis of variance (ANOVA) and Tukey’s honestly significant**
 **difference (HSD) test.**

23. Figures 3E,F: It is good to see employing Raman technology for measuring relative TAG
contents. Can a standard curve be created using cells with known TAG contents?

**Response:** As advised, we have now created a standard curve using cells with known TAG
contents measured by TLC-GC-MS and Raman spectra, as shown in a new Fig. S18 (below).

**Fig. S18. Quantification of TAG content (A) and yield (B) via ramanome.** The TAG content
($\mu\text{g}/\text{mg DW}$) and yield (mg/L) predicted using PLSR models via ramanome (Y axis; two of
the three ramanomes were used for train and one for test) were plotted versus the
corresponding value measured with TLC-GC-MS at the population level (X axis). PLSR:
partial least square regression. TLC-GC-MS: Thin-layer chromatography coupled with gas
chromatography-mass spectrometry. Correlation coefficients (R^2) were shown for train data
(red), test data (green) and all data (black).

Accordingly, the Methods text reads (Line 613-624): “For calculating oil content, both the
ramanome dataset and its corresponding TAG content/yield dataset of the TAG-producing
process of wild-type *N. oceanica* IMET1, which consists of the physiological state at each of
five timepoints in triplicate cultures (0 h, 24 h, 48 h, 72 h, 96 h) under N-, were collected as a
reference dataset. TAG content ($\mu\text{g}/\text{mg DW}$) and yield (mg/L) quantification models were
respectively established based on this reference dataset. Two of the triplicate cultures were
used as the training dataset and the remaining one as the test dataset for model validation.
PLSR models were established using the averaged SCRS and the corresponding TAG
content or yield by the training dataset and then validated by the test dataset¹³⁻¹⁵. The PLSR
model for TAG content ($\mu\text{g}/\text{mg DW}$) feature coefficient (R^2) of 1 for the training dataset and
0.9662 for the test dataset, with the overall coefficient of 0.9801. Similarly, the PLSR models
for TAG yield feature overall R^2 of 0.9801. These models were applied to quantify the TAG
content and TAG yield for new samples.”

24. It is also interesting to see effects of the green light: It is generally thought that green light is
not used for photosynthesis (and I don't recall any green light receptors), and can be assumed no

effects on lipid metabolism; however, it has effects under N- condition. How could it be?

**Response:** We appreciate this statement. Actually, green light (GL) was found to be effective
on photosynthesis in *Chlamydomonas reinhardtii* and *Chlorella variabilis*, indicating a
common response to GL (probably due to enhanced energy transfer from light-harvesting
chlorophyll protein complexes (LHCs) to both type-1 and type-2 photosystems (i.e., PSI and
PSII) in microalgae¹⁶. Therefore, our results in Fig. 3D and Fig. 3F (shown below) should
more or less support this point of GL-effect on lipid metabolism.

**Fig. 3. Blue light reduces the binding of NobZIP77 to the *NoDGAT2B* promoter. (D, F) TAG**
**content of WT (D) and NobZIP77ko-1 (F). Values shown as mean ±SD (n = 3). Letters above**
**the bars indicate significant difference ($p \leq 0.01$), based on one-way analysis of variance**
**(ANOVA) and Tukey's honestly significant difference (HSD) test.**

25. Another point to note: Blue light still has positive effects in *g77* KO line under N- condition,
which leaves a question for additional blue light sensors.

**Response:** Blue light still has positive effects in *NobZIP77*-knockout line under N- condition,
which leaves a possibility for additional blue light sensors. Actually, there are three
aureochrome-type blue light receptors in *N. oceanica*.

26. Figure 4A: Nuclear localization for g77 should be confirmed by nuclear staining using DAPI.
Localization of NoDGAT2B is not clear. It should be co-localized with sDer1:RFP fusion. If it is
associated with lipid droplets, it can also be co-stained with Bodipy or Nile red.

**Response:** As advised, for the nuclear localization of g77, we have now confirmed it by
nuclear staining using DAPI. Please see details in our response to Comment #12.

As for NoDGAT2B, to validate its intracellular localization, we have constructed a vector
pXJ542 with “P_{vcp}-NoDGAT2B-GFP-T_{tub}-P_{NO20G00500}-sDer1-sfCherry-T_{NO08G03280}-P_{tub}-ble-T_{vcp}”
as a core cassette (shown below), and transformed it into *N. oceanica*. In the
pXJ542-transformed lines, only the GFP signal but not the sfCherry signal was detected
(shown below), despite our multiple repeated attempts.

**Vector map of pXJ542.** P_{vcp}/T_{vcp}: promoter/terminator of *violaxanthin/chlorophyll a binding*
*protein*, P_{NO20G00500}: promoter of *NO20G00500* (see gene details in NanDeSyn:
<http://www.nandesyn.org>); T_{NO08G03280}: terminator of *NO08G03280* (details in NanDeSyn);
T_{tub}: terminator of *α-tubulin*; *ble*: zeocin resistance gene; *GFP*: green fluorescent protein,
*sfCherry*: optimized cheery fluorescent protein.

**Co-localization of NoDGAT2B and sDer1.** GFP, green fluorescent protein; TML,
transmission light; sfCherry, optimized cheery fluorescent protein; PAF, plastid
autofluorescence; scale bar, 2 μm.

On the other hand, in the NoDGAT2B-localizing *N. oceanica* lines of pXJ505 (see Line
233-235 in Results of the main text), our Nile red-staining experiments have confirmed the
localization of NoDGAT2B at lipid droplets (see Fig. 4B below).

**Fig. 4. A mechanistic model of N- induced TAG synthesis via the BL-NobZIP77-NoDGAT2B**
 **pathway in *N. oceanica*. (B) Subcellular localization of NoDGAT2B. GFP, green fluorescent**
 **protein; TML, transmission light; PAF, plastid autofluorescence; scale bar, 2 μ m.**

 Therefore, based on these new experimental results, we have revised the Results as (Line
 237-238): “the NoDGAT2B:gfp fusion protein was colocalized with the Nile red-stained lipid
 droplet (LD), suggesting LD targeting (Fig. 4B).”.

 Correspondingly, we inserted the following description of NoDGAT2B localization
 experiments into Methods (Line 644-647) “To confirm the subcellular localization of
 NoDGATA2B, the NoDGAT2B:GFP line and *N. oceanica* WT were stained with 0.25 mg/mL
 Nile red (Solarbio) for 30 min at room temperature and then twice-washed with distilled
 water. Fluorescence of Nile red was excited at 488 nm and detected at 560-590 nm.”

 27. Figure 5: Experiments were done with different time schedules, and it is hard to directly
 compare data directly. Time values are misplaced in Figure 5A.

 **Response:** As advised, now Figure 5 has been revised to ensure an identical time schedule
 among Fig. 5A, 5B and 5C (shown below). Meanwhile, placement of time values was
 corrected in Fig. 5A.

 **Fig. 5. A rational approach to elevate oil productivity by exploiting the NobZIP77 signaling**
 **pathway. TAG yield (mg/L) was compared between (i) wild-type (WT) under white light**
 **(WL) and blue light (BL) plus N- (A), (ii) WT and NobZIP77ko-1 (KO) under WL and N+**
 **(B), or (iii) WT-BL and KO-BL under N- (C). Values shown as mean \pm SD (n = 3). *:**

significant change ($p \leq 0.01$) versus WT-WL.

28. Figure 6: Explain why blue light induction was done for KO mutants only. To claim its effect,
the same treatment should be applied to WT.

**Response:** In both Fig. 5 (shown above) and Fig. S16 (shown below), we have presented the
data for blue light induction of WT under both N+ and N-. The results suggested that BL is a
key factor in TAG improvement without inhibiting growth, especially under N-. Therefore,
in the experiment design of Fig. 6, we did NOT include blue light induction of WT. In fact,
we tend to use Fig. 6 to present the advantage of BL plus *NobZIP77*-KO-approach, which
was compared to continuous white-light plus WT (i.e., traditional way for microalgal
cultivation).

To avoid potential confusion, as advised, we have inserted the following into Fig. 6 legend
(Line 409-412): “This experiment was conducted to present the advantage of BL plus
*NobZIP77*-KO-approach, as compared to continuous WL plus WT (the conventional way of
microalgal cultivation). Data for BL induction of WT under both N+ and N- are presented in
both Fig. 5 and Fig. S16.”

**Fig. S16. Phenotypes of *N. oceanica* NobZIP77 mutant and WT cultivated under blue light.**
**Growth kinetics (cell density, 10^7 cells/mL), TAG content (mg/g DW) and TAG yield (mg/L)**
**were compared for (A) between wild type (WT) under white light (WL) and WT under blue**
**light (BL), (B) WT and NobZIP77ko-1 under WL, and (C) WT under WL and**
**NobZIP77ko-1 under BL, under both nitrogen-replete (N+) and nitrogen-depleted (N-)**
**conditions. Values shown as mean \pm SD (n = 3). *: significant change ($p \leq 0.01$).**

29. * The letters (B and C) are misplaced.

Response: Corrected as advised.

30. Figure S1A: Actually, expression patterns for *g77* transcripts and protein are not consistent. Protein level increased at six hours, and then dramatically decreased at 12 hrs; however, transcripts levels were relatively consistent. Explain why.

**Response:** In fact, the data of expression patterns for g77 transcripts and protein are from
different experimental batches. Protein level of g77 increased a little, while decreased at
other four timepoints.

To avoid potential confusion, as advised, we have inserted below into Fig. S1A legend: “The
transcript and protein expression patterns, which are consistent at 3 h, 12 h, 24 h and 48 h
under N- induction. The transcriptome data was from our group ¹⁷. Fold change was
calculated as $\log_2(\text{FPKM}(\text{Tx}, \text{N-})/\text{FPKM}(\text{Tx}, \text{N+}))$ (FPKM, the normalized abundance of
transcript; Tx, time point). The protein data was also from our past study ¹⁸. Fold change
was calculated as (LFQ intensity (N-) - LFQ intensity (N+)). LFQ, label-free relative
quantification.”

31. More importantly, expression patterns for NoDGAT2B and other targets (also pointed out in
Table S1). True interacting targets should be selected from this.

**Response:** To clarify why NoDGAT2B was selected, we have inserted the following
explanation into Results (Line 138-145): “Notably, NoDGAT2B (g6725; Genbank ID
KX867957), a novel member of the *N. oceanica* type-2 DGAT family ^{9,10}, is the only LRG
that exhibits a temporal transcription pattern precisely opposite to NobZIP77 (consistent
with the transcriptome data; Fig. 2A; Table S1). The NoDGAT2B transcript is 127.5%
higher in NobZIP77i-2 while 26.0% lower in NobZIP77o-2 (average fold-change over 0 h, 6 h,
24 h and 48 h under N-; versus EV; Fig. 2A), which suggests its repression by NobZIP77.
Such a trend of transcript fold-change is also shared by the PAP (g2171) and the lipase
(g4187) at the early phase of N- (Fig. S7). These results suggest NoDGAT2B as one key target
of NobZIP77.”

32. Figure S2: There are multiple bZIPs in algae and many more in plants. Explain how the
representative bZIPs were chosen. Are they all related to blue light sensing?

**Response:** As advised, we have now revised the relevant statement in Fig. S2A legend as
(Line 833-837): “Alignment of amino acid sequences among bZIP domains from model
species including animals, higher plants, fungi and microalgae. The bZIP domains were
chosen from published proteins sequences that carry the domain. Notably, among the
proteins shown, only g77 carries the LOV domain (which is related to blue light sensing).”

33. Figure S4: Expression levels of g77 were analyzed up to 48 hrs, but TAG was analyzed up to
96 hrs. Explain why.

**Response:** Please refer to our response to Item #15 above.

34. Figure S4C showed only TAG per total lipids. It is recommended to show a graph for total
lipid contents.

**Response:** As advised, we have now inserted Fig. S4F and the corresponding results (Line

111-113) for total lipid contents: “As for total fatty acids (TFA) content, no significant
differences from EV was observed in *NobZIP77* overexpression or knockdown lines (Fig. S4F).”

**Fig. S4. Phenotypes of *NobZIP77* overexpression and knockdown in *N. oceanica*. (F) The**
**content of total fatty acids (TFAs) under N+ and N-. Values shown as mean \pm SD (n = 3). *:**
**significant change ($p \leq 0.01$) versus EV.**

35. In addition, expression patterns of *g77* is more or less consistent as shown in Figure S4A, but
its impact on TAG content (S4C) and TAG yields (S4D) are different (mostly at 0 and 24 hrs).
Explain why.

**Response:** As advised, we have inserted the following into Results (Line 116-119): “Notably,
among the two overexpression lines (and also among the knockdown lines), the expression
pattern of *NobZIP77* is consistent, but not TAG content or yield. This can be due to the
unpredictable effect of integration of overexpression cassette or silencing of the RNAi
constructs.”

36. *The gene name was initially defined as *g77*, but *NobZIP77* was also used later. It would be
better to use consistent names.

**Response:** As advised, we have shown both “*g77*” and “*NobZIP77*” at its first appearance
(Line 90-91; since at that stage it was not clear yet *g77* belongs to the bZIP family), and then
use “*NobZIP77*” at all instances afterwards.

37. Figure S5: Please indicate gene names together with their id. Induction patterns for different
 targets are different, and some are opposite. Explain why.

**Response:** As advised, we have now specified gene names together with their ID in Fig. S7
 (formerly Fig. S5), as shown below.

**Fig. S7. Relative transcript abundance of six lipid-related target genes of NobZIP77 in EV,**
 **NobZIP77o-2 and NobZIP77i-2 at 0 h, 6 h, 24 h and 48 h under N-. Values shown as mean \pm**
 **SD (n = 3). *: significant change ($p \leq 0.01$) versus EV, empty vector control; PAP,**
 **Phosphatidic acid phosphohydrolase; LSD, lipid-sensing domain containing protein; FAE,**
 **fatty acid elongase.**

Moreover, we have now inserted the following explanation into Results (Line 141-145): “The
 *NoDGAT2B* transcript is 127.5% higher in NobZIP77i-2 while 26.0% lower in NobZIP77o-2
 (average fold-change over 0 h, 6 h, 24 h and 48 h under N-; versus EV; Fig. 2A), which
 suggests its repression by NobZIP77. Such a trend of transcript fold-change is also shared by
 the PAP (g2171) and the lipase (g4187) at the early phase of N- (Fig. S7).”

38. Figure S7: LOV domains can be found in different types of proteins, and it would be
 interesting to reveal identities of proteins that contain the domain.

**Response:** As advised, we have now inserted into Fig. S14B (formerly Fig. S7B) the identity
 and functional annotation of proteins, as shown below.

**Fig. S14. Phylogenetic analysis of the LOV domain of NobZIP77. (B) Cladogram of selected**

LOV protein sequences from higher plants, fungi and microalgae. Neighbor-joining (NJ)
 method was used for tree construction. Cladogram was plotted based on actual branch
 length. GenBank accession numbers are provided in brackets. The numbers beside the
 branch stand for bootstrap value for NJ. “-” indicated values < 50. Red arrow represents
 NobZIP77.

39. Overall English writing needs improvement by professional writers, since many statements
 were not clear and can be misleading; for example, in Abstract: "a blue-light-oil-induction
 strategy that exposes NobZIP77-knockout *N. oceanica* to white light and then to blue light doubles
 peak TAG productivity" What does it mean? It is not clear (and actually confusing) how the blue
 and white lights are used and how they are related, leaving a room for confusion and
 misunderstanding.

**Response:** As advised, we have now improved the English writing throughout the
 manuscript. In particular, the sentence in Abstract is now changed to:

“Moreover, a new oil-induction strategy that exposes the *NobZIP77*-knockout line to BL
 under N- doubles peak TAG productivity.”

40. In my version of the manuscript, references 1-38 are missing.

**Response:** We have double checked the original manuscript and these references were there,
 as shown in the screenshot below.

bioreactor and bioprocess design.
Finally, such a concise pathway that directly links light sensing to metabolite
synthesis was not previously considered possible. In animal, plant and microbes,
known pathways that regulate metabolite synthesis in response to light quality
usually consist of multiple layers of signal transduction³⁸. On the other hand,
auxochromes, although known to regulate photomorphogenesis, sexual cycle and
biological rhythm³⁹, have not been implicated in the synthesis of carbon-storage
compounds. Therefore, unravelling of the BL-3/bZIP77-NoDGAT2B pathway, as a
simplest genetic circuit that directly links BL to metabolite synthesis, advocates
*Nannochloropsis* spp. as an advantageous model system for optogenetics and should
enable the rational design of opto-controlled algal and plant cell factories.
References
1. Wood, E.J. 'Lipids', Edn. 7. (Lehninger, New York; 2000).
2. Gimpel, J.A., Specht, E.A., Georgianna, D.R. & Mayfield, S.P. Advances in
microalgal engineering and synthetic biology applications for biofuel production.
*Curr Opin Chem Biol* 17, 489-495 (2013).
3. Hu, Q. *et al.* Microalgal triacylglycerols as feedstocks for biofuel production:
perspectives and advances. *Plant J* 54, 621-639 (2008).
4. Wijffels, R.H. & Barbosa, M.J. An outlook on microalgal biofuels. *Science* 329,
796-799 (2010).
5. Rodolfi, L. *et al.* Microalgae for oil. Strain selection, induction of lipid synthesis

and outdoor mass cultivation in a low-cost photobioreactor. *Biotechnology for*
*Biofuels* 102, 100-112 (2009).
6. Sun, X. *et al.* Effect of nitrogen-starvation, light intensity and iron on
triacylglyceride/carbohydrate production and fatty acid profile of *Nannochloris*
*oleoabundans* HK-129 by a two-stage process. *Biotechnol Bioeng* 155, 204-212
(2014).
7. Aroca, N., Pienkos, P.T., Pruthi, V., Polari, K.M. & Guarnieri, M.T. Leveraging
algal omics to reveal potential targets for augmenting TAG accumulation.
*Biotechnol Adv* 36, 1274-1292 (2018).
8. Xin, Y. *et al.* Producing designer oils in industrial microalgae by rational
modulation of co-evolving type-2 diacylglycerol acyltransferases. *Mol Plant* 10,
1523-1539 (2017).
9. Xin, Y. *et al.* Biosynthesis of triacylglycerol molecules with a tailored PUFA
profile in industrial microalgae. *Mol Plant* 20, 30374-30385 (2019).
10. Zienkiewicz, K. *et al.* *Nannochloropsis*, a rich source of diacylglycerol
acyltransferases for engineering of triacylglycerol content in different hosts.
*Biotechnology for Biofuels* 10, 8-27 (2017).
11. Du, Z.Y. & Benning, C. Triacylglycerol accumulation in photosynthetic cells in
plants and algae. *Subcell Biochem* 86, 179-205 (2016).
12. Banerjee, A., Maiti, S.K., Guria, C. & Banerjee, C. Metabolic pathways for lipid
synthesis under nitrogen stress in *Chlamydomonas* and *Nannochloropsis*.
*Biotechnol Lett* 39, 1-11 (2017).

13. Ajjawi, I. *et al.* Lipid production in *Nannochloropsis gaditana* is doubled by
decreasing expression of a single transcriptional regulator. *Nat Biotechnol* 35,
647-652 (2017).
14. Kang, N.K. *et al.* Increased lipid production by heterologous expression of
AtWRK1 transcription factor in *Nannochloropsis salina*. *Biotechnology for*
*Biofuels* 10, 231-244 (2017).
15. Kwon, S. *et al.* Enhancement of biomass and lipid productivity by
overexpression of a bZIP transcription factor in *Nannochloropsis salina*.
*Biotechnol Bioeng* 115, 331-340 (2018).
16. Ryu, A.J. *et al.* Development and characterization of a *Nannochloropsis* mutant
with simultaneously enhanced growth and lipid production. *Biotechnology for*
*Biofuels* 13, 38 (2020).
17. Wei, H. *et al.* A type-I diacylglycerol acyltransferase modulates triacylglycerol
biosynthesis and fatty acid composition in the oleaginous microalga,
*Nannochloropsis oceanica*. *Biotechnology for Biofuels* 10, 174 (2017).
18. Hu, J. *et al.* Genome-wide identification of transcription factors and
transcription-factor binding sites in oleaginous microalgae *Nannochloropsis*. *Sci*
*Rep* 4, 5454 (2014).
19. Li, J. *et al.* Chereography of transcriptomes and lipidomes of *Nannochloropsis*
reveals the mechanisms of oil synthesis in microalgae. *Plant Cell* 26, 1645-1665
(2014).
20. Jun, D., Xiaonan, L. & Haiyan, H. Systematic prediction of cis-regulatory
elements in the *Chlamydomonas reinhardtii* genome using comparative genomics.
*Plant Physiol* 160, 613-623 (2012).
21. Yoo, S.-D., Cho, Y.-H. & Sheen, J. *Arabidopsis* mesophyll protoplasts: a
versatile cell system for transient gene expression analysis. *Nat Protoc* 2,
1565-1572 (2007).
22. Jacoby, M. *et al.* bZIP transcription factors in *Arabidopsis*. *Trends Plant Sci* 7,
106-111 (2002).
23. Heinz, U. & Schlichting, I. Blue light-induced LOV domain dimerization
enhances the affinity of Auxochrome 1a for its target DNA sequence. *Elife* 5,
e11860 (2015).
24. Maiti, A.B. & Chetkova, E.M. Auxochromes - blue light receptors.
*Biochemistry* 83, 662-673 (2018).
25. He, Y. *et al.* Label-free, simultaneous quantification of starch, protein and
triacylglycerol in single microalgal cells. *Biotechnol Biofuels* 10, 275-292
(2017).
26. Moog, D., Stork, S., Reischner, S., Grouche, C. & Meier, U.G. *In vivo*
localization studies in the *Stramenopiles* alga *Nannochloropsis oceanica*. *Protein*
166, 161-171 (2015).
27. Suknik, A., Livne, A., Apt, K.E. & Grossman, A.R. Characterization of a gene
encoding the light-harvesting violaxanthin-chlorophyll protein of
*Nannochloropsis* sp. (*Eustigmatophyceae*). *J Phycol* 36, 563-570 (2010).
28. Simionato, D. *et al.* The response of *Nannochloropsis gaditana* to nitrogen

starvation includes *de novo* biosynthesis of triacylglycerols, a decrease of
chloroplast galactolipids, and reorganization of the photosynthetic apparatus.
*Eukaryot Cell* 12, 665-676 (2013).
29. Takahashi, F. *et al.* AUREOCHROME, a photoreceptor required for
photomorphogenesis in stramenopiles. *P Natl Acad Sci USA* 104, 19625-19630
(2007).
30. Yang, X. & Cai, X.F. Designing microalgal oils. *Mol Plant* 12, 472-473 (2019).
31. Katsuka, H. Phototropic responses of *vaucheria-geminata* to intermittent
blue-light stimuli. *Plant Physiol* 63, 1107-1110 (1979).
32. Gehring, W. & Rosbash, M. The coevolution of blue-light photoreception and
circadian rhythms. *J Mol Evol* 57, S286-S289 (2003).
33. Kroth, P.G., Wilhelm, C. & Kottke, T. An update on auxochromes:
Phylogeny-mechanism-function. *J Plant Physiol* 217, 20-26 (2017).
34. Wang, L. *et al.* Autonomic behaviors in lipase-actives oil droplets. *Angew Chem*
*Int Edit* 58, 1067-1071 (2019).
35. Polinet, E., Takeuchi, T., Du, Z.Y., Benning, C. & Furo, E. Non-transgenic
marker-free gene disruption by an epigenomal CRISPR system in the oleaginous
microalga, *Nannochloropsis oceanica* CCM1779. *ACS Synth Biol* 7, 962-968
(2018).
36. Sanchez, C. *et al.* Combinatorial control of light induced chromatin remodeling
and gene activation in *Neurospora*. *PLoS Genet* 11, e1005105 (2015).
37. Fermo, L., Yizhar, O. & Deisseroth, K. The development and application of

optogenetics. *Annu Rev Neurosci* 34, 389-412 (2011).
38. Wang, W.X. *et al.* Photoexcited CRYPTOCHROME1 interacts with
dephosphorylated BES1 to regulate brassinosteroid signaling and
photomorphogenesis in *Arabidopsis*. *Plant Cell* 36, 1989-2005 (2018).
Acknowledgements
This work was supported by grants from National Key Research and
Development Program (2018YFA0902501), Natural Science Foundation of China
(31600059, 31425002, 31900074, 31900047 and 31800071), Postdoctoral Science
Foundation of China (2018M642716) and Natural Science Foundation of Shandong,
China (ZR201807090262 and ZR201709180185).
Author contributions
450 J.X., Y.X. and P.Z. designed research; P.Z. and X.T. performed the EMSA assay;
P.Z. and X.T. performed the protoplast transformation and transient expression
assays; P.Z., Y.X. Q.W. and N.L. generated and screened transgenic lines of *N.*
*oceanica*; P.Z., Y.X. and Y.L. conducted the GC-MS assay; Y.H. and P.Z. performed
Raman sampling, measurement and analysis; C.S. and P.Z. performed the protein
localization assay; Y.X. and P.Z. performed phylogenetic analysis; Q.H. provided
inputs; J.X., Y.X. and P.Z. analyzed data and wrote the paper.
Competing financial interests
The authors declare competing financial interests: details are available in the
online version of the paper.

41. Line 83: *Totally*?

Response: As advised, we have now replaced “Totally”, and the sentence now reads: “Altogether, 32 TFs respond to N-, with eleven up-regulated and fourteen down-regulated (seven of them showing distinct trends of regulation among time points).” (Line 76-77).

Comments from the Reviewer #3

42. It is well known that microalgae produce a high amount of TAGs upon nitrogen depletion, *Nannochloropsis* is especially famous for this and proposed to be one major algae with interesting industrial potential. However, the signaling mechanisms triggering the metabolic switch from a carbohydrate/protein metabolism to a lipid metabolism are not well understood.

*In this manuscript, the authors, based on bioinformatics analyses, identified an interesting*
*transcription factor (NobZIP77) which might be involved in regulating lipid content after nitrogen*
*depletion. They produce knockdown, overexpressing and KO mutants of this factor and*
*characterize them with respect to growth and lipid production, also including testing the effect of*
*different light colors. They identify this transcription factor to be an aureochrome, a blue light*
*photoreceptor with transcription factor activity, a peculiarity so far only found in stramenopiles.*
*Moreover, the authors propose a putative target gene of this TF, a diacylglycerol acyltransferase*
*(NoDGAT2B) for which they additionally produce knockdown and overexpressing lines and*
*characterize them with respect to growth and lipid production. Moreover, they conclusively*
*demonstrate an interaction of NobZIP77 with NoDGAT2B promoter by EMSA analyses and GUS*
*reporter assays. They show that NobZIP77 represses NoDGAT2B expression, and this repression*
*is released upon blue light exposure. Interestingly, they show that using this knowledge and the*
*KO mutant, higher lipid yields can be obtained without compromising cell growth, which is*
*normally, using the N-limitation strategy, a major problem for upscaling microalgae oil*
*production to an economically interesting scale.*

*The results are novel and widely interesting for the field. The experimental strategy is*
*well-thought, the data presentation is clear, the manuscript well written and the interpretation of*
*the data is mostly justified, including an interesting working model.*

**Response: We thank the reviewer for these very positive and encouraging comments.**

*43. Besides several major and minor issues specified below, there is one dominant problem which*
*definitely needs to be addressed exhaustively:*

*Actually there is only a significant differential regulation of NobZIP77 under nitrogen*
*deplete conditions in the range of 12-48 hours, which made the authors initially chose this factor*
*for further analysis. However, comparing the ov. and sil. mutants to the WT (or EV) under these*
*conditions, there is little or no effect on the TAG accumulation (more specified below in the*
*specific comments), the largest effect seems actually always to be at nitrogen replete condition*
*(time point 0), which makes all interpretation regarding the effects of nitrogen limitation on TAG*
*accumulation via NobZIP77 difficult to understand. A major reason for this may be that both,*
*silencing and overexpression of NobZIP77, was very little effective, with reduction of transcript*
*level (not talking about the protein level which has not been shown) only to be in the range of 20 %*
*and overexpression only approximatively 2 fold. While the effects of overexpression, especially of*
*TFs, in general can be very ambiguous, a knockdown and especially a KO clearly provides more*
*reliable results. As the authors have produced KO lines of NobZIP77, I strongly recommend to*
*repeat experiments of Fig.1c, 2a and especially S4 with the KO line. This may clarify to which*
*extent NobZIP77 plays a direct role in regulating TAG and FA synthesis under nitrogen depletion.*
*Based on Suppl. Fig. 8, the effect might be marginal of even not present, in line with Fig. S4, and*
*possibly NobZIP only plays a role under N+, which then would need to be discussed accordingly.*

**Response: We agree this statement. Now we had repeat experiments of formerly Fig. 1C, and**
**Fig. S4 on that NobZIP77-KO lines. The results exhibited a significant difference of TAG**
**contents between KO lines and WT, under early phase of N-. We have now inserted the Fig.**
**1D and Fig. S6, as shown below.**

**Fig. 1. Identification of NobZIP77 (g77) as a key TF on N- induced TAG synthesis. (D) TAG**
 **content and TAG yield of the *NobZIP77* knockout and complementation lines under N+.**
 **Values shown as mean \pm SD (n = 3). *: significant change ($p \leq 0.01$) versus WT.**

 **Fig. S6. Phenotypes of *NobZIP77* knockout and complementation in *N. oceanica*. (A) Growth**
 **kinetics of the *NobZIP77* transgenic lines and wild type (WT) under N+. (B) TAG content**
 **under N+ (0 h) and N- (24 h, 48 h, 72 h and 96 h after induction). (C) TAG yield at 0 h, 24 h,**
 **48 h, 72 h and 96 h after onset of N-. (D) Fatty-acid composition of TAG at 0 h under N-. (E)**
 **TFA content at 0 h, 24 h, 48 h, 72 h and 96 h after onset of N-. Values shown as mean \pm SD (n**
 **= 3). *: significant change ($p \leq 0.01$) versus WT. NobZIP77ko1 and ko2: knockout lines;**
 **NobZIP77c1 and c2: complementation lines.**

Correspondingly, we have inserted the following into Results (Line 119-131): “To further
pinpoint its *in vivo* role, *NobZIP77* was complemented in its knockout lines (Fig. 1D; Fig. S5;
Methods). Under N+, knockout and complementation showed essentially no effects on
microalgal growth (Fig. S6A). However, significant difference in TAG content was observed
in *NobZIP77*-knockout lines, yet disappeared in the complemented lines. Specifically, versus
wild type (WT) and under N-, TAG content was 31.6 - 62.9% higher over 0 to 48 h for
*NobZIP77ko-1*, while 70.4% and 29.9% higher at 0 h and 24 h for *NobZIP77ko-2* (Fig. 1D
and Fig. S6B). For TAG yield, *NobZIP77* knockout resulted in 61.6% and 63.6% increase at
0 h respectively (52.7% and 44.4% at 24 h; under N-; Fig. S6C). Notably, TAG-associated
C16:1 and C18:2 are 26.0-32.4% and 59.0-59.5% lower in the *NobZIP77*-knockout lines
respectively under N+ (versus WT; Fig. S6D). As for TFA content, no difference was found
versus WT via knockout or complementation, under either N- or N+ (Fig. S6E). These
results are consistent with the *NobZIP77* knockdown and overexpression experiments
above.”

44. 1) Line 100: “Under N+, *NobZIP77* overexpression results in 9.03% and 1.23% lower
growth rates than EV, while *NobZIP77* knockdown lines remained unchanged (Fig. S4B).

How do you calculate a 1.23 % lower growth rate based on this figure, while there is no
change in overexpression lines? What is the math behind? And is this at all significant? (By the
way: two digits after the comma here is a bit over precise). I am also wondering why the growth is
exactly linear during your 8 days incubation period. This means neither is there any exponential
increase in growth nor a clear stationary phase. Everyday there are around 3 million cells more
than the day before, a somehow weird situation in algae growth experiments.

**Response:** As advised, the numbers are calculated based on the average growth rate during
eight days incubation period. To avoid any potential confusion, we have now changed the
statement from “Under N+, ... remained unchanged (Fig. S4B).” to “Under N+, *NobZIP77*
overexpression results in 9.03% and 1.23% lower growth rates (the average over the
eight-day culture) than EV, while *NobZIP77* knockdown lines remained unchanged (Fig.
S4B).” (Line 97-99).

45. 2) Regarding the amount of TAGs (Fig. S4) I am a bit skeptical (this comment relates to the
main comment above): First, there are huge differences between silencing line 2 compared to
silencing line 1, despite basically the same reduction in RNA along all time points. And actually
only silencing line 2 shows significantly elevated TAG contents compared to the Wt at time point
24, 48 and 72 hours. Second, there is no real difference in TAG content of overexpressing lines
compared to the silencing lines at time point 48 and 96 and also at time point 72 Wt, both
overexpressing lines and the first silencing line show very similar TAG profiles.

**Response:** Please refer to our response to Item #43 above.

46. 3) I strongly miss any experimental data demonstrating that the generated
*NobZIP77*-CRISPR/Cas9 lines are indeed KO lines. It is described in the paper but has to be
shown showing e.g. the PCR result including sequencing of the mutated fragment and/or Southern

blot and/or (if antibody is available) Western blot. Also, as the observed differential effects on
TAG accumulation in the KO line compared to WT are rather moderate, not to say small, also
complemented strains should be created to exclude any off-target effect.

**Response:** As advised, we have provided the genome sequences of the editing sites in the
NobZIP77-knockout lines (shown below and Fig. S5).

**Fig. S5.** Genome sequences of the editing sites in the NobZIP77-knockout lines and wild type
of *N. oceanica*. Sequences of the generated NobZIP77-CRISPR/Cas9 lines are verified as
NobZIP77-KO lines with one cytosine (C) deleted in NobZIP77ko1 and two cytosines (C)
deleted in NobZIP77ko2.

Moreover, as advised, complemented strains (NobZIP77c1 and NobZIP77c1) of
NobZIP77-KO lines had been created (as shown in Fig. 1D and Fig. S6 above). No change
was observed among these complemented strains and WT, thus off-target effect should be
excluded.

47. 4) L199: “In NobZIP77ko-1, the TAG content is reduced by 11.78% under BL plus N-,
revealing a temporal lag in TAG content in response to BL under N- as compared to the wild-type
(Fig. 3E; Fig. 3F), although their light-quality-responsive pattern appears similar, i.e., stronger

stimulation of TAG by BL than by GL (for NobZIP77ko-1, under N-, the TAG content exhibits
19.69% increase under GL and 49.83% increase under BL versus the dark; Fig. 3F). Therefore,
N-induced TAG synthesis is mediated by NobZIP77 via its sensing of BL.”

I only partly agree: Clearly, TAG synthesis is stimulated by light, as N.oc. needs carbon and
reducing equivalents for this. The reason for the much stronger effect of BL than green light is
that the authors in both cases applied the same amount of light intensity, i.e. 50 $\mu\text{mol photons m}^{-2}$
s⁻¹. As blue light is much better absorbed than green light by the cell, simply more carbon and
reducing equivalents are available for TAG synthesis under BL than under GL. The fact that the
KO of NobZIP77 only slightly diminishes the amount of TAG accumulation under BL indicates
that it is only little involved in this response, but for sure it is not a prime regulator, otherwise
there should be no increase in TAG in BL at all. Hence, please formulate much more carefully.

In future experiments the authors should pay attention to the fact that when comparing the
effect of different light qualities, the differential absorption of the light colors by the cells has to be
corrected, e.g. by calculating QPhar (e.g. check Jungandreas et al. 2014, PLoS One).

**Response:** We greatly appreciate this thoughtful suggestion about correcting for differential
absorption of the light color by the cells. Since this Q_{Phar} method is not involved in our
updated results, it will be carefully considered in our future experiments.

48. 5) Also, while there is extensive data on gene regulation of NobZIP77 and NoDGAT2B under
white light and N-, nothing is provided on BL/GL. Please provide these data for the experimental
conditions where you analyzed TAG.

**Response:** As advised, we have now provided additional experimental results (via qPCR) in
Fig. 3C and Fig. 3E (shown in Item #22).

Accordingly, into Results, the following was inserted (Line 215-220): “For WT, under BL
(but not green light (GL)), the NoDGAT2B transcript is 1.6- and 13.3-folds higher than
under the dark (for both N+ and N-; quantified via qRT-PCR; Fig. 3C). Intriguingly, for
NobZIP77ko1, no change of NoDGAT2B transcript was detected under either BL or GL
versus the dark (Fig. 3E). Therefore, BL but not GL can specifically induce
NobZIP77-inhibited NoDGAT2B expression.”

49. 6) Suppl. Fig. 8a: Why is there such a little difference in growth between N+ and N-
conditions?

Starting from an OD of 4 under N replete conditions the cells reach an OD750 of around 18
after 11 days, while they reach something like 15 under N- conditions – really not much regarding
the massive importance of nitrogen for cell growth. Also: Measuring an OD at 750 could include
cell debris as well as bacteria, too, easily introduced and propagating during such a long
experiment, which may bias the interpretation of the growth curve. By the way: Measured ODs
about 1 generally have to be treated with care due to the Lambert-Beer law, but ODs larger than
10 are really not trustful anymore. I strongly recommend showing these growth experiments by
really calculating the cell number, either using a growth chamber or a Coulter counter.

Response: As advised, cell growth had been re-measured based on cell density, and the results illustrated in the revised Fig. S16 (as shown in Item #28).

Moreover, these results support our original conclusion that growth rate is not interfered by light wavelength or *NobZIP77* overexpression (Line 264-267).

50. 7) *Nature Comm* limits references to 70 max, but this limit is not at all reached and there are several passages, where more references would be beneficial and actually needed. E.g., line 46: “In oleaginous microalgae, oil accumulation is typically induced by environmental stresses, such as nutrient deprivation, high light or heat 3, 5”, more than the relatively old two references should be provided. Similarly, L50: “For example, nitrogen depletion (N-) has been intensively studied and practically used for triggering microalgal TAG accumulation 6”, many more works have dealt with this issue in various microalgae besides *Neochloris oleabundans*, the species the only reference provided deals with.

Response: As advised, we have inserted five additional references, as listed below.

“In oleaginous microalgae, oil accumulation is typically induced by environmental stresses, such as nutrient deprivation, high light or heat¹⁹⁻²¹.” (Line 42-43).

“For example, nitrogen depletion (N-) has been intensively studied and practically used for triggering microalgal TAG accumulation^{7,8}.” (Line 45-47).

51. 8) *Legend Fig 1.:* Is it correct to explain FPKM as “absolute abundance of transcript”? I’d rather call it the normalized abundance of transcript (on gene length and total reads).

Response: Revised as advised in Line 357.

52. 9) *Fig. S1:* How has the protein abundance been quantified? I could not find any methodological explanation here.

Response: As advised, we have inserted statement into Fig. S1A legend: “The protein data was also from our past study¹⁸. Fold change was calculated as (LFQ intensity (N-) - LFQ intensity (N+)). LFQ, label-free relative quantification.” Please also refer to our response to **Item #30** above.

53. 10) L83: “Totally 32 TFs respond to N-, with eleven upregulated and 14 downregulated.” For me, this sums up to 25?

Response: As advised, we have clarified this point, by changing this statement to: “Altogether, 32 TFs respond to N-, with eleven up-regulated and fourteen down-regulated (seven of them showing distinct trends of regulation among time points).”

54. 11) Fig.1a: What about g8021: From the transcription profile it showed by far the strongest
response and is actually the only factor clearly up-regulated under N-. Any ideas about its
concrete function?

**Response:** The transcription factor g8021 was clearly up-regulated under N-, but it was
predicted to have only one lipid-related target gene (glycerol-3-phosphate dehydrogenase
mitochondrial precursor), which does not seem to be directly implicated in TAG synthesis.
Therefore, we are exploring this gene's function by genetic engineering in a separate study.

55. 12) Suppl. Fig. 5, Which of these genes is NoDGAT2B? Not possible to identify based on the
given gene names.

**Response:** Actually, NoDGAT2B transcript level was shown in Fig. 2A. Here we used Fig. S7
to show the other six NobZIP77-regulated LRGs. As advised, we have now specified the gene
names. The revised Fig. S7 is shown in Item #37.

56. 13) L125: "Among these LRGs, NoDGAT2B (Genbank ID KX867957), a novel member of the
983 N. oceanica type-2 DGAT family 8, 9, exhibited 127.5% higher transcript level in NobZIP77i-2
while 26.0% lower transcript level in NobZIP77o-2 (compared to EV; Fig. 2A), suggesting its
repression by NobZIP77." On which of the depicted time points do these % numbers refer?

**Response:** In fact, here we showed an average fold-change between EV and
NobZIP77-transgenic lines using all time points (i.e., 0h, 6h, 24h and 48h after N- induction).
As advised, we have now revised the sentence as: "The NoDGAT2B transcript is 127.5%
higher in NobZIP77i-2 while 26.0% lower in NobZIP77o-2 (average fold-change over 0 h, 6 h,
24 h and 48 h under N-; versus EV; Fig. 2A), which suggests its repression by NobZIP77."
(Line 141-143).

57. 14) Legend Fig. 2C. Please explain somewhere that "2B" stands for "NoDGAT2B" promoter

**Response:** As advised, we have now replaced the labels of "2B" with "NoDGAT2B" in this
panel. We have also checked across all figures (and their legends) to ensure proper
labels/explanation.

58. 15) Paragraph NobZIP77 as a blue-light sensor that releases the NoDGAT2B promoter under
blue light. I absolutely agree that the sequence and domain features support NobZIP77 to be an
aureochrome. I just want to add that also the inverse order of sensor and effector domain, with the
sensor domain (LOV) at the C-terminus and the effector domain (bZip) rather at the N-terminus
strongly support this TF to be an aureochrome. Such an arrangement is strongly typical for any
photoreceptor except the aureochrome.

**Response:** We appreciate this reviewer for the valuable suggestions! As advised, below was
inserted: "Moreover, its characteristic order of sensor and effector domain, with the sensor
domain (LOV) at the C-terminus and the effector domain (bZIP) at the N-terminus, strongly

suggests NobZIP77 as an aureochrome²².” (Line 205-207).

59. 16) Fig. S7C. I think there is an error in the figure legend: “Red and blue frames indicate
bZIP and LOV domains, respectively. Heptad leucine residues of bZIP domains are in blue, while
the eleven conserved amino acid residues that are associated with flavin binding are in red.” It
has to be “BLUE and RED frames indicate bZIP and LOV domains, respectively....”

**Response: Corrected as advised. We apologize for this typo.**

60. 17) Fig.4a: Although based on the fluorescence figure I also consider it likely that NobZIP77
is located in the nucleus, a proof is lacking here. Please also stain your GFP cell lines with DAPI
or Hoechst and then check in the overlay the match of NobZIP77-GFP with Hoechst/DAPI
fluorescence.

**Response: Please refer to our responses to Item #12 and #26 above.**

61. 18) Regarding the very small error bars in Fig. 5 and Fig. S8 for the lipid measurements, it is
stated that the values are means of three replicates: Are these lipid measurements from the same
culture, where three aliquots have been analyzed in GC-MS measurements, or had there been
cultured three independent cultures for a period of 11 days from where the lipid analyses had
been performed respectively? For a measurement of lipid content, no visible error bars are
somehow not to be expected.

**Response: In fact, we cultured three independent samples for a period of 11 days from where
the lipid analyses had been performed by Raman spectrum-based quantification,
respectively. It should be emphasized that the Raman spectrum-based quantification could
provide us with considerably accurate results for the measurement of lipid content, which
could result in a small error bar.**

62. 19) L331: “On the other hand, aureochromes, although known to regulate
photomorphogenesis, sexual cycle and biological rhythm, have not been implicated in the
synthesis of carbon-storage compounds.”

Agree, nothing is published yet, but please see Jungandreas et al. (2014, PLoS One) and
discussion there about the strong metabolic shift upon red/blue light transitions in diatoms and
possible signaling triggers.

**Response: We appreciate this thoughtful suggestion. As advised, we have now revised the
sentence as “On the other hand, aureochromes, although known to regulate
photomorphogenesis, sexual cycle and biological rhythm²², have not been implicated in the
synthesis of carbon-storage compounds (although strong metabolic shift takes place upon
red/blue light transitions in diatoms⁴¹).” (Line 345-348).**

63. 20) Methods: A detailed section is dedicated to growth of Wt and NobZIP77-KO cultivation
experiments. However, a very large part of the manuscript describes data obtained with silencing

*and overexpressing lines (NobZIP77 and NoDGAT2B). Please provide cultivation conditions for*
*these strains, too.*

**Response:** As advised, to provide detailed description of cultivation conditions for the
**silencing and overexpression lines, we have inserted the following into Methods (Line**
**433-439):**

**“To characterize the *NobZIP77* and *NoDGAT2B* transgenic lines, mid-logarithmic phase**
**algal cells (OD₇₅₀ of 2.6) were collected for validating successful transformants via PCR**
**amplification and Sanger sequencing (Table S2). Positive lines were then cultured for further**
**verification of transcript change. For phenotyping, the transgenic lines (plus the controls,**
**e.g., WT transformed with empty vector or EV) were grown to OD₇₅₀ of 4.5±0.5 for five**
**days, and then the content and FA profiles of both TAG and total lipids were tracked at 0 h,**
**24 h, 48 h, 72 h and 96 h under N-.”**

**Reference cited in this response letter:**

- 1. Ajjawi, I. et al. Lipid production in *Nannochloropsis gaditana* is doubled by decreasing expression of a
single transcriptional regulator. *Nat Biotechnol* **35**, 647-652 (2017).
- 2. Ding, W. et al. Melatonin: A multifunctional molecule that triggers defense responses against high light and
nitrogen starvation stress in *Haematococcus pluvialis*. *J Agr Food Chem* **66**, 7701-7711 (2018).
- 3. Hinojosa-Vidal, E. et al. Characterization of the responses to saline stress in the symbiotic green microalga
*Trebouxia* sp TR9. *Planta* **248**, 1473-1486 (2018).
- 4. Sun, X. et al. Effect of nitrogen-starvation, light intensity and iron on triacylglyceride/carbohydrate
production and fatty acid profile of *Neochloris oleoabundans* HK-129 by a two-stage process. *Bioresource*
*Technol* **155**, 204-212 (2014).
- 5. Goncalves, E.C. et al. Nitrogen starvation-induced accumulation of triacylglycerol in the green algae:
evidence for a role for ROC40, a transcription factor involved in circadian rhythm. *Plant J* **85**, 743-757
(2016).
- 6. Arora, N., Pienkos, P.T., Pruthi, V., Poluri, K.M. & Guarnieri, M.T. Leveraging algal omics to reveal
potential targets for augmenting TAG accumulation. *Biotechnol Adv* **36**, 1274-1292 (2018).
- 7. Jungandreas, A. et al. The acclimation of *Phaeodactylum tricorutum* to blue and red light does not
influence the photosynthetic light reaction but strongly disturbs the carbon allocation pattern. *Plos One* **9**,
e99727 (2014).
- 8. Hu, J.Q. et al. Genome-wide identification of transcription factors and transcription-factor binding sites in
oleaginous microalgae *Nannochloropsis*. *Sci Rep* **4**, 5454 (2014).
- 9. Xin, Y. et al. Producing designer oils in industrial microalgae by rational modulation of co-evolving type-2
diacylglycerol acyltransferases. *Mol Plant* **10**, 1523-1539 (2017).
- 10. Xin, Y. et al. Biosynthesis of triacylglycerol molecules with a tailored PUFA profile in industrial microalgae.
*Mol Plant* **12**, 474-488 (2019).
- 11. Zienkiewicz, K. et al. *Nannochloropsis*, a rich source of diacylglycerol acyltransferases for engineering of
triacylglycerol content in different hosts. *Biotechnol Biofuels* **10** (2017).
- 12. Goncalves, E.C., Wilkie, A.C., Kirst, M. & Rathinasabapathi, B. Metabolic regulation of triacylglycerol

- accumulation in the green algae: identification of potential targets for engineering to improve oil yield.
*Plant Biotechnol J* **14**, 1649-1660 (2016).
- 13. He, Y.H. et al. Label-free, simultaneous quantification of starch, protein and triacylglycerol in single
microalgal cells. *Biotechnol Biofuels* **10**,107388 (2017).
- 14. Wang, T.T. et al. Quantitative dynamics of triacylglycerol accumulation in microalgae populations at
single-cell resolution revealed by raman microspectroscopy. *Biotechnol Biofuels* **7**, 58-70 (2014).
- 15. Ritchie, R.J. Consistent sets of spectrophotometric chlorophyll equations for acetone, methanol and ethanol
solvents. *Photosynth Res* **89**, 27-41 (2006).
- 16. Ueno, Y., Aikawa, S., Kondo, A. & Akimoto, S. Adaptation of light-harvesting functions of unicellular
green algae to different light qualities. *Photosynth Res* **139**, 145-154 (2019).
- 17. Li, J. et al. Choreography of transcriptomes and lipidomes of *Nannochloropsis* reveals the mechanisms of
oil synthesis in microalgae. *Plant Cell* **26**, 1645-1665 (2014).
- 18. You, W.X. et al. Integration of proteome and transcriptome refines key molecular processes underlying oil
production in *Nannochloropsis oceanica*. *Biotechnol Biofuels* **13**, 109 (2020).
- 19. Hu, Q. et al. Microalgal triacylglycerols as feedstocks for biofuel production: perspectives and advances.
*Plant J* **54**, 621-639 (2008).
- 20. Rodolfi, L. et al. Microalgae for oil: strain selection, induction of lipid synthesis and outdoor mass
cultivation in a low-cost photobioreactor. *Biotechnol Bioeng* **102**, 100-112 (2009).
- 21. Zienkiewicz, K., Du, Z.Y., Ma, W., Vollheyde, K. & Benning, C. Stress-induced neutral lipid biosynthesis
in microalgae - Molecular, cellular and physiological insights. *Bba-Mol Cell Biol L* **1861**, 1269-1281
(2016).
- 22. Matiiv, A.B. & Chekunova, E.M. Aureochromes - blue light receptors. *Biochemistry* **83**, 662-673 (2018).

If you have any additional suggestions please do not hesitate to contact me via phone (Office:
86-532-80662651; Mobile: 86-15063049248) or email.

With warmest regards,

Jian XU, Ph.D

Professor and Director, Single-Cell Center

Qingdao Institute of BioEnergy and Bioprocess Technology, Chinese Academy of Sciences

Postal: 189 Songling Rd, Qingdao, Shandong, China 266101

Email: xujian@qibebt.ac.cn Tel/Fax: 86-532-80662651/2654

Reviewers' Comments:

Reviewer #1:

Remarks to the Author:

The data is improved significantly, in that they now have some data that the cloned TF actually binds to DNA related to promoters. They have also now cited other papers that have similar results on induction to TG synthesis in algae. The paper is now acceptable at the believable data level, but I am not sure this paper is a particular break through that the field has been waiting for.

=====

Reviewer #2 previous comments:

This manuscript described a transcription factor, NobZIP77 (g77), as a blue light sensor and a regulator of NoDGAT2B, leading to improved TAG production. There were merits in the manuscript, including description of the new TF and its possible functions for lipid metabolism. However, its direct interaction with the promoter of NoDGAT2B is overall weak: the in vitro EMSA assay is incomplete, and no in vivo evidence (such as ChIP assays).

- They did not add any additional in vivo data for NobZIP77 binding to NoDGAT2B, although they did show that fusion with the bZIP77 and GFP likely localize to the nucleus

There were conflicts between its expression and that of the target, leaving an open question for different targets (such as lipases listed in Table S1).

- This remains unresolved, they claim BZIP77 is a transcriptional repressor not an activator, but that doesn't really explain the lack of correlation between the two, they should be inversely correlated and they are not

Moreover, its localization as a TF in the nucleus is not complete.

- This is now done by GFP fusion, not great but it does suggest nuclear localization

Table S1: The data presented in this table look interesting, but lack practical information, such as FPKM for both TFs and targets (LRGs), to show how much TF expression was induced and how target expression is affected.

In addition, targets may have different (and opposite) impacts on lipid metabolism. For example, targets of g77 include DGATs and lipases, which warrants explanation.

- Yes, there is actually a poor correlation between TAG accumulation and BZIP77 Ko or over expression with only a 30% change in TAG levels across both of these

Figure 1B: If g77 is a blue light sensor, the structure should be drawn with and without its chromophore.

- They did this

Figures 2A and 2D: Expression of NoDGAT2B and TAG analyses were done in different time scale. Why?

- They added a new figure, but the TAG differences just are not that great

Figure 2B: The reporter assay could have been better if performed in Nannochloropsis.

- They made a new figure and it does look better

Figure 2C: In my understanding, the super shift should be caused by adding an antibody against g77. I think that the different sizes of shifted bands may represent formation of oligomers or dramatic conformational changes of g77.

- They made new figure with increasing protein (BZIP77), but these still look different than what

reviewer 2 was expecting. The gel shift assays do not look like other published TF binding assays

However, effects of unlabeled DNA are weak (even with 10 times more of it, as shown lanes 2,3,4), which may suggest non-specific binding of the promoter sequence of NoDGAT2B. This should be considered critically, because this data is the only evidence for direct interaction between g77 and NoDGAT2B promoter. It is also recommended to perform the same EMSA with non-specific cold DNA (e.g., same amount of unrelated genomic DNA fragments) and compare to that of specific cold probe.

- They added a new figure, but not with the exact controls requested here, but did use other DNA fragments (different promoters) for the binding assays

Figure 2D: In addition to the % TAG (as shown here), it would be interesting to see TAG contents, since many DGATs are known to increased TAG contents and yields.

It is expected to have opposite effects between NoDGAT2B ox and kd on TAG contents, which was shown only on time 0. Effects of ox disappeared at later times, which requires explanation. One suggestion may be that NoDGAT2B may not be related to N-, but with N+ conditions.

- They added a figure for this, but the response is really pretty poor if the BZIP77 was truly a gene regulator, a 30% change is just not what is expected for a TF regulator, this scale should be an order of magnitude not 30%

Figure 3A-C: Quantitation of DNA binding should be measured with the shifted band, rather than the saturated signals of the unbound. Actually, shifted bands appeared faster under the dark. Another question is about the same EMSA assay with the white light (rather than dark), since all other experiments were done with white lights as a control.

- They added figure showing different binding activity of blue light, white light or dark

Figure 3C: Were EMSA assay repeated to reveal statistic significance? Binding with dark and blue light doesn't look significantly different. It would also be better to see blue light effects with N+ conditions.

- I did not see any error bars or repeated assays

Figures 3E,F: It is good to see employing Raman technology for measuring relative TAG contents. Can a standard curve be created using cells with known TAG contents?

It is also interesting to see effects of the green light: It is generally thought that green light is not used for photosynthesis (and I don't recall any green light receptors), and can be assumed no effects on lipid metabolism; however, it has effects under N- condition. How could it be?

- They added data and comments on green light

Another point to note: Blue light still has positive effects in g77 KO line under N- condition, which leaves a question for additional blue light sensors.

Figure 4A: Nuclear localization for g77 should be confirmed by nuclear staining using DAPI.

- They did this

Localization of NoDGAT2B is not clear. It should be co-localized with sDer1:RFP fusion. If it is associated with lipid droplets, it can also be co-stained with Bodipy or Nile red.

- They added this and it does co-localize

Figure 5: Experiments were done with different time schedules, and it is hard to directly compare data directly. Time values are misplaced in Figure 5A.

- They added several time points for this, but it makes it even less conclusive, as now the response in the KO and knocking are not very different

Figure 6: Explain why blue light induction was done for KO mutants only. To claim its effect, the same treatment should be applied to WT.

* The letters (B and C) are misplaced.

- They fixed this

Figure S2: There are multiple bZIPs in algae and many more in plants. Explain how the representative bZIPs were chosen. Are they all related to blue light sensing?

- They added some discussion but just a sentence saying that g77 was chosen because of DNA binding attributes

Figure S4C showed only TAG per total lipids. It is recommended to show a graph for total lipid contents.

- They added this, not that different in any of the mutants

In addition, expression patterns of g77 is more or less consistent as shown in Figure S4A, but its impact on TAG content (S4C) and TAG yields (S4D) are different (mostly at 0 and 24 hrs). Explain why.

*The gene name was initially defined as g77, but NobZIP77 was also used later. It would be better to use consistent names.

- They said the g77 is before knowing what it was BZIP77 after they confirmed what it was ... seems odd way to explain that

Overall English writing needs improvement by professional writers, since many statements were not clear and can be misleading; for example, in Abstract: "a blue-light-oil-induction strategy that exposes NobZIP77-knockout *N. oceanica* to white light and then to blue light doubles peak TAG productivity" What does it mean?

It is not clear (and actually confusing) how the blue and white lights are used and how they are related, leaving a room for confusion and misunderstanding.

- They changed that sentence, but it is still just as confusing

In my version of the manuscript, references 1-38 are missing.

- They show those reference are there

My overall impression is that the authors did add several of the experiments that reviewer 2 asked for. This new data did resolve several of reviewer 2 questions on BZIP77 nuclear localization, improved the BZIP77 promoter binding assays, and showed that DGAT2B localizes to lipid bodies. On these counts, I think reviewer 2 would be ok with the revisions. The data that BZIP is actually binding to the DGAT2B gene and regulating DGAT2B gene expression remains pretty tenuous, and the discussion on this is both speculative and at times confusing, perhaps because the data is just not all that consistent with the hypothesis. This is especially true in both the expression data for the BZIP77 gene and the DGAT2B expressing (they should be inversely correlated if BZIP77 is actually a DGAT2B DNA binding protein that represses transcription), and in the TAG accumulation data. It is hard to believe that this BZIP77 DNA protein (here called a TF), is really a key blue light regulator of TAG accumulation when there is only a 30% difference between the over-expression of the BZIP77 gene, and a mutant in DGAT2B that is supposedly the BZIP77 binding site? On that front I think reviewer 2 would still have objections to the conclusion that this BZIP77 protein is a DNA repressor of DGAT2B gene expression as a key blue light regulatory of TAG accumulation.

Reviewer #3:

Remarks to the Author:

The authors performed quite a number of additional experiments, which clearly improved this interesting manuscript and solidify the proposed mechanism of an aureochrome regulating TAG accumulation.

There are still a few issues I'd like to be addressed.

1. Fig 1c,d: Please specify which of the symbols refers to the TAG yield and which to the TAG productivity.

2. L109: "Furthermore, in NobZIP77-knockdown lines, under N+, TAG-associated C16:1 is 59.0-106.4% higher..."

I appreciate that the authors present an inset in Fig. S4e, enlarging the values of the fatty acids with minor contents. Please include also here the data for 16:1, to see visually what is claimed above.

3. L118 and the following: The authors here present the data of the complementing lines, which they newly collected for the new manuscript. Just from a stylistic point of view: Instead of starting directly with explaining the complemented lines, created in the KO background, I would first start introducing the KO lines, stating that they behave similar as the silencing lines, and then move to the complementing line. Otherwise, one wonders where the KO lines are suddenly coming out of the blue.

4. L 125: "Specifically, versus wild type (WT) and under N-, TAG content was 51.0% and 178.1% higher at 0 h and 24 h for NobZIP77ko-1, while 160.0% and 49.5% higher at 0 h and 24 h for NobZIP77ko-2.

There are errors in the order of the % numbers. Both KO mutants basically behave the same, at 0h and 24 h, so please correct.

5. Fig.2 E,F,G: If one looks at the TAG of the WT, it varies at T0 between 2.5 in Wt (E) and 7 (G) and at T24N- between ~12 and 14. This would make many of the observed significances between the WT and various mutants non-significant, if one would not calculate statistics based on the values for the Wt in the subpanels, but e.g. compare the TAG contents of the OE lines in (E) with the WT in (G) at T0. Therefore, I had previously asked about how biological replicates were obtained. Although there is variability, in lab conditions things like TAG (I don't talk about absolute values of transcriptomics, which is another story) should be relatively stable if algae are kept under identical conditions. What I suggest here is that the authors calculate the mean of the WT of all panels (so including all replicates) and then check if the mutants are significantly different.

6. Fig. S11: I don't understand why in B (TAG per total lipids), starting from 48 h, there is no significant difference between EV and Bi, while in C (%TAG per total lipids) suddenly there is. Please explain.

7. Growth experiments Fig. S4, S6 and Fig S13. When one only looks at the Wt and the EV control, one can see major differences in all three growth experiments. However, in all three experiments, especially S6 and S13, standard deviation is really small. Again, the authors do not specifically mention what kind of replicates they have done (despite my request in the previous reviewing process) and whether these are true biological replicates. Based on the minimum standard deviation, but the huge differences between Wt e.g. in S6 and S13, I am pretty convinced that the authors actually used one stock cultures, split it into three aliquots and immediately started the growth analysis. In lab growth analyses we here have some kind of a grey zone, as we always start from the same initial culture. However, while such an approach gives very nice results in terms of standard error for all kind of subsequent analyses (growth, gene expression, TAG,...) it is not really what one considers as independent biological replicates. If, for instance, during such an experiment of 10 days a

lamp for the growth light gets out of order at day 6, or the temperature in the culture room would change also at day 6, the authors would not notice this, as all their three replicates are there at the same time, taken from the same starting culture, and hence error bars would still be extremely small. Instead, statistical tests might indicate some miraculous change in physiology at day 6 to 7 highly significant.

For the data presented here, the authors would normally need to take the mean of Fig. S6 and Fig. S13 for the Wt, and then the whole picture would look quite different. I believe that the results presented here reflect the true differences between Wt and the mutants, and therefore I do not request additional experiments with true independent biological replicates, but I would like to see that the authors specify in their methods how exactly they obtained their "replicates" for their analyses. And while the authors in the "Reporting summary" ticked "confirmed" for "A statement on whether measurements were taken from distinct samples or whether the same sample was measured repeatedly", I don't find any of such information in the manuscript, only the indication, "experiments were performed in n replicates". This is still a serious point for me I this time would like to be addressed properly.

8. L213: "As for TFA content, no changes were found in either double-knockout lines (Fig. S15E)."

Not true. In contrast to the results with the NoDGAT2B-KO (Fig. S13) and the NobZIP77 KO (Fig. S6), in the double KO there is a significant difference (Fig. S15). Moreover, and this makes me really skeptical, the TFA content is only ~ 20 % in Fig. S15 compared to Fig. S13+FigS6. Please explain.

9. L250: The 11.7 % less TAG under BL in the NobZIP77ko-1 line: Is this significant compared to the Wt?

10. I like the model of the authors how BL could lead to the increase of lipids under N-. The lead idea here is that under N- there is less chlorophyll shielding the nucleus and hence more blue light reaches it and activates the aureochrome LOV domain. However, there is no difference in chlorophyll amount even after 12 h N- (Fig. 4 D). Unfortunately, the authors only provide normalized data for NoDGAT2B in their manuscript for the Wt. If I interpret the FPKM values in Table S1 correctly, there is already a pronounced difference in noDGAT2B transcription between N- and N+ conditions at much earlier time points, which would not fit to the proposed hypothesis. Can the authors explain this?

11. There is a recent study demonstrating that knocking out an aureochrome photoreceptor in diatoms has a massive effect on the general transcriptome, and here especially also on the lipid metabolism (Mann et al. 2020, iScience). This links those results to the ones presented here and shows that aureochromes may fulfill similar functions in triggering the metabolism in distinct groups within the stramenopiles. The corresponding article should be shortly discussed within the framework of the presented results, and the statement L369: "On the other hand, aureochromes, although known to regulate photomorphogenesis, sexual cycle and biological rhythm, have not been implicated in the synthesis of carbon-storage compounds" should be modified.

Bernard Lepetit

Reviewers' comments:

Reviewer #1 (Remarks to the Author):

1. *The data is improved significantly, in that they now have some data that the cloned TF actually binds to DNA related to promoters. They have also now cited other papers that have similar results on induction to TG synthesis in algae. The paper is now acceptable at the believable data level, but I am not sure this paper is a particular break through that the field has been waiting for.*

Response: We have performed all the experiments requested by the reviewers and editors during the past two rounds of reviews. We appreciate your help improving our manuscript.

Reviewer #2 previous comments:

2. *This manuscript described a transcription factor, NobZIP77 (g77), as a blue light sensor and a regulator of NoDGAT2B, leading to improved TAG production. There were merits in the manuscript, including description of the new TF and its possible functions for lipid metabolism. However, its direct interaction with the promoter of NoDGAT2B is overall weak: the in vitro EMSA assay is incomplete, and no in vivo evidence (such as ChIP assays).
- They did not add any additional in vivo data for NobBIP77 binding to NoDGAT2B, although they did show that fusion with the bZIP77 and GFP likely localize to the nucleus*

Response: As suggested above, we have now tested the *in vivo* binding between NobZIP77 and the NoDGAT2B promoter (pNoDGAT2B) by chromatin immunoprecipitation (ChIP) qPCR assays in *N. oceanica*. These new experimental results did verify the direct binding of NobZIP77 to the ACGT-harboring pNoDGAT2B sequence *in vivo* (Line 163-172).

Legend of the inserted figure panel for ChIP-qPCR reads: “Fig. 2 *NoDGAT2B* is a downstream target of NobZIP77 which inhibits *NoDGAT2B* transcription by binding to the promoter. (C) ChIP-qPCR analysis of NobZIP77 binding to the promoter of *NoDGAT2B* (pNoDGAT2B) *in vivo*. In pNoDGAT2B, the region used for ChIP-qPCR is marked (transverse line, top panel). The bZIP binding motif is indicated as arrowheads. The numbers in fragment indicate positions of the nucleotides at the 5' or 3' end of the fragment relative to the translation start site. β -actin was used as an internal reference gene. The fold change was calculated via the $2^{-\Delta\Delta Ct}$ method between mutant and WT. $\Delta\Delta Ct = (\Delta Ct (Ct_{[ChIP DNA]} - Ct_{[Input DNA]}; mutant) - (\Delta Ct (Ct_{[ChIP$

DNA]-Ct [Input DNA]); WT). WT, *N. oceanica* wild type; 77, *gfp:NobZIP77*-overexpression line.”

The inserted panel is shown below:

Accordingly, below texts were inserted into Results (Line 163-172): “Next, we tested the *in vivo* binding between NobZIP77 and pNoDGAT2B, by chromatin immunoprecipitation (ChIP) qPCR assays in *N. oceanica*. A GFP gene fused immediately downstream of the full-length *NobZIP77* cDNA was transformed into *N. oceanica* and positive expression in transgenic lines (OE1) was confirmed with laser scanning confocal microscopy (Fig. S10). Under N+, OE1 and WT cells (cell density of 2×10^7 cells/ml) were subject to chromatin extraction and immunoprecipitation with anti-GFP antibody (Methods). The ChIP products were analyzed by qPCR and the fold change of pNoDGAT2B amount calculated via $2^{-\Delta\Delta CT}$ (β -actin as internal reference). Notably, the amount of pNoDGAT2B in OE1 is 2.2-fold of that in WT (Fig. 2C), supporting the direct binding of NobZIP77 to the ACGT-harboring pNoDGAT2B sequence *in vivo*.”

Moreover, below paragraphs were inserted into Methods (Line 771-788): “To validate the direct binding between NobZIP77 and pNoDGAT2B, a ChIP-qPCR experiment was performed as described^{57, 58}. Briefly, 400 mL of algal cells from WT and transgenic *N. oceanica* overexpressing GFP- NobZIP77 (OE1, Fig. S10) were harvested and crosslinked with 1% formaldehyde. Then algal cells were ground into fine powder with liquid nitrogen, and thereafter, chromatin was extracted and sheared into 200- to 2000-bp fragments by sonication. Subsequently, the crude chromatin lysates served as the input and were incubated overnight at 4 °C with or without anti-GFP monoclonal antibody (Abmart, Shanghai, China). The

immunoprecipitated complexes were then pre-cleared with protein-A agarose beads (Cell Signal Technology, Danvers, MA). Then the precipitated DNA was extracted with phenol and chloroform, and then purified with ethanol.

For ChIP-qPCR experiments, immunoprecipitated DNA and control DNA, with three biological replicates for each, were mixed with primers and LightCycler 480 SYBR Green I master mix (Roche), respectively. Products were amplified and fluorescent signals acquired with a LightCycler 480 detection system. The fold enrichment of target in the immunoprecipitated DNA relative to the control was calculated from an average of three replicate qPCRs. Relative enrichment of the target region of NoDGAT2B was normalized against β -actin. Sequences of the primers are listed in Table S2.”

For additional details, please refer to our response to Comment #1 above.

3. *There were conflicts between its expression and that of the target, leaving an open question for different targets (such as lipases listed in Table S1).*
 - *This remains unresolved, they claim BZIP77 is a transcriptional repressor not an activator, but that doesn't really explain the lack of correlation between the two, they should be inversely correlated and they are not*

Response: For “correlation between the expression and that of the target”, as well as that in terms of phenotypes between NobZIP77 mutants and NoDGAT2B mutants, we have now summarized all the experimental evidence in a table below, for clarity. In each of the experiments, the correlation is clear, and supports our model that NobZIP77 is a transcriptional repressor of NoDGAT2B.

			Expected observation	Actual observation (Max)	Correlation that supports our model?
in vivo	Overexpression	NobZIP77	NoDGAT2B transcript level ↓	26.0% ↓ (Fig. S8)	Yes
			TAG content ↓	TAG: 35.2% ↓ (Fig. 1C)	
		NoDGAT2B	TAG content ↑	TAG: 83.9% ↑ (Fig. 2F)	
	Knock down	NobZIP77	NoDGAT2B transcript level ↑	127.5% ↑ (Fig. S8)	Yes

		TAG content ↑	TAG: 196.7% ↑ (Fig. 1C)	
	NoDGAT2B	TAG content ↓	TAG: 64.3% ↓ (Fig. 2F)	
Knock out	NobZIP77	NoDGAT2B transcript level↑	128.9% ↑ (Fig. 2A)	Yes
		TAG content ↑	TAG: 178.1% ↑ (Fig. 1D)	
	NoDGAT2B	TAG content ↓	TAG: 82.6% ↓ (Fig. 2G)	
GFP reporter system in NobZIP77 - pNoDGAT2B transformed lines (v.s. pNoDGAT2B transformed line)		GFP transcript level↓	69.7% ↓ (Fig. 2C)	Yes
		Fluorescence intensity of GFP ↓	55.8% ↓ (Fig. 2D)	
ex vivo	GUS activity in A. thaliana reporter (pNoDGAT2B)- effector (NobZIP77) system	GUS activity ↓	31.9% ↓ (Fig. 2B)	Yes

Notably, we have now inserted the *ex vivo* evidence for *NobZIP77* as a transcriptional repressor of *NoDGAT2B* in the original submission (Line 158-169 in the first version submission “NCOMMS-20-28862A”) back into the current version (Line 173-185):

“Furthermore, to probe the functional consequence of the binding between *NobZIP77* and *pNoDGAT2B*, transient *NoDGAT2B* transcription level was measured via a reporter-effector system in *Arabidopsis* mesophyll protoplast (TEAMP, Methods) ²⁶. *Arabidopsis* mesophyll protoplasts were employed in transient analysis of promoter activity *in vivo*, and β-glucuronidase (GUS) used as reporter for interaction with effector. Specifically, the 900bp *pNoDGAT2B* was linked to the GUS reporter gene to create the reporter construct (*pNoDGAT2B* promoter-:*GUS*), and the full-length *NobZIP77* coding sequence (CDS) was ligated downstream of the 35S promoter to generate the effector construct (*CaMV35S* promoter-:*NobZIP77*). The two constructs were co-transfected into *A. thaliana* leaf protoplasts and the GUS activity measured

(Methods). These *ex vivo* experiments revealed that relative GUS activity of the reporter-effector co-transfected protoplasts was 31.9% lower than the protoplasts transfected with the reporter construct alone, indicating an inhibitory effect on *NoDGAT2B* transcription due to the binding of NobZIP77 to p*NoDGAT2B* (Fig. 2D).”

Fig. 2 *NoDGAT2B* is a downstream target of NobZIP77 which inhibits *NoDGAT2B* transcription by binding to the promoter. (D) The enzymatic activities of glucuronidase (GUS) between NobZIP77-*NoDGAT2B* co-transfected or *NoDGAT2B* transfected *Arabidopsis* leaf protoplasts. The expression level of the luciferase (LUC) was used as a control. Ctr, p*NoDGAT2B* transfected *Arabidopsis* leaf protoplasts; 77-2B, NobZIP77-p*NoDGAT2B* co-transfected *Arabidopsis* leaf protoplasts.

For additional evidence that validates NobZIP77 as a transcriptional repressor of *NoDGAT2B* and that supports the BZIP77-*NoDGAT2B*-TAG pathway, please refer to our response to Comment #1 above.

4. Moreover, its localization as a TF in the nucleus is not complete.

- This is now done by GFP fusion, not great but it does suggest nuclear localization

Response: For the nuclear localization of NobZIP77, we have now further confirmed it by DAPI staining (Fig. 4A, as shown below). Specifically, below was inserted into Results (Line 291-293): “The NobZIP77:gfp fusion protein is colocalized with the DAPI-stained nuclei, indicating nucleus targeting (Fig. 4A).”

Fig. 4. A mechanistic model of N- induced TAG synthesis via the BL-NobZIP77-*NoDGAT2B* pathway in *N. oceanica*. (A) Subcellular localization of

NobZIP77. GFP, green fluorescent protein; TML, transmission light; DAPI, 4',6-diamidino-2- phenylindole; PAF, plastid autofluorescence; scale bar, 2 μ m.

5. *Table S1: The data presented in this table look interesting, but lack practical information, such as FPKM for both TFs and targets (LRGs), to show how much TF expression was induced and how target expression is affected. In addition, targets may have different (and opposite) impacts on lipid metabolism. For example, targets of g77 include DGATs and lipases, which warrants explanation.*

- Yes, there is actually a poor correlation between TAG accumulation and BZIP77 Ko or over expression with only a 30% change in TAG levels across both of these

Response: As for “30% change in TAG level”, actually, for TAG content, in NobZIP77i-1 and NobZIP77i-2, it is 53.3% and 196.7% higher than EV respectively (0 h under N-; Fig. 1C and Fig. S4C). Moreover, under N-, TAG content is 178.1% and 51.0% higher at 0 h and 24 h for NobZIP77ko-1, while 160.0% and 49.5% higher at 0 h and 24 h for NobZIP77ko-2 (Fig. 1D and Fig. S6B). Therefore, NobZIP77 exerts a very significant (instead of MINOR) effect on the regulation of TAG accumulation at 0 h and 24 h N-. This situation, i.e., significant change in TAG

content in some, but not all, of the time points on TF engineering, was also observed in other microalgal transcription factor engineering studies, as shown below:

Microalgae	TF	TF Engineering strategy	Time points sampled	Time points (TAG-content change)	Fold change of TAG content	Reference
Chlamydomonas	NRR1	Targeted insertion	0h, 24h, 48h after N-	48h after N-	52% decrease	(Boyle et al. 2012)
Chlamydomonas	crbZIP1	Knockdown	0h, 12h after tunicamycin treatment	12h	5.8-9.4 fold increase	(Yamaoka et al. 2019)
Chlamydomonas	PSR1	Random insertion	1d, 2d, 3d, 8d after P-	8d	90% decrease	(Ngan et al. 2015)
Nannochloropsis	NsbZIP1	Overexpression	4d, 8d, 12d under N+ and N-	12d	33% higher under N+ and 88% higher under N-	(Kwon et al. 2018)
Nannochloropsis	NobZIP77	Overexpression, knockdown, and knockout	0h, 24h, 48h, 72h, and 96h under N-	0h and 24h under N- (see above text for details)	e.g., 196.7% higher (see above text)	This work

In our previous round of revision, we have already provided explanation in Results (Line 80-86) for the possibility that targets of NobZIP7 include *NoDGAT2B* and perhaps also lipases. The text (Line 80-86) reads: “..., scaffold00007.g77 (or g77, Genbank ID MT273120), which is down-regulated by N-, targets seven such genes that include a diacylglycerol acyltransferase (DGAT) and a few additional enzymes that are related to lipid metabolism (Fig. S1A, Table S1). Notably, transcripts in the LRG of g77 exhibit distinct or even opposite fold-changes, e.g., up-regulation of the lipase gene g10411 yet downregulation of the DGAT gene of g6725 (Table S1). As the activities of lipases and the DGAT are exactly opposite (oil degradation and assembly respectively), these results indicate g77 as a negative regulator of TAG synthesis.” At this stage, we believe NobZIP77 is possibly a master regulator that has a potential role in other metabolic pathways too, but untangling this network is beyond the scope of this study.

6. *Figure 1B: If g77 is a blue light sensor, the structure should be drawn with and without its chromophore.*
- They did this

Response: Thanks for the confirmation.

7. *Figures 2A and 2D: Expression of NoDGAT2B and TAG analyses were done in different time scale. Why?*
- *They added a new figure, but the TAG differences just are not that great*

Response: Like studies that overexpress or knockdown TF, overexpression of DGAT genes might not result in dramatic change in cellular TAG content, particularly considering that *NoDGAT2B* is only one of the eleven putative TAG-synthetic DGATs in *N. oceanica* (we have previously experimentally validated five of them as truly TAG-synthetic: 2A, 2C, 2D, 2J and 2K; (Wang et al. 2014; Li et al. 2014; Xin et al. 2017; Xin et al. 2019)). Moreover, for example, even though *NoDGAT2A* exhibits the strongest TAG-synthetic activity among the *NoDGAT2s* in our recently published pDEP-RACS-based screening of yeast (Wang et al. 2020), overexpression of *NoDGAT2A* in *N. oceanica* led to only 31.0% increase in TAG content at 96h under N- conditions (Xin et al. 2017), which is at an less (or similar) degree of TAG-content changes than the overexpression of *NoDGAT2B* here.

For additional discussion, please refer to our response to Comment #1 above.

8. *Figure 2B: The reporter assay could have been better if performed in Nannochloropsis.*
- *They made a new figure and it does look better*

Response: Thanks for the confirmation.

9. *Figure 2C: In my understanding, the super shift should be caused by adding an antibody against g77. I think that the different sizes of shifted bands may represent formation of oligomers or dramatic conformational changes of g77.*
- *They made new figure with increasing protein (BZIP77), but these still look different than what reviewer 2 was expecting. The gel shift assays do not look like other published TF binding assays*

Response: In fact, Reviewer 2 recommended performing the same EMSA with non-specific cold DNA (quote: “It is also recommended to perform the same EMSA with non-specific cold DNA (e.g., same amount of unrelated genomic DNA fragments) and compare to that of specific cold probe.”) and other DNA fragments (quote: “In addition, bZIPs are usually transcriptional activator, not repressors (Figure 2A). With these possible conflicts, it is recommended to run EMSA with different promoters, such as one of the lipase genes (Table S1), which may have opposite impact on lipid metabolism, compared to DGATs.”). As advised, we had already included the additional data in both Results and Methods.

Firstly, to validate NobZIP77-targeted genes, we have performed an EMSA assay with five NobZIP77-targeted genes, including *g3857* (TAG-lipase), *g2171* (PAP), *g10041* (lipase), *NoDGAT2B* and *g5641* (FAE). The results are now presented in a new supplementary figure of Fig. S11 (shown below).

The legend of Fig. S11 reads: “Fig. S11. EMSA-based validation of NobZIP77-targeted genes. EMSA results revealed that NobZIP77 directly binds to the promoters of *NoDGAT2B*, *g5641* (FAE), *g10411* (lipase), *g2171* (PAP) and *g3857* (TAG-lipase). Unlabelled DNA of the promoters in 100-fold molar excess were treated with the NobZIP77 protein. PAP, Phosphatidic acid phosphohydrolase; FAE, fatty acid elongase; LS, lower shift complex (a 1:1 complex of DNA and the bZIP dimer); SS, super shift complex; FP, free-DNA probe; COP, competitor oligonucleotide primer.”

Secondly, we performed the EMSAs with different promoters, including three predicted target genes [*NoDGAT2B*, *g3857* (TAG-lipase) and *g10041* (lipase)] and the negative controls [*g4867* (hsp70) and *g9477* (actin)]. The results revealed that the recombinant NobZIP77 can bind specifically with promoters of predicted target genes, rather than other promoters, such as *g4867* and *g9477*. Thus, we now provide a new version of Fig. 2B (formerly Fig. 2C), as shown below.

The legend of Fig. 2B now reads: “(B) EMSA validation of specific binding between NobZIP77 and the promoters of its targeted genes (i.e., *NoDGAT2B*, *g3857* and *g10411*). *g3857*, TAG-lipase; *g10041*, lipase; *g4867*, hsp70A; *g9477*, actin. Unlabelled

DNA of the promoters in 100-fold molar excess were treated with the NobZIP77 protein. LS, lower shift complex (a 1:1 complex of DNA and the bZIP dimer); SS, super shift complex; FP, free-DNA probe; COP, competitor oligonucleotide primer.”

Accordingly, in Results, the following was inserted (Line 151-162): “To probe this hypothesis and test whether NobZIP77 directly interacts with the promoter of *NoDGAT2B* (*pNoDGAT2B*; and of additional predicted target genes or PTGs), electrophoretic mobility shift assays (EMSAs) were used to identify specific band shifts of ACGT-harboring promoter probes (the binding sites of bZIP-type TF genes ²⁵). Recombinant maltose-binding protein (MBP) - NobZIP77 was expressed in *Escherichia coli* and purified for EMSA (Methods). ACGT-harboring probes of the

NobZIP77-PTG promoters [g3857 (TAG-lipase), g2171 (PAP), g10041 (lipase), NoDGAT2B and g5641 (FAE)] result in mobility shift, indicating the direct binding between the NobZIP77-PTG promoters and NobZIP77 (Fig. S9). Notably, the degree of mobility shift is greatly reduced by adding unlabeled fragment of NobZIP77-PTG promoter (Fig. S9), yet no binding between NobZIP77 and the negative controls of g4867 (hsp70) and g9477 (actin) was detected (Fig. 2B), which supports the specificity of binding between NobZIP77 and pNoDGAT2B.”

Moreover, in Methods, below was revised (Line 594-601): “As for probes, the promoters of predicted NobZIP77 target genes and the negative controls were amplified with primers labeled with biotin at the 5' end (Table S3). To test the direct bindings between NobZIP77 and above-mentioned promoters, the EMSA assays were conducted via a LightShift[®] Chemiluminescent EMSA Kit (Thermo Fisher, USA).”

As for the comment that “*The gel shift assays do not look like other published TF binding assays*”, the different sizes of shifted bands are due to the binding to the promote DNA of the dimeric and monomeric forms of NobZIP77 (note that Reviewer #2 has already offered this speculation), as was explained by a previous study (Takahashi et al. 2007).

10. However, effects of unlabeled DNA are weak (even with 10 times more of it, as shown lanes 2,3,4), which may suggest non-specific binding of the promoter sequence of NoDGAT2B. This should be considered critically, because this data is the only evidence for direct interaction between g77 and NoDGAT2B promoter. It is also recommended to perform the same EMSA with non-specific cold DNA (e.g., same amount of unrelated genomic DNA fragments) and compare to that of specific cold probe.

- They added a new figure, but not with the exact controls requested here, but did use other DNA fragments (different promoters) for the binding assays

Response: Actually, we had now inserted the additional data for the binding assays, including both the non-specific cold DNA controls (as requested above; Fig. 2B) and other DNA fragments (i.e., various promoters that also include those of the lipase genes; Fig. 2B and Fig. S9; also requested by Reviewer 2 – quote: “*In addition, bZIPs are usually transcriptional activator, not repressors (Figure 2A). With these possible conflicts, it is recommended to run EMSA with different promoters, such as one of the lipase genes (Table S1), which may have opposite impact on lipid metabolism, compared to DGATs.*”). Please refer to our response to Comment #10 above.

As for the comment of “*This should be considered critically, because this data only evidence for direct interaction between g77 and NoDGAT2B promoter*”, in fact, as advised,

we have now performed the ChIP assay to test the specific binding between NobZIP77 and the *NoDGAT2B* promoter *in vivo*, which validated that NobZIP77 directly regulates *NoDGAT2B* transcription by binding to the latter's promoter *in vivo* (Line 163-172). For details, please refer to our response to Comment #3 above.

11. *Figure 2D: In addition to the % TAG (as shown here), it would be interesting to see TAG contents, since many DGATs are known to increased TAG contents and yields. It is expected to have opposite effects between NoDGAT2B ox and kd on TAG contents, which was shown only on time 0. Effects of ox disappeared at later times, which requires explanation. One suggestion may be that NoDGAT2B may not be related to N-, but with N+ conditions.*
- *They added a figure for this, but the response is really pretty poor if the BZIP77 was truly a gene regulator, a 30% change is just not what is expected for a TF regulator, this scale should be an order of magnitude not 30%.*

Response: Please refer to our responses to Comment #6 (and Comment #1) above.

12. *Figure 3A-C: Quantitation of DNA binding should be measured with the shifted band, rather than the saturated signals of the unbound. Actually, shifted bands appeared faster under the dark.*
- Another question is about the same EMSA assay with the white light (rather than dark), since all other experiments were done with white lights as a control.*
- *They added figure showing different binding activity of blue light, white light or dark*

Response: Thanks for the confirmation.

13. *Figure 3C: Were EMSA assay repeated to reveal statistic significance? Binding with dark and blue light doesn't look significantly different. It would also be better to see blue light effects with N+ conditions.*
- *I did not see any error bars or repeated assays*

Response: Actually, the EMSA assays in this work were all repeated to reveal statistic significance (Fig. 3A and Fig. S19, Line 264-269), as shown below, with error bars.

The legend of Fig. 3A reads: “(A) DNA binding curves of NobZIP77 under the dark, white light (WL) or blue light (BL). The curves were generated by quantifying the relative amount of shifted bands of NobZIP77 bound to p*NoDGAT2B* in EMSA. Values shown as mean \pm SD. *: significant change ($p \leq 0.01$) versus darkness.”

The legend of Fig. S19 reads: “Fig. S19. Blue light reduces the binding of NobZIP77 to the *NoDGAT2B* promoter. EMSAs under dark (A), blue light (BL) (B) or white light (WL) (C) in the presence of 5 nM *NoDGAT2B* promoter DNA and varying amounts of the purified NobZIP77 protein (a 1:2 dilution series of NobZIP77 with the maximum concentration of 24mM). SS (super shift complex) was used for gray value quantification, and compared among dark, BL and WL.”

Correspondingly, the following was inserted into Results (Line 264-269): “To probe the effect of BL on NobZIP77-*NoDGAT2B* interaction, EMSA was performed in the dark, or under continuous illumination of white light (WL) or BL (Fig.S18; Methods). The shifting of NobZIP77-*pNoDGAT2B* band was observed starting at the NobZIP77 concentration of 0.012 mM under the dark, at 1.5 mM under BL, and at 0.75 mM under WL (Fig. 3A). Thus, the binding between NobZIP77 and *pNoDGAT2B*, which shuts down *NoDGAT2B*-mediated TAG assembly, can be relieved by BL (and to a lesser degree, by WL).”

Moreover, into Methods below was inserted (Line 607-618): “To probe the effect of WL and BL on the NobZIP77-*NoDGAT2B* interaction, EMSA was performed either in the dark or under continuous WL or BL illumination. The labeled DNA probes (5nM) and varying amounts of purified NobZIP77 (a 1:2 dilution series of NobZIP77, with the maximal concentration of 24mM) were incubated in the binding buffer. The

protein DNA mixtures were incubated at 25 °C for 60 min under darkness or continuous WL (400 $\mu\text{W cm}^{-2}$) or BL illumination (400 $\mu\text{W cm}^{-2}$ at 450 nm), and then analyzed on 6% native polyacrylamide gels. Gel electrophoresis was performed at 4°C for 60 min (100 V) under darkness or continuous illumination of WL (30 $\mu\text{W cm}^{-2}$) or BL (30 $\mu\text{W cm}^{-2}$). Finally, the DNA was electroblotted onto a nitrocellulose membrane and detected via chemiluminescence. Super shifted (SS) bands of NobZIP77 bound to pNoDGAT2B were used for gray value quantification with Image-Pro Plus 6.0. For each gel, SS gray value of 24 mM NobZIP77 was defined as 100%.”

14. *Figures 3E, F: It is good to see employing Raman technology for measuring relative TAG contents. Can a standard curve be created using cells with known TAG contents? It is also interesting to see effects of the green light: It is generally thought that green light is not used for photosynthesis (and I don't recall any green light receptors), and can be assumed no effects on lipid metabolism; however, it has effects under N-condition. How could it be?*
- They added data and comments on green light

Response: Thanks for the confirmation.

15. *Another point to note: Blue light still has positive effects in g77 KO line under N-condition, which leaves a question for additional blue light sensors. Figure 4A: Nuclear localization for g77 should be confirmed by nuclear staining using DAPI.*
- They did this

Response: Thanks for the confirmation.

16. *Localization of NoDGAT2B is not clear. It should be co-localized with sDer1:RFP fusion. If it is associated with lipid droplets, it can also be co-stained with Bodipy or Nile red.*
- They added this and it does co-localize

Response: Thanks for the confirmation.

17. *Figure 5: Experiments were done with different time schedules, and it is hard to directly compare data directly. Time values are misplaced in Figure 5A.*
- They added several time points for this, but it makes it even less conclusive, as now the response in the KO and knocking are not very different

Response: We believe the “KO and knocking are not very different” in fact means “KO and WT are not very different”. Similar to TAG content of the engineered

NobZIP77, huge changes usually present in partial time point(s). Please also refer to our response to Comment #1 (from Editor) and #6 (from Reviewer 2) above.

18. *Figure 6: Explain why blue light induction was done for KO mutants only. To claim its effect, the same treatment should be applied to WT.*

** The letters (B and C) are misplaced.*

- They fixed this

Response: Thanks for the confirmation.

19. *Figure S2: There are multiple bZIPs in algae and many more in plants. Explain how the representative bZIPs were chosen. Are they all related to blue light sensing?*

- They added some discussion but just a sentence saying that g77 was chosen because of DNA binding attributes

Response: We have already inserted into Discussion the explanation of how the representative bZIPs were chosen (and that only g77 is related to blue light sensing) (Line 974-978). As for “why g77 was chosen”, among the 125 identified TFs in *N. oceanica*, 16 TFs were predicted to regulate lipid-related genes (LRGs; Fig. 1A, Table S1); among them, g77 targets the largest number of LRGs (Fig. S1A, Table S1). Therefore, g77 was chosen for further studies in this work.

To further clarify this, we have inserted into Results (Line 73-86): “In the industrial oleaginous microalga *N. oceanica* IMET1, 125 TFs were identified via characteristic domains of plant TFs, and a global regulation network predicted links of 35 TFs to 2801 target genes¹¹. To pinpoint key TFs on N- induced TAG synthesis, we analyzed the transcript levels of these TFs over six time points (3, 4, 6, 12, 24 and 48 h) under N+ and N- conditions²³. Altogether, 32 TFs respond to N-, with eleven up-regulated and fourteen down-regulated (seven of them showing distinct trends of regulation among time points). Of them, sixteen TFs were predicted to regulate lipid-related genes (LRGs; Fig. 1A, Table S1)¹¹, and among them, scaffold00007.g77 (or g77, Genbank ID MT273120), which is down-regulated by N-, targets seven such genes that include both diacylglycerol acyltransferases (DGATs) and lipases (the enzymes play opposite roles on TAG metabolism (Fig. S1A, Table S1). As the activities of lipases and the DGAT are exactly opposite (oil degradation and assembly respectively), these results indicate g77 as a negative regulator of TAG synthesis.”.

20. *Figure S4C showed only TAG per total lipids. It is recommended to show a graph for total lipid contents.*

- They added this, not that different in any of the mutants

Response: Actually, NobZIP77-NoDGAT2B mainly targets TAG synthesis rather than total lipid metabolism, thus it is expected that no or little change in total lipid contents would take place in the mutants.

21. *In addition, expression patterns of g77 is more or less consistent as shown in Figure S4A, but its impact on TAG content (S4C) and TAG yields (S4D) are different (mostly at 0 and 24 hrs). Explain why.*

**The gene name was initially defined as g77, but NobZIP77 was also used later. It would be better to use consistent names.*

- They said the g77 is before knowing what it was BZIP77 after they confirmed what it was ... seems odd way to explain that

Response: As advised, we have cited both “g77” and “NobZIP77” at its first appearance (Line 91-92; since at that stage it was not clear yet that g77 belongs to the bZIP family), and then used “NobZIP77” at all instances afterwards.

22. *Overall English writing needs improvement by professional writers, since many statements were not clear and can be misleading; for example, in Abstract: "a blue-light-oil-induction strategy that exposes NobZIP77-knockout N. oceanica to white light and then to blue light doubles peak TAG productivity" What does it mean?*

It is not clear (and actually confusing) how the blue and white lights are used and how they are related, leaving a room for confusion and misunderstanding.

- They changed that sentence, but it is still just as confusing

Response: As advised, we have now improved the English writing throughout the manuscript. In particular, the sentence in Abstract is now changed to (Line 31-33):

“Moreover, a new oil-induction strategy was invented that doubles peak TAG productivity, by exposing the NobZIP77-knockout line to BL under N-.”

23. *In my version of the manuscript, references 1-38 are missing.*

- They show those reference are there

Response: Thanks for the confirmation.

24. *My overall impression is that the authors did add several of the experiments that reviewer 2 asked for. This new data did resolve several of reviewer 2 questions on BZIP77 nuclear localization, improved the BZIP77 promoter binding assays, and showed that DGAT2B localizes to lipid bodies. On these counts, I think reviewer 2 would be ok with the revisions. The data that BZIP is actually binding to the DGAT2B gene and regulating DGAT2B gene expression remains pretty tenuous, and the discussion on this is both speculative and at times confusing, perhaps because the data is just not all that consistent with the hypothesis. This is especially true in both the*

expression data for the BZIP77 gene and the DGAT2B expressing (they should be inversely correlated if BZIP77 is actually a DGAT2B DNA binding protein that represses transcription), and in the TAG accumulation data. It is hard to believe that this BZIP77 DNA protein (here called a TF), is really a key blue light regulator of TAG accumulation when there is only a 30% difference between the over-expression of the BZIP77 gene, and a mutant in DGAT2B that is supposedly the BZIP77 binding site? On that front I think reviewer 2 would still have objections to the conclusion that this BZIP77 protein is a DNA repressor of DGAT2B gene expression as a key blue light regulatory of TAG accumulation.

Response: Please refer to our response to Editor (Comment #1) and to Comment #4 (about the correlation in gene expression or TAG phenotype between *NobZIP77* mutants and *NoDGAT2B* mutants) above.

Reviewer #3 (Remarks to the Author):

25. The authors performed quite a number of additional experiments, which clearly improved this interesting manuscript and solidify the proposed mechanism of an aureochrome regulating TAG accumulation. There are still a few issues I'd like to be addressed.

1). Fig 1c, d: Please specify which of the symbols refers to the TAG yield and which to the TAG productivity.

Response: As advised, now we have added the description of TAG content and TAG yield into Fig. 1 Legend, as below:

Fig. 1. Identification of *NobZIP77* (g77) as a key TF on N- induced TAG synthesis. (C) TAG content and TAG yield of the *NobZIP77* overexpression and knockdown lines under N+. (D) TAG content and TAG yield of the *NobZIP77* knockout and complementation lines under N+. Bar chart, TAG content; scatter plot, TAG yield.

*26. 2). L109: "Furthermore, in *NobZIP77*-knockdown lines, under N+, TAG-associated C16:1 is 59.0-106.4% higher..." I appreciate that the authors present an inset in Fig. S4e, enlarging the values of the fatty acids with minor contents. Please include also here the data for 16:1, to see visually what is claimed above.*

Response: As advised, now the data for 16:1 has been included in Fig. S4E, as below:

27. 3). L118 and the following: The authors here present the data of the complementing lines, which they newly collected for the new manuscript. Just from a stylistic point of view: Instead of starting directly with explaining the complemented lines, created in the KO background, I would first start introducing the KO lines, stating that they behave similar as the silencing lines, and then move to the complementing line. Otherwise, one wonders where the KO lines are suddenly coming out of the blue.

Response: We appreciate these constructive comments. As advised, we have revised these sentences to (Line 120-134):

“To settle this problem, NobZIP77 was knocked out in *N. oceanica* (Methods). Under N+, knockout lines showed essentially no effects on microalgal growth (Fig. S6A). However, significant difference in TAG content was observed in NobZIP77-knockout lines. Specifically, versus wild type (WT) and under N-, TAG content was 178.1% and 51.0% higher at 0 h and 24 h for NobZIP77ko-1, while 160.0% and 49.5% higher at 0 h and 24 h for NobZIP77ko-2 (Fig. 1D and Fig. S6B). For TAG yield, NobZIP77 knockout resulted in 61.6% and 63.6% increase at 0 h respectively (52.7% and 44.4% at 24 h; under N-; Fig. S6C). Notably, TAG-associated C18:0 and C18:2 are 26.0-32.4% and 59.0-59.5% lower in the NobZIP77-knockout lines respectively under N+ (versus WT; Fig. S6D). As for TFA content, no difference was found versus WT via knockout, under either N- or N+ (Fig. S6E). These results indicated that the knockout lines behave similarly as the silencing lines. To further pinpoint its *in vivo* role, NobZIP77 was genetically complemented in its knockout lines (Fig. 1D; Fig. S5; Methods). Importantly, these complementation lines showed essentially identical phenotype to WT in *N. oceanica*. These results firmly pinpoint NobZIP77 as a TF that inhibits N-induced TAG synthesis.”

28. 4). L 125: “Specifically, versus wild type (WT) and under N-, TAG content was 51.0% and 178.1% higher at 0 h and 24 h for NobZIP77ko-1, while 160.0% and 49.5% higher

at 0 h and 24 h for NobZIP77ko-2. There are errors in the order of the % numbers. Both KO mutants basically behave the same, at 0h and 24 h, so please correct.

Response: We appreciate the comments. Now we revised these sentences to:

“Specifically, versus wild type (WT) and under N-, TAG content was 178.1% and 51.0% higher at 0 h and 24 h for NobZIP77ko-1, while 160.0% and 49.5% higher at 0 h and 24 h for NobZIP77ko-2.”

29. 5). Fig.2 E,F,G: If one looks at the TAG of the WT, it varies at T0 between 2.5 in Wt (E) and 7 (G) and at T24N- between ~12 and 14. This would make many of the observed significances between the WT and various mutants non-significant, if one would not calculate statistics based on the values for the Wt in the subpanels, but e.g. compare the TAG contents of the OE lines in (E) with the WT in (G) at T0. Therefore, I had previously asked about how biological replicates were obtained. Although there is variability, in lab conditions things like TAG (I don't talk about absolute values of transcriptomics, which is another story) should be relatively stable if algae are kept under identical conditions. What I suggest here is that the authors calculate the mean of the WT of all panels (so including all replicates) and then check if the mutants are significantly different.

Response: We appreciate the constructive comments. As advised, we have revised these figures, as shown below:

30. 6). Fig. S11: I don't understand why in B (TAG per total lipids), starting from 48 h, there is no significant difference between EV and Bi, while in C (%TAG per total lipids) suddenly there is. Please explain.

Response: We apologize for our typo. The correct y-axis of Fig. S12B (formerly Fig. S11B) should be “TAG content ($\mu\text{g}/\text{mg DW}$)” instead of “TAG per total lipids”. This explains the apparent discrepancy raised by reviewer (we are indebted to this comment). The corresponding Results and legends had been accurate.

31. 7). Growth experiments Fig. S4, S6 and Fig S13. When one only looks at the Wt and the EV control, one can see major differences in all three growth experiments. However, in all three experiments, especially S6 and S13, standard deviation is really small. Again, the authors do not specifically mention what kind of replicates they have done (despite my request in the previous reviewing process) and whether these are true biological replicates. Based on the minimum standard deviation, but the huge differences between Wt e.g. in S6 and S13, I am pretty convinced that the authors actually used one stock cultures, split it into three aliquots and immediately started the growth analysis. In lab growth analyses we here have some kind of a grey zone, as we always start from the same initial culture. However, while such an approach gives very nice results in terms of standard error for all kind of subsequent analyses (growth, gene expression, TAG, ...) it is not really what one considers as independent biological replicates. If, for instance, during such an experiment of 10 days a lamp for the growth light gets out of order at day 6, or the temperature in the culture room would change also at day 6, the authors would not notice this, as all their three replicates are there at the same time, taken from the same starting culture, and hence error bars would still

be extremely small. Instead, statistical tests might indicate some miraculous change in physiology at day 6 to 7 highly significant.

For the data presented here, the authors would normally need to take the mean of Fig. S6 and Fig. S13 for the Wt, and then the whole picture would look quite different. I believe that the results presented here reflect the true differences between Wt and the mutants, and therefore I do not request additional experiments with true independent biological replicates, but I would like to see that the authors specify in their methods how exactly they obtained their “replicates” for their analyses. And while the authors in the “Reporting summary” ticked “confirmed” for “A statement on whether measurements were taken from distinct samples or whether the same sample was measured repeatedly”, I don’t find any of such information in the manuscript, only the indication, “experiments were performed in n replicates”. This is still a serious point for me I this time would like to be addressed properly.

Response: As a matter of fact, for all our experiments involving algal culture, the samples came from three DIFFERENT and INDIVIDUAL columns (all data are provided in the “Source Data-NCOMMS-20-28862-T” folder). As for the question from Reviewer #3 about the growth data, our previous data (Fig. S4B) were produced by manual counting with cell counting chamber, which resulted in a larger error bar; in contrast, the more recent data (Fig. S6A and Fig. S13A) were produced automatically by the “Countstar algae” cell counter, which resulted in a minor error bar.

As advised, we have now inserted below very specific statements about the setup of biological replicates into Methods (Line 548-554) “For all *N. oceanica* experiments in this work, the culture was centrifuged in 3500 g for 5 min and then divided into individual 150ml-aliquots, i.e., in three separate biological replicates. The time point of these aliquots was regarded as 0 h. These aliquots were then separately cultured under various nitrogen levels or light wavelength conditions, and then sampled at 6 hours and up to 16 days after above treatments. For each time point, 1ml or 30 ml aliquots were sampled for growth measurement, Raman microspectroscopy, lipid profiling or RNA extraction.”

32. 8). L213: “As for TFA content, no changes were found in either double-knockout lines (Fig. S15E).” Not true. In contrast to the results with the NoDGAT2B-KO (Fig. S13) and the NobZIP77 KO (Fig. S6), in the double KO there is a significant difference (Fig. S15). Moreover, and this makes me really skeptical, the TFA content is only ~ 20 % in Fig. S15 compared to Fig. S13+FigS6. Please explain.

Response: For the first part of this comment, we agree with the reviewer. We have now revised the sentence as: “As for TFA content, no changes were found in either double-knockout lines (except for 48 h under N-; Fig. S16E).”

For the second part of this comment, we apologize for our error in confusing the units. The y-axis in Fig. S16E (formerly Fig. S15E) should have been “TFA content (mg/g DW)”, instead of “TFA content (mg/L)”. This is now corrected, as shown below.

33. 9). L250: The 11.7 % less TAG under BL in the NobZIP77ko-1 line: Is this significant compared to the Wt?

Response: The p value (t -test) TAG under BL under N- between NobZIP77ko-1 and WT is < 0.05 , thus it is significant. We have now inserted the p value into the sentence as below,

“Moreover, in NobZIP77ko-1, TAG content is reduced by 11.78% under BL plus N-, revealing a temporal lag of TAG content in response to BL under N- versus WT ($p < 0.05$, Fig. 3D; Fig. 3F)”

34. 10). I like the model of the authors how BL could lead to the increase of lipids under N-. The lead idea here is that under N- there is less chlorophyll shielding the nucleus and hence more blue light reaches it and activates the aureochrome LOV domain. However, there is no difference in chlorophyll amount even after 12 h N- (Fig. 4 D). Unfortunately, the authors only provide normalized data for NoDGAT2B in their manuscript for the Wt. If I interpret the FPKM values in Table S1 correctly, there is already a pronounced difference in noDGAT2B transcription between N- and N+

conditions at much earlier time points, which would not fit to the proposed hypothesis. Can the authors explain this?

Response: We appreciate this thoughtful comment. It is possible that there are additional regulatory mechanisms that underlie the change of *NoDGAT2B* transcript abundance at the very early stage of N-. As future directions, we have starting to probe the complete regulatory network of *NoDGAT2B* (and those for the other seven *NoDGAT2s* that we have proved as functional (Xin et al. 2017; Xin et al. 2019; Wang et al. 2020).

35. 11). *There is a recent study demonstrating that knocking out an aureochrome photoreceptor in diatoms has a massive effect on the general transcriptome, and here especially also on the lipid metabolism (Mann et al. 2020, iScience). This links those results to the ones presented here and shows that aureochromes may fulfill similar functions in triggering the metabolism in distinct groups within the stramenopiles. The corresponding article should be shortly discussed within the framework of the presented results, and the statement L369: “On the other hand, aureochromes, although known to regulate photomorphogenesis, sexual cycle and biological rhythm, have not been implicated in the synthesis of carbon-storage compounds“ should be modified.*

Response: We appreciate this suggestion and have cited this study. The statement L369 has now been changed to: “On the other hand, aureochromes, although known to regulate photomorphogenesis, sexual cycle, and biological rhythm, have rarely been implicated in the synthesis of carbon-storage compounds (although it was reported that strong metabolic shift takes place upon red/blue light transitions in diatoms^{42, 43}).”

References cited in this response letter:

- Boyle, N. R., M. D. Page, B. Liu, I. K. Blaby, D. Casero, J. Kropat, S. J. Cokus, A. Hong-Hermesdorf, J. Shaw, S. J. Karpowicz, S. D. Gallaher, S. Johnson, C. Benning, M. Pellegrini, A. Grossman, and S. S. Merchant. 2012. 'Three acyltransferases and nitrogen-responsive regulator are implicated in nitrogen starvation-induced triacylglycerol accumulation in *Chlamydomonas*', *J Biol Chem*, 287: 15811-25.
- Kwon, S., N. K. Kang, H. G. Koh, S. E. Shin, B. Lee, B. R. Jeong, and Y. K. Chang. 2018. 'Enhancement of biomass and lipid productivity by overexpression of a bZIP transcription factor in *Nannochloropsis salina*', *Biotechnol Bioeng*, 115: 331-40.
- Li, J., D. Han, D. Wang, K. Ning, J. Jia, L. Wei, X. Jing, S. Huang, J. Chen, Y. Li, Q. Hu, and J. Xu. 2014. 'Choreography of transcriptomes and lipidomes of *Nannochloropsis* reveals the mechanisms of oil synthesis in microalgae', *Plant Cell*, 26: 1645-65.

- Ngan, C. Y., C. H. Wong, C. Choi, Y. Yoshinaga, K. Louie, J. Jia, C. Chen, B. Bowen, H. Cheng, L. Leonelli, R. Kuo, R. Baran, J. G. García-Cerdán, A. Pratap, M. Wang, J. Lim, H. Tice, C. Daum, J. Xu, T. Northen, A. Visel, J. Bristow, K. K. Niyogi, and C. L. Wei. 2015. 'Lineage-specific chromatin signatures reveal a regulator of lipid metabolism in microalgae', *Nat Plants*, 1: 15107.
- Takahashi, F., D. Yamagata, M. Ishikawa, Y. Fukamatsu, Y. Ogura, M. Kasahara, T. Kiyosue, M. Kikuyama, M. Wada, and H. Kataoka. 2007. 'AUREOCHROME, a photoreceptor required for photomorphogenesis in stramenopiles', *Proc Natl Acad Sci U S A*, 104: 19625-30.
- Wang, D., K. Ning, J. Li, J. Hu, D. Han, H. Wang, X. Zeng, X. Jing, Q. Zhou, X. Su, X. Chang, A. Wang, W. Wang, J. Jia, L. Wei, Y. Xin, Y. Qiao, R. Huang, J. Chen, B. Han, K. Yoon, R. T. Hill, Y. Zohar, F. Chen, Q. Hu, and J. Xu. 2014. 'Nannochloropsis genomes reveal evolution of microalgal oleaginous traits', *PLoS Genet*, 10: e1004094.
- Wang, X., Y. Xin, L. Ren, Z. Sun, P. Zhu, Y. Ji, C. Li, J. Xu, and B. Ma. 2020. 'Positive dielectrophoresis-based Raman-activated droplet sorting for culture-free and label-free screening of enzyme function in vivo', *Sci Adv*, 6: eabb3521.
- Xin, Y., Y. Lu, Y. Y. Lee, L. Wei, J. Jia, Q. Wang, D. Wang, F. Bai, H. Hu, Q. Hu, J. Liu, Y. Li, and J. Xu. 2017. 'Producing designer oils in industrial microalgae by rational modulation of co-evolving yype-2 diacylglycerol acyltransferases', *Mol Plant*, 10: 1523-39.
- Xin, Y., C. Shen, Y. She, H. Chen, C. Wang, L. Wei, K. Yoon, D. Han, Q. Hu, and J. Xu. 2019. 'Biosynthesis of triacylglycerol molecules with a tailored PUFA profile in industrial microalgae', *Mol Plant*, 12: 474-88.
- Yamaoka, Y., S. Shin, B. Y. Choi, H. Kim, S. Jang, M. Kajikawa, T. Yamano, F. Kong, B. Légeret, H. Fukuzawa, Y. Li-Beisson, and Y. Lee. 2019. 'The bZIP1 transcription factor regulates lipid remodeling and contributes to ER stress management in *Chlamydomonas reinhardtii*', *Plant Cell*, 31: 1127-40.

Finally, thank you and the three reviewers for considering this revision, and we look forward to your reply.

With warmest regards,

Jian XU, Ph.D, Professor and Director, Single-Cell Center, Qingdao Institute of BioEnergy and Bioprocess Technology, Chinese Academy of Sciences; Senior Editor, *mSystems*; Postal:

189 Songling Rd, Qingdao, Shandong, China 266101; Email: xujian@qibebt.ac.cn Tel/Fax:
86-532-80662651/2654

Reviewers' Comments:

Reviewer #3:

Remarks to the Author:

The authors addressed all my previous concerns in this third submission satisfactorily.

Bernard Lepetit